



# Characterizing model errors in chemical transport modelling of methane: Using GOSAT XCH4 data with weak constraint four-dimensional variational data assimilation

Ilya Stanevich[1], Dylan B. A. Jones[1], Kimberly Strong[1], Martin Keller[1], Daven K. Henze[2,3], Robert J. Parker[4,5], Hartmut Boesch[4,5], Debra Wunch[1], Justus Notholt[6], Christof Petri[6], Thorsten Warneke[6], Ralf Sussmann[7], Matthias Schneider[8], Frank Hase[8], Rigel Kivi[9], Nicholas M. Deutscher[10], Voltaire A. Velazco[10], Kaley A. Walker[1], and Feng Deng[1]

[1]Department of Physics, University of Toronto, Toronto, Ontario, Canada
[2]Department of Mechanical Engineering, University of Colorado Boulder, Boulder, CO, USA
[3]California Institute of Technology, Pasadena, CA, USA
[4]Earth Observation Science, Department of Physics and Astronomy, University of Leicester, Leicester, UK
[3]National Centre for Earth Observation (NCEO), University of Leicester, Leicester, UK
[6]Institute of Environmental Physics, University of Bremen, Bremen, Germany
[7]Karlsruhe Institute of Technology (KIT), Institute of Meteorology and Climate Research (IMK-IFU), Garmisch-Partenkirchen, Germany
[8]Karlsruhe Institute of Technology (KIT), Institute of Meteorology and Climate Research (IMK-ASF), Karlsruhe, Germany
[9]Finnish Meteorological Institute, Sodankylä, Finland
[10]Centre for Atmospheric Chemistry, School of Chemistry, University of Wollongong, Wollongong, NSW, Australia

*Correspondence to:* Ilya Stanevich (stanevich@atmosp.physics.utoronto.ca)

**Abstract.**

We examined biases in the global GEOS-Chem chemical transport model for the period of February-May 2010 using weak constraint (WC) four-dimensional variational (4D-Var) data assimilation and dry-air mole fractions of $CH_4$ ($XCH_4$) from the Greenhouse gases Observing SATellite (GOSAT). The ability of the observations and the WC 4D-Var method to mitigate

model errors in $CH_4$ concentrations was first investigated in a set of observing system simulation experiments (OSSEs). We then assimilated the GOSAT $XCH_4$ retrievals and found that they were capable of differentiating the vertical distribution of model errors and of removing a significant portion of biases in the modelled $CH_4$ state. In the WC 4D-Var assimilation, corrections were added to the modeled $CH_4$ state at each model time step to account for model errors and improve the model fit to the assimilated observations. Compared to the conventional strong constraint (SC) 4D-Var assimilation, the WC method was

able to significantly improve the model fit to independent observations. Examination of the WC state corrections suggested that a significant source of the model errors was associated with discrepancies in the model $CH_4$ in the stratosphere. The WC state corrections also suggested that the model vertical transport in the troposphere at mid- and high-latitudes is too weak. The problem was traced back to biases in the uplift of $CH_4$ over the source regions in eastern China and North America. In the tropics, the WC assimilation pointed to the possibility of biased $CH_4$ outflow from the African continent to the Atlantic in the

mid-troposphere. The WC assimilation in this region would greatly benefit from glint observations over the ocean to provide additional constraints on the vertical structure of the model errors in the tropics. We also compared the WC assimilation at the





$4° \times 5°$ and $2° \times 2.5°$ horizontal resolutions and found that the WC corrections to mitigate the model errors were significantly larger at $4° \times 5°$ than at $2° \times 2.5°$ resolution, indicating the presence of resolution-dependent model errors. Our results illustrate the potential utility of the WC 4D-Var approach for characterizing model errors. However, a major limitation of this approach is the need to better characterize the specified model error covariance in the assimilation scheme.

## 5 1 Introduction

Atmospheric concentrations of methane ($CH_4$), the second most important anthropogenic greenhouse gas, have been rapidly raising since 1850 (Etheridge et al., 1992). However, atmospheric measurements in recent decades show that the rate of $CH_4$ increase in the atmosphere has varied and its behaviour is not well understood (Dlugokencky et al., 2009). Significant effort has been put into characterizing surface emissions of $CH_4$ in order to attribute its recent trends. In this context, a number of satellites

have been launched to measure atmospheric $CH_4$ in order to constrain its sources. These include Envisat carrying the Scanning Imaging Absorption Spectrometer for Atmospheric Cartography (SCIAMACHY) (Schneising et al., 2011), the Greenhouse Gases Observing Satellite (GOSAT) carrying the Thermal And Near-infrared Sensor for carbon Observation Fourier Transport Spectrometer (TANSO-FTS) (Kuze et al., 2009), Sentinel-5p with the Tropospheric Monitoring Instrument (TROPOMI) on-board (Veefkind et al., 2012), and the Greenhouse Gases Satellite (GHGSat). Proposed missions include the Methane Remote

Sensing Lidar Mission (MERLIN) (Kiemle et al., 2014), GOSAT-2 (Nakajima et al., 2017), the Geostationary Carbon Cycle Observatory (GeoCARB) (Polonsky et al., 2014) and the recently-announced MethaneSat. However, current regional $CH_4$ emissions remain largely uncertain (e.g., Saunois et al., 2016). One of the biggest challenges for reducing uncertainty on emission estimates is the relatively weak signal of emissions in the atmospheric column of $CH_4$, which puts tight requirements on the accuracy of satellite measurements. However, while future satellite instruments and improved spectroscopy are expected

to provide better $CH_4$ measurements, errors in the atmospheric models used to simulate $CH_4$ remain poorly characterized. While random model errors can be accounted for in flux inversion analyses, the impact of biases in chemistry and transport are often neglected. In the case of $CH_4$, which is a relatively long-lived gas with an atmospheric lifetime of about 9 years (Prather et al., 2012), chemistry plays a critical role in long-term trends (McNorton et al., 2016), whereas transport, alone or coupled with chemistry, defines how total surface emissions are distributed on a regional scale. Therefore, transport errors, such as those

produced by numerical advection schemes, biases and uncertainties of meteorological fields, and parametrization of sub-grid scale processes may significantly undermine our ability to use models to relate emissions to atmospheric observations, and thus our ability to improve $CH_4$ emission estimates (Prather et al., 2008; Locatelli et al., 2015; Patra et al., 2011).

One potential solution is to apply a bias correction to the model in the context of the inversion analysis. Simple bias correction schemes with uniform or latitudinally dependent bias estimates have been attempted before (Bergamaschi et al., 2009; Fraser

et al., 2013; Monteil et al., 2013; Alexe et al., 2015; Locatelli et al., 2015), mostly to correct poor description of the modeled stratosphere. Here we explore the utility of a "weak constraint" (WC) four-dimensional variational (4D-Var) data assimilation method to characterize forward model errors. In contrast to the traditional "strong constraint" (SC) 4D-Var method, the WC scheme does not assume that the model is perfect. The WC 4D-Var method was introduced by Sasaki (1970) and used in





numerical weather prediction (NWP) models by Derber (1989), Zupanski (1997) and Trémolet (2006, 2007). It was first applied by Keller (2014) in the GEOS-Chem simulation of atmospheric carbon monoxide (CO) to characterize model bias. One of the first attempts to apply bias correction in chemical data assimilation was done in the framework of the suboptimal Kalman filter by Lamarque et al. (2004), who used the bias estimation approach of Dee and Da Silva (1998) to constrain

the CO state using measurements from the Measurement of Pollution in the Troposphere (MOPITT) instrument. The study pointed to the possibility of errors in the model vertical transport, however most of the estimated biases were attributed to poor a priori estimates of CO surface emissions in the model. The major challenge for this type of analysis for $CH_4$ is the limited information available about the global vertical distribution of $CH_4$ in the atmosphere. There are satellite observations that contain information about the $CH_4$ distribution in the middle and upper troposphere, such as the thermal infrared $CH_4$

retrievals from the Tropospheric Emission Spectrometer (TES) on-board the NASA Aura satellite (Worden et al., 2012), or in the stratosphere, such as the solar occultation measurements from the Atmospheric Chemistry Experiment Fourier Transform Spectrometer (ACE-FTS, Bernath et al., 2005) on-board SCISAT. However, the accuracy of these measurements, based on validation studies (for example, De Mazière et al., 2008; Wecht et al., 2012), may not be sufficient to detect model errors. The most accurate satellite measurements are those of the total dry-air mole fraction of $CH_4$ in the total atmospheric column

($XCH_4$) obtained by TANSO-FTS on-board GOSAT. However, these measurements provide less vertical information on $CH_4$ than those from TES or ACE-FTS, although the latter are less sensitive to surface emissions. Highly accurate aircraft $CH_4$ profile measurements would be an ideal source of information, but they are limited in space and time. We explore here the information content of GOSAT $CH_4$ observations and show that despite being designed to constrain surface emissions, they contain sufficient information to help characterize possible model errors. We assimilate the GOSAT observations using the WC

4D-Var data assimilation approach to estimate biases in GEOS-Chem. This approach is shown to provide a valuable tool for diagnosing and determining the origin of model errors.

This paper is organized as follows. Section 2 gives an overview of the forward model, the observations, and the WC 4D-Var method. It also contains a description of the various sensitivity studies conducted through a series of Observing System Simulation Experiments (OSSEs). In Sect. 3, we present the results of the sensitivity experiments, as well as the results of the

assimilation of real GOSAT observations. Sect. 4 provides an interpretation of the pattern of model biases estimated from the GOSAT assimilation. Finally, conclusions are given in Sect. 5.

## 2   Data and Methods

### 2.1   The GEOS-Chem Model

For all assimilation experiments we use version v35 of the GEOS-Chem adjoint, which is based on version v8-02-01 of the for-

ward model, with updates up to v9-02 (Henze et al., 2007). The GEOS-Chem CTM (www.geos-chem.org) is driven by archived meteorological fields from the Goddard Earth Observing System (GEOS-5.2.0) produced by the NASA Global Modelling and Assimilation Office (GMAO). The meteorological fields are regridded from their native resolution of $0.5° \times 0.67°$ with 72 vertical levels to $4° \times 5°$ and $2° \times 2.5°$ with 47 vertical levels. The vertical grid spacing in the troposphere varies from about 150



m in the lower part to about 1 km in the upper part. CH$_4$ is advected using the multi-dimensional flux-form semi-lagrangian (FFSL) scheme by Lin and Rood (1996). Convection is implemented based on the relaxed Arakawa-Schubert scheme (Moorthi and Suarez, 1992). The model uses a simple treatment of turbulent mixing in the boundary layer by instantaneously mixing species from the surface to the top of the planetary boundary layer (PBL). The GEOS-Chem CH$_4$ sources and sinks used here

are described in detail in Wecht et al. (2014). Anthropogenic CH$_4$ sources include emissions from natural gas and oil extraction, coal mining, livestock, landfills, waste water treatment, rice cultivation, biofuel burning and other minor sources and are based on the 2004 anthropogenic inventory from the Emission Database for Global Atmospheric Research (EDGAR) v4.2 (European Commission Joint Research Centre/Netherlands Environmental Assessment Agency, 2009). Natural CH$_4$ sources include wetland emissions after Kaplan (2002) and Pickett-Heaps et al. (2011), termite emissions (Fung et al., 1991) and open

fire emissions from the daily Global Fire Emissions Database Version 3 (GFED3) (van der Werf et al., 2010; Mu et al., 2011). The CH$_4$ emissions at $4° \times 5°$ and $2° \times 2.5°$ resolutions are slightly different due to the dependence of wetland emissions on the meteorological fields. Therefore, for consistency of the analysis of model errors, the emissions were regridded from the coarser to the finer resolution. The main loss of CH$_4$ (about 90% of the total loss) in the atmosphere is due to oxidation by OH, with the remaining 10% sink mainly due to soil absorption and oxidation in the stratosphere. CH$_4$ chemistry is performed in

off-line mode in which changes in CH$_4$ concentrations do not feed back on other species. Tropospheric OH fields in the model are prescribed as a three-dimensional monthly mean climatology from a tropospheric chemistry simulation in GEOS-Chem v5-03 (Park et al., 2004). Stratospheric CH$_4$ loss frequencies are from archived climatology of the NASA Global Modelling Initiative (GMI) model (Murray et al., 2012).

     The adjoint model is described by Henze et al. (2007) and has been used for assimilation of CH$_4$ observations by Wecht et al.

(2012, 2014), Turner et al. (2015), Bousserez et al. (2016), and Tan et al. (2016). For the analysis presented here, we focus on the period of 1 February 2010 to 31 May 2010. The CH$_4$ fields were spun up at a resolution of $4° \times 5°$ and $2° \times 2.5°$ for about 5.5 years until July 2009. From July 2009 to January 2010 we assimilated the GOSAT Proxy XCH$_4$ retrievals (Parker et al., 2015) to obtain monthly mean emission estimates at $4° \times 5°$ resolution. The optimized emissions were then regridded and used to perform forward model simulations at $2° \times 2.5°$ resolution for the same period from July 2009 to January 2010. The

updated model fields on 1 February 2010 at both model resolutions were taken as initial condition for the analysis period. As a result, the initial conditions at both resolutions contain similar amounts of CH$_4$ in the atmosphere. However, CH$_4$ is distributed differently, reflecting the balance between emissions and transport at each model resolution.

## 2.2   Measurements

### 2.2.1   GOSAT

We obtain information about the CH$_4$ distribution in the atmosphere from XCH$_4$ retrievals from the TANSO-FTS on-board GOSAT, which has a three-day repeat orbit period. The instrument has a surface footprint of 10.5 km in diameter and records spectra at about 13:00 local time. We use version 5.2 of the University of Leicester (UoL) GOSAT Proxy XCH$_4$ data product. The retrieval algorithm is explained in detail in Parker et al. (2011, 2015). In this algorithm, simplified spectral retrievals of



XCO$_2$ and XCH$_4$ are obtained in spectral bands centred at 1.65 $\mu$m and 1.61 $\mu$m, respectively. The final total column-averaged dry-air mole fraction of CH$_4$ is obtained by multiplying the retrieved XCH$_4$/XCO$_2$ ratio by modelled XCO$_2$ fields. This is useful for cancelling out common spectral features caused by light path modifications due to thin clouds, aerosol scattering, and instrumental artefacts in close spectral bands. However, a reliable knowledge of the XCO$_2$ data is required. The Proxy method

provides significantly greater observational coverage, especially in tropical areas, compared to the "full-physics" retrievals. The weakness of the approach is in the fact that the modeled CO$_2$ fields may still contain biases that are not accounted for in the final XCH$_4$ product. Version 5.2 of the XCH$_4$ data does not include retrievals from spectra recorded over oceans (glint observations). This is in contrast to the later versions 6 and 7 which, however, use the same algorithm for XCH$_4$ retrievals over land. Furthermore, in our analysis we exclude all retrievals over Greenland and poleward of 75° (including retrievals over

snow).

The original XCH$_4$ retrievals utilized XCO$_2$ fields based on the median of three models: GEOS-Chem (from the University of Edinburgh), LMDZ/MACC-II, and CarbonTracker (National Oceanic and Atmospheric Administration (NOAA)), that was smoothed with GOSAT CO$_2$ averaging kernels (Parker and GHG-CCI the group, 2016). CO$_2$ fields in all three models were produced by assimilating in-situ surface CO$_2$ observations. In this work, we replaced the original modeled CO$_2$ fields with

optimized CO$_2$ fields from a GEOS-Chem CO$_2$ surface flux assimilation analysis that used GOSAT XCO$_2$ retrievals over land (Deng et al., 2014). For the period of interest (February-May 2010), the XCH$_4$ retrievals using both proxy CO$_2$ fields are unbiased against each other with a scatter of 3 ppb and a correlation of $R = 0.99$. Figure 1 shows the mean difference between the two products and points to some systematic regional discrepancies of up to 8 ppb, however, in 90% of all 4° × 5° grid cells covered by the measurements, the differences are smaller than 3 ppb. Sensitivity tests that were conducted showed that

a posteriori inversion results using the new CO$_2$ fields generally produced a comparable fit to independent CH$_4$ measurements from the Total Carbon Column Observing Network (TCCON, Wunch et al., 2011) and from the NOAA Earth System Research Laboratory (ESRL) global cooperative air sampling network (Dlugokencky et al., 2016). The use of the alternative CO$_2$ fields did not change any of the findings about model errors in our study. There may still be unidentified biases in both retrieval products. However, the fact that both CO$_2$ fields were obtained using different methodologies gives us confidence in our

results.

GOSAT CH$_4$ retrievals contain about 1 degree of freedom for signal (DOFS) and have relatively flat averaging kernels in the troposphere that slowly decrease in the stratosphere. Therefore, they contain little vertical information about the atmosphere at the time of measurement. We use these averaging kernels to smooth the GEOS-Chem CH$_4$ fields and map them into the measurement space of the GOSAT retrievals. The absence of vertical information in the measurements is a challenge for

constraining the 3D structure of model errors. However, we expect vertical structure to emerge from atmospheric transport patterns. For example, the majority of the CH$_4$ mass enters the North American domain through the western boundary in the jet stream in the upper troposphere. Therefore, XCH$_4$ observations over North America would be more sensitive to past CH$_4$ concentrations in the upper troposphere upwind over the Pacific and Asia.

Errors in GOSAT Proxy XCH$_4$ retrievals with the original XCO$_2$ data were assessed against co-located TCCON ground-

based measurements by Hewson et al. (2015). That validation study found that GOSAT retrievals contain random errors of





12.55 ppb and systematic errors of 4.8 ppb (although per-site biases ranged from -2.15 ppb (Wollongong) to 13.44 ppb (Garmisch)). However, errors away from TCCON sites could be larger. Overall, GOSAT and TCCON were highly correlated with a correlation coefficient of 0.86. Buchwitz et al. (2017) obtained similar results with random errors of 11.9 ppb and systematic errors of 5.7 ppb for GOSAT Proxy $XCH_4$ retrievals against co-located TCCON retrievals. Such precision could be enough in many regions of the world to improve knowledge about $CH_4$ a priori surface emissions. However, the presence of potential model errors significantly undermines this assumption. Therefore, here we explore the potential utility of the weak constraint 4D-Var scheme to discern model biases using the $XCH_4$ data.

Figure 2 shows the mean $XCH_4$ fields from February to May 2010 modeled by GEOS-Chem at $4° \times 5°$ resolution (top panel) and as measured by GOSAT (bottom panel). A number of features in modeled $XCH_4$ can be identified. There is a clear inter-hemispheric difference with smaller $XCH_4$ in the Southern hemisphere. Enhanced $XCH_4$ concentrations can be observed over China, India, equatorial Africa and South America, with weaker signals also present over Europe and the eastern US, which, to first approximation, are related to local surface $CH_4$ emissions. Major features generally agree between the modelled and observed $XCH_4$ fields, however there are also a number of discrepancies that are discussed in Section 3.

### 2.2.2 Validation Data

The a priori and constrained model $CH_4$ fields are validated against in situ NOAA-ESRL $CH_4$ measurements (Dlugokencky et al., 2016) and measurements from the third HIAPER Pole-to-Pole Observations (HIPPO-3) aircraft campaign (Wofsy et al., 2011), against TCCON ground-based $XCH_4$ retrievals (Wunch et al., 2011), and ACE-FTS space-based $CH_4$ retrievals (Boone et al., 2005).

The NOAA network operates by collecting air flask samples which are later analysed by gas chromatography with flame ionization detection. At stationary sites, samples are collected once per week. Shipborne samples from sites in the Pacific Ocean and the South China Sea are collected once every three weeks and weekly, respectively, per latitude band. Measurements are reported relative to the NOAA X2004A $CH_4$ scale. The absolute uncertainty of the scale is 0.2% (about 3 ppb), and measurements are reproducible to within 1-3 ppb.

Airborne data are provided by the HIPPO-3 aircraft campaign which took place between 20 March 2010 and 20 April 2010. The campaign sampled the atmospheric curtain from the North Pole to the coast of Antarctica through the central Pacific Ocean and from the surface to 14 km altitude. We used $CH_4$ measurements performed by a quantum cascade laser spectrometer (QCLS) at 1 Hz frequency. QCLS measurements have precision of 0.5 ppb and accuracy of 1 ppb, while the mean bias relative to simultaneous flask-based measurements is 0.44 ppb (Santoni et al., 2014). We exploited the Merged 10-second Meteorology, Atmospheric Chemistry, and Aerosol Data product (Wofsy et al., 2012), which was derived from 1-sec measurements by applying a median filter.

TCCON is a global network of ground-based high-resolution Fourier transform infrared (FTIR) spectrometers retrieving $XCH_4$ from solar absorption spectra in the near-infrared band. We used the GGG2014 version of TCCON $XCH_4$ data from multiple stations around the globe (Kivi and Heikkinen, 2016; Kivi et al., 2017; Blumenstock et al., 2017; Griffith et al., 2017; Hase et al., 2017; Notholt et al., 2017; Sherlock et al., 2017; Sussmann and Rettinger., 2017; Warneke et al., 2017; Wennberg





et al., 2017b, a). The estimated accuracy and precision of XCH$_4$ retrievals are less than 0.5% and 0.3%, respectively (Wunch et al., 2015). Retrievals are bias corrected based on comparisons with calibrated aircraft and AirCore profiles.

ACE-FTS on-board SCISAT performs solar occultation measurements over a range of tangent heights. The satellite makes 15 occultations for both sunrise and sunset per day separated by about 24° in longitude. Measurements cover an altitude range

from the cloud tops in the upper troposphere up to 150 km. Spectra are recorded continuously during 2-sec scans, which implies that the altitude and tangent point changes slightly during the scan. As a result, the instrument has low horizontal resolution of about 300 km in the limb direction. The vertical resolution determined by the instrument field-of-view is about 3 km at a tangent point 3000 km away from the satellite. However, vertical sampling ranges from 2 to 6 km depending on viewing geometry. In this study, we use the most recent v3.6 CH$_4$ retrievals with geolocation information (Boone et al., 2013;

Waymark et al., 2014). Version 3.6 only differs from version 3.5 in that a local computer was used to process v3.5 while a shared supercomputing system was used for v3.6. Olsen et al. (2017) compared ACE-FTS v3.5 and MIPAS CH$_4$ vertical profiles coincident with TANSO-FTS measurements, and found small differences above the tropopause except in the tropics. The mean differences were larger than 20% below about 450 hPa, within 5% between 450 and 40 hPa, and larger than 5% above 40 hPa.

## 2.3 The Weak Constraint 4D-Var Approach

The GEOS-Chem forward model can be represented by an operator $M$ which acts on the model state $\mathbf{x}_i$ and model parameters $\mathbf{p}$ at time step $i$ to produce a new model state $\mathbf{x}_{i+1}$ at the next time step as follows:

$$\mathbf{x}_{i+1} = M(\mathbf{x}_i, \mathbf{p}). \tag{1}$$

The model state at each time step is represented by a 3D field of CH$_4$ concentrations, and model parameters usually include

CH$_4$ emissions from each model surface grid cell. In Eq. 1, it is assumed that there are no errors in propagating the state forward in time. This is the assumption that is employed in standard 4D-Var, which is also referred to as "strong constraint" 4D-Var because the model trajectory is used as a strong constraint in the optimization. However, as described by Trémolet (2006), Eq. 1 can be modified to account for model errors by adding corrections $\mathbf{u}_{i+1}$ to the CH$_4$ state at time step $i+1$ so that the model forecast becomes

$$\mathbf{x}_{i+1} = M(\mathbf{x}_i, \mathbf{p}) + \mathbf{G}\mathbf{u}_{i+1}, \tag{2}$$

where $\mathbf{G}$ is an operator that maps corrections $\mathbf{u}_i$ into the model state. Here the corrections $\mathbf{u}_{i+1}$ are referred to as forcing terms, which is distinct from the adjoint forcing commonly used in 4D-Var. The operator $\mathbf{G}$ can also be understood as a mask that defines the spatial regions in the 3D model state where corrections need to be applied. Hence, the second term in Eq. 2 represents additional sources and sinks of CH$_4$ in the region of the atmosphere defined by $\mathbf{G}$. The 4D-Var problem to estimate

surface emissions is transformed into a 3D sources and sinks estimation problem in which a cost function is minimized with





respect to both model parameters $\mathbf{p}$ and state corrections $\mathbf{u}$. In this case, the WC 4D-Var cost function is expressed as

$$J(\mathbf{p},\mathbf{u}_i) = \sum_{i=0}^{N} \frac{1}{2}(\mathbf{y}_i - \mathbf{H}\mathbf{x}_i)^T \mathbf{R}_i^{-1}(\mathbf{y}_i - \mathbf{H}\mathbf{x}_i) +$$
$$+ \frac{1}{2}(\mathbf{p}-\mathbf{p}_a)^T \mathbf{B}^{-1}(\mathbf{p}-\mathbf{p}_a) + \sum_{i=1}^{N} \frac{1}{2}\mathbf{u}_i^T \mathbf{Q}_i^{-1}\mathbf{u}_i, \tag{3}$$

where $N$ is the number of one hourly time steps, $\mathbf{y}_i$ is the vector of XCH$_4$ observations during the time step $i$, $\mathbf{H}$ is the observation operator that maps the modelled CH$_4$ state into measurement space at the location of GOSAT XCH$_4$ observations,

$\mathbf{R}_i$ represents the observation error covariance matrix, $\mathbf{p}_a$ is the a priori estimates of model parameters, $\mathbf{B}$ is the a priori error covariance matrix, and $\mathbf{Q}_i$ defines the a priori model error covariance matrix. As described by Trémolet (2006), $\mathbf{u}_i$ can be considered to represent model errors on time scales as short as each model time step or as long as the full assimilation period, and is assumed to be constant over the appropriate interval. In the case where the forcing is estimated over the full assimilation window, the optimized forcing will represent a constant model bias over the whole model trajectory.

The WC 4D-Var approach was implemented into the GEOS-Chem model by Keller (2014) and here we describe that approach. The cost function (Eq. 3) is minimized subject to the equality constraints in Eq. 2 by adding the model constraints to the cost function to create the following Lagrangian function:

$$\mathcal{L}(\mathbf{p},\mathbf{x}_i,\boldsymbol{\lambda}_i,\mathbf{u}_i) = \frac{1}{2}(\mathbf{p}-\mathbf{p}_a)^T \mathbf{B}^{-1}(\mathbf{p}-\mathbf{p}_a) +$$
$$\sum_{i=0}^{N} \frac{1}{2}(\mathbf{y}_i - \mathbf{H}\mathbf{x}_i)^T \mathbf{R}_i^{-1}(\mathbf{y}_i - \mathbf{H}\mathbf{x}_i) + \sum_{i=1}^{N} \frac{1}{2}\mathbf{u}_i^T \mathbf{Q}_i^{-1}\mathbf{u}_i -$$
$$\sum_{i=1}^{N} \boldsymbol{\lambda}_i^T [\mathbf{x}_i - M(\mathbf{x}_{i-1},\mathbf{p}) - \mathbf{G}\mathbf{u}_i], \tag{4}$$

where $\lambda_i$ are the Lagrange multipliers. We define gradients of the Lagrangian $\mathcal{L}$ with respect to $\boldsymbol{\lambda}_i$, $\mathbf{x}_i$, $\mathbf{p}$ and $\mathbf{u}_i$ by the following

system of equations:

$$\frac{\partial \mathcal{L}}{\partial \boldsymbol{\lambda}_i} = \mathbf{x}_i - M(\mathbf{x}_{i-1},\mathbf{p}) - \mathbf{G}\mathbf{u}_i, \tag{5}$$

$$\frac{\partial \mathcal{L}}{\partial \mathbf{x}_i} = -\mathbf{H}^T \mathbf{R}_i^{-1}[\mathbf{y}_i - \mathbf{H}\mathbf{x}_i] - \boldsymbol{\lambda}_i + \left(\frac{\partial M}{\partial \mathbf{x}_i}\right)^T \boldsymbol{\lambda}_{i+1}, \tag{6}$$

$$\frac{\partial \mathcal{L}}{\partial \mathbf{x}_N} = -\mathbf{H}^T \mathbf{R}_i^{-1}[\mathbf{y}_N - \mathbf{H}\mathbf{x}_N] - \boldsymbol{\lambda}_N, \tag{7}$$

$$\frac{\partial \mathcal{L}}{\partial \mathbf{p}} = \mathbf{B}^{-1}(\mathbf{p}-\mathbf{p}_a) + \sum_{i=1}^{N} \left(\frac{\partial M}{\partial \mathbf{p}}(\mathbf{x}_{i-1},\mathbf{p})\right)^T \boldsymbol{\lambda}_i, \tag{8}$$





$$\frac{\partial \mathcal{L}}{\partial \mathbf{u}_i} = \mathbf{Q}_i^{-1}\mathbf{u}_i + \mathbf{G}^T\boldsymbol{\lambda}_i, \tag{9}$$

where $\mathbf{M}^T = (\frac{\partial M}{\partial \mathbf{x}_i})^T$ is the adjoint of the tangent linear model $\mathbf{M}$. At the minimum, the $\mathcal{L}$ gradients are equal to zero. In this case, Eq. 5 transforms into Eq. 2, whereas Eqs. 6-7 give the adjoint model equations

$$\boldsymbol{\lambda}_N = -\mathbf{H}^T\mathbf{R}_i^{-1}\left[\mathbf{y}_N - \mathbf{H}\mathbf{x}_N\right],$$
$$\boldsymbol{\lambda}_i = \left(\frac{\partial M}{\partial \mathbf{x}_i}\right)^T\boldsymbol{\lambda}_{i+1} - \mathbf{H}^T\mathbf{R}_i^{-1}\left[\mathbf{y}_i - \mathbf{H}\mathbf{x}_i\right]. \tag{10}$$

Values of $\boldsymbol{\lambda}_i$ are derived from the forward and adjoint model integrations and are substituted into Eqs. 8-9. In general, $\frac{\partial \mathcal{L}}{\partial \mathbf{u}_i}$ and $\frac{\partial \mathcal{L}}{\partial \mathbf{p}}$ do not equal zero as the minimum has yet to be reached by iteratively minimizing the Lagrangian function $\mathcal{L}$. In GEOS-Chem this is done using the L-BFGS-B algorithm (Byrd et al., 1995). Finally, the entire optimization algorithm consists of the following steps:

1. Run the forward model (Eq. 2) from time $t_1$ to $t_N$ using the current estimates of $\mathbf{p}$ and $\mathbf{u}_i$.

2. Run the adjoint model and simultaneously accumulate the estimate of $\boldsymbol{\lambda}_i$ based on Eq. 10.

3. Calculate the gradients of $\mathcal{L}$ with respect to $\mathbf{p}$ and $\mathbf{u}_i$ using Eqs. 8-9 and estimates of $\boldsymbol{\lambda}_i$.

4. Update the estimates of $\mathbf{p}$ and $\mathbf{u}_i$ using the L-BFGS-B optimization algorithm based on the descent direction defined by $\frac{\partial \mathcal{L}}{\partial \mathbf{u}_i}$ and $\frac{\partial \mathcal{L}}{\partial \mathbf{p}}$.

5. Repeat steps 1-4 until convergence is reached.

Generally, at some point during the convergence process the inversion will start fitting the noise in GOSAT observations. This can be prevented by stopping the iterative algorithm when the reduced chi-squared value for the fitted model approximately equals unity. In practice, the real uncertainty on GOSAT XCH$_4$ retrievals is unknown due to unaccounted errors in the CO$_2$ fields, for example, so we used a different approach. For each WC inversion that was performed, we monitored the evolution of the optimized model fields and compared them to independent observations (from TCCON, the NOAA in situ network,
and the HIPPO-3 aircraft campaign). The iterative process was terminated when the fit to independent observations did not improve any further or started to get worse, based on the assumption that after this threshold the optimization began to fit noise in GOSAT observations. On average, the level of noise was estimated to correspond to GOSAT XCH$_4$ uncertainty of about 10 ppb.

We utilized the reported uncertainty on the GOSAT XCH$_4$ retrievals (with the median value of approximately 10 ppb)
and inflated it to match the GOSAT scatter against TCCON observations (approximately 13 ppb). It was assumed that the observation errors are uncorrelated, so that $\mathbf{R}$ was assumed to be diagonal. The a priori error covariance matrix $\mathbf{B}$ was also





assumed to be diagonal, with 50% uncertainty on $CH_4$ emissions in each surface grid box. Emissions were not split into separate categories but optimized as monthly totals in each surface grid box. GOSAT provides global coverage with a period of three days. Therefore, we did not attempt to characterize global pattern of model errors on shorter time scales and explored keeping the forcing terms constant over a time interval that varied from a minimum of three days up to one month. Little is

5 known about the a priori structure of the model errors, so in the design of the cost function, a priori estimates of model errors were set to zero.

The WC algorithm optimizes scaling factors (SFs) for both the forcing terms and the model parameters (surface emissions). Emission SFs are ratios of optimized emissions to a priori emissions, while forcing SFs are the ratios of optimized forcing terms to a constant scaling parameter $\tilde{u}$. The WC inverse method becomes sensitive to the choice of scaling parameter when working

with multidimensional problems. This choice does not affect the Lagrangian $\mathcal{L}$ (Eq. 4), however, it does change the relative magnitude of $\mathcal{L}$ gradients with respect to forcing terms $\frac{\partial \mathcal{L}}{\partial \mathbf{u}_i}$ (Eq. 9) and to surface emissions $\frac{\partial \mathcal{L}}{\partial \mathbf{p}}$ (Eq. 8). The state vector of the WC inversion is largely dominated by the number of forcing SFs as opposed to the emission SFs (with a ratio of up to 500:1). Due to the high dimensionality of the problem, the L-BFGS-B optimization algorithm can search only a fraction of parameter space in the direction of the largest gradient descent. Therefore, it becomes sensitive to the relative magnitude of the forcing

gradients $\frac{\partial \mathcal{L}}{\partial \mathbf{u}_i}$ versus the emission gradients $\frac{\partial \mathcal{L}}{\partial \mathbf{p}}$. For large values of $\tilde{u}$ (for example, $\tilde{u} > 50$ ppb), the algorithm descends in the direction of the forcing gradient and the WC inversion is transformed into the so-called "full state assimilation". Meanwhile, small values of $\tilde{u}$ (for example, $\tilde{u} < 0.05$ ppb) force the algorithm to minimize the cost function in the direction of emission gradients ("flux assimilation"). The value of $\tilde{u} = 1.0$ ppb was empirically chosen to perform simultaneous optimization of the emissions and forcing terms ("flux+state assimilation").

Application of the WC 4D-Var method is sensitive to the specification of the covariance matrix $\mathbf{Q}$, which is difficult to characterize (Trémolet, 2007). We adopted a diagonal structure of matrix $\mathbf{Q}$ as our standard option. This implies there was no explicit temporal or spatial correlation assumed between model errors. However, some correlation is implicitly present in the model and emerges from both atmospheric transport patterns and the definition of the constant forcing time window. Still, assigning adequate model error uncertainty is one of the major challenges for using the WC method. Generally, there is no

single recipe for that, as model errors come from a variety of sources, with different characteristics and, moreover, vary on daily to seasonal time scales. Additionally, in practice, there is usually no way to properly validate whether the inversion correctly attributed biases in $CH_4$ fields as being caused by surface emissions, model errors, or observational biases. This later statement is related to the fact that surface emission, observational bias, and some model errors may leave similar signatures in the $CH_4$ fields that would not be easy to distinguish even with perfect observational coverage. The situation may even be worse for $CH_4$

biases if incorrect emissions and model errors mask each other and do not show up in the model comparison with the GOSAT data.

Given these issues, our focus here is not on estimating surface emissions of $CH_4$. Instead, we use the WC 4D-Var method to optimally constrain the $CH_4$ state and explore the nature of the errors in the model $CH_4$ simulation. We performed two types of inversions: "full state assimilation" and "flux+state assimilation". Given that little is known about the distribution of model





errors, in both cases we chose a uniform spatial and temporal structure of model error uncertainty $q$ so that the model error covariance is defined as $\mathbf{Q} = q^2 \mathbf{I}$.

We conducted a series of parameter tuning experiments where the WC 4D-Var analysis was performed using values of $q$ ranging from 0.05 ppb to about 2000 ppb, and optimized CH$_4$ fields were validated against independent observations. The

5 experiments showed that for values of $q$ above 50 ppb, the fit of optimized CH$_4$ fields to independent observations did not change noticeably. However, for values of $q$ below 50 ppb, the fit deteriorated as $q$ became smaller. Therefore, $q$ was set to 50 ppb. It is important to note that the magnitude of estimated forcing terms changes with changing $q$, but the general pattern of positive and negative corrections was not significantly affected by the choice of $q$. In the experiments described in Sect. 2.4, we found that the WC method was still able to significantly improve the model and capture the bias in the CH$_4$ state. Therefore, we

considered a uniform structure for $\mathbf{Q}$ to be a satisfactory assumption for this initial assessment of model errors in the context of the WC 4D-Var analysis.

### 2.4 Configuration of Sensitivity Experiments

To evaluate the performance of the WC 4D-Var scheme, we conducted a series of OSSEs. We began with an analysis of the sensitivity of the GOSAT observational coverage to the CH$_4$ state in the model. Following Liu et al. (2015) and Byrne et al.

(2017), we constructed the sensitivity function

$$J = \sum_{n=1}^{N} (\text{XCH}_4^{model})_n, \tag{11}$$

where $\text{XCH}_4^{model}$ are modelled CH$_4$ dry-air mole fractions sampled at the times and locations of the GOSAT observations and convolved with GOSAT scene-dependent averaging kernels. The sensitivity function is summed over all $N$ observations available over the assimilation period. The sensitivity of the GOSAT observations to the modelled state is obtained by taking

the derivative of $J$ with respect to the state, using the model adjoint

$$\frac{\partial J}{\partial x_{i,j,k,t}} = \sum_{n=1}^{N} \frac{\partial (\text{XCH}_4^{model})_n}{\partial x_{i,j,k,t}}, \tag{12}$$

where $x_{i,j,k,t}$ is the CH$_4$ at longitude $i$, latitude $j$, altitude $k$, and time $t$. The sensitivities are expressed in units of $\text{ppb} \, \text{kg}^{-1}$ and can be understood as the propagation backward in time of the GOSAT averaging kernels weighted by the value proportional to the local surface pressure. The sensitivities can be summed in space or time to give an aggregated view of the sensitivity of the

25 GOSAT observational coverage to the modelled CH$_4$ state.

We conducted four OSSEs in order to evaluate the performance of the WC 4D-Var method in regards to mitigating artificially introduced model errors for February-May 2010. In particular, we investigated model biases from four different sources: surface emissions, vertical transport, chemical loss, and initial conditions. The "true" model state was defined as optimized CH$_4$ global fields obtained from an inversion analysis to constrain estimates of monthly CH$_4$ fluxes using GOSAT XCH$_4$ Proxy retrievals

during the same time period. We also refer to these constrained fluxes as "true" CH$_4$ surface emissions. The CH$_4$ initial conditions are described in Sect. 2.1. This "true" model state was used to produce pseudo GOSAT XCH$_4$ measurements by





sampling it at the corresponding times and locations of the real GOSAT measurements and then convolving them with GOSAT averaging kernels. The perturbed model was defined by introducing bias in the "true" model from one of the four specified sources of model bias. Then the pseudo-observations were used to constrain and mitigate biases in the perturbed model $CH_4$ state. The performance of the pseudo-inversion was evaluated by comparing the recovered $CH_4$ fields to the "true" ones. The

analyses were conducted for the standard period of four months (February-May 2010), but most of the results are presented for the second month of the assimilation period, March 2010. This gives the model errors time to accumulate during February, and provides two months of pseudo-data, in April and May, to constraint the $CH_4$ state in March. No noise was added to pseudo-observations. Given that and the fact that, usually, the state is most optimally constrained in the middle of the assimilation period, we believe that the OSSEs should reveal the best performance of the WC method.

The emission bias was introduced by replacing the "true" $CH_4$ emissions with the original a priori emissions. Convection and chemistry were artificially biased by completely turning them off in the model for the duration of the assimilation period. Finally, a bias in initial conditions was introduced by replacing the "true" initial conditions with the ones obtained by running the forward model without convection and with 70% of the a priori emissions from 1 July 2009 to 1 February 2010, the beginning of the assimilation period. The applied biases for these four OSSEs were intentionally designed to be extreme; for

the real world applications, we expect less extreme model errors.

We configured the WC method to carry out "full state assimilation" (as described in Sect. 2.3) and have the freedom to determine independently the location of the bias. The constant forcing time window was set equal to three days and the forcing terms were optimized throughout the entire atmosphere (the mask **G** equals unity everywhere). This particular configuration may not be optimal to mitigate a specific type of bias in a real assimilation with limited observational coverage. Here, we

intend to investigate the performance of the measurements and the assimilation method when no information is given about the sources and magnitude of model errors. We also conducted SC 4D-Var assimilation experiment for comparisons with the WC approach in the OSSE with biased surface emissions.

### 2.5   Configuration of the Assimilation with Real GOSAT Data

For the assimilation of the real GOSAT $CH_4$ data, we used the same WC 4D-Var configuration as in the OSSEs, but with some

modifications. The $CH_4$ initial conditions are as described in Sect. 2.1. The $CH_4$ state was constrained during the standard four-month period of February-May 2010 using GOSAT observations during the same period. Modifications to the WC 4D-Var configuration included tuning the length of the constant forcing time window $T$ and the horizontal and vertical structure of the forcing mask **G**. We also performed additional tuning of the method in order to explore the nature of the model errors. The quality of the constrained $CH_4$ fields was evaluated against independent observations. Additionally, we compared results

of the WC inversions with results of the SC surface flux assimilation.

The a priori model validation presented in Sect. 3.2.2 pointed to the fact that the stratosphere in GEOS-Chem at $4° \times 5°$ resolution, particularly, at high latitudes, may be positively biased. The OSSE results also suggested that the WC assimilation may benefit from additional constraints on stratospheric forcing terms. Therefore, for the assimilation of the real GOSAT data we imposed a negativity bound in the L-BFGS-B algorithm for the optimization of the forcing terms in the extra-tropical





stratosphere (above about 210 hPa and poleward of 44°) at 4° × 5° resolution. No bound was imposed on forcing terms in the 2° × 2.5° resolution assimilation.

In one set of experiments, we performed "full state assimilation" and changed the length of the time window over which the forcing terms are held constant in the assimilation. The forcing mask **G** comprised the entire atmosphere, and biases in the $CH_4$ state potentially induced by incorrect surface emissions were treated as just another source of model errors included in forcing terms. The length for the forcing window was varied from three to 30 days. Short time windows would be more appropriate if the model were affected by temporally changing biases such as those related to transient mesoscale eddies. However, the observations may not be able to constrain the short time scales. Also, for short temporal correlation length scales, there is a higher risk that the inversion will fit noise or possible biases in observations. In contrast, the use of long time windows introduces additional temporal correlations between forcing terms that may be suitable only for mitigation of stationary systematic biases in the model, such as those related to surface emissions, chemistry or stationary transport errors.

In a different set of experiments, we carried out WC 4D-Var "source+state assimilation' and explored the optimal design of the forcing mask **G**. Here, the forcing window was set equal to three days. First, we explored the vertical structure of **G**. The algorithm was configured to optimize forcing terms in 1) the whole atmosphere, 2) above 750 hPa, 3) above 500 hPa, and 4) above 200 hPa. Then the horizontal structure of **G** was modified as well. Forcing terms were applied globally throughout the stratosphere and in the troposphere only over four separate regions: the three regions defined by the boundaries of the GEOS-Chem nested model domains (North America (NA), Europe (EU), and China with South-East Asia (CH)) and over Equatorial Africa (EQAf). In these experiments we also attempted to identify the origin of the biases affecting the model at the location of the TCCON and NOAA measurement sites.

All the above experiments were conducted at the 4° × 5° model resolution. In one additional experiment, we applied the WC 4D-Var "full state assimilation" to constrain errors in GEOS-Chem at 2° × 2.5° resolution. We used the standard configuration with a forcing time window of three days. The only difference between the 4° × 5° and the 2° × 2.5° assimilation was in the initial conditions, which are described in Sect. 2.1.

## 3 Results

### 3.1 OSSE Experiments

We began with the analysis of the sensitivity of the GOSAT observation coverage to the $CH_4$ state in the model (see Eq. 12). Shown in Fig. 3, as a function of altitude and latitude, is the total zonal adjoint sensitivity (upper panel), which is a sum of adjoint sensitivities over time and longitude. Additionally, we included a vertical slice of the total adjoint sensitivity across 34°N latitude, which is a sum of adjoint sensitivities over time (lower panel). As suggested by the upper panel in Fig. 3, the entire GOSAT observing system in February to May 2010 has the greatest sensitivity to $CH_4$ changes in the upper troposphere and the lower stratosphere (UTLS) in the northern hemisphere. This can be explained by the fact that winds in the UTLS region are stronger than in the lower troposphere, hence any change in the $CH_4$ fields in the former region will eventually affect a larger number of GOSAT measurement locations in the model. The sensitivity in the tropics is approximately half that





in mid-latitudes. The lower panel in Fig. 3 also shows that over the oceans (which are not covered by the GOSAT observations used in this study), such as over the Pacific Ocean, between 120°E and 130°W), the sensitivity is reduced near the surface and is increased in the UTLS where most of $CH_4$ mass flux from Asia enters the North American domain. Increased sensitivity between approximately 30°W and 30°E is due to the large number of GOSAT measurements over the Sahara desert.

In the first OSSE, we tested the ability of both the SC and WC 4D-Var schemes to reproduce mean $CH_4$ atmospheric concentrations for the case of biased emissions. It is expected that the SC 4D-Var method will produce better results than the WC 4D-Var due to the fact that, when using the SC method, we implicitly supply the assimilation with knowledge about the source of the bias. The results of the OSSE are presented in Fig. 4, which shows the mean difference between the recovered state and the "true" $CH_4$ fields. The results confirm that the SC 4D-Var method better removes $CH_4$ biases due to emissions.

As shown in Fig. 5, the major challenge for the WC 4D-Var was to constrain $CH_4$ fields in the boundary layer below about 800 hPa above large emission sources. Here the method failed to properly correct the vertical structure of the model biases. Due to weak vertical sensitivity of the pseudo-data, it is difficult for the WC 4D-Var method to mitigate strong localized vertical bias. Instead, it compensates for the bias by applying relatively weak $CH_4$ state adjustment of the opposite sign in the column of the atmosphere above, particularly in the stratosphere (see Fig. 5).

In order to improve the WC 4D-Var performance, additional information is required about the location (which can be specified using forcing mask **G**) and properties of the model errors. For example, the equivalent of perfect temporal correlation can be accounted for using a constant forcing time window (one month instead of three days for emissions). Improvement in the performance can also be achieved by providing additional information on the structure of the model errors, such as by assigning a forcing error covariance matrix **Q** with non-uniform vertical error structure, exhibiting larger uncertainty in the

boundary layer and smaller errors in the free troposphere. A simple example of the gain in the performance by using a 30-day forcing time window is shown in Fig. 5. In another example, shown in the same figure, we changed the forcing mask **G** so that model errors are estimated only in the troposphere (approximately, from the surface to 200 hPa). However, this is a rather loose constraint on the potential location of model errors and did not result in a significant improvement in $CH_4$ state in the troposphere beyond the previous experiments.

In the second OSSE, we investigated the ability of the WC 4D-Var method to mitigate errors in vertical transport by turning off convection in the model. This resulted in enhanced $CH_4$ concentrations in the lower troposphere and reduced $CH_4$ in the upper troposphere over the main source regions as seen in Fig. 6. Furthermore, the positive $CH_4$ anomalies in the lower troposphere were partly advected downstream. For example, over Equatorial Africa and South America, instead of being convectively lofted over the continent, $CH_4$ emissions were transported westward in the lower and middle troposphere (see

Fig. 6, first column, third row). As shown in the figure, the state corrections capture the general horizontal and vertical structure of the a priori bias. The largest corrections are co-located with the regions of deep convection. Positive corrections are found in the upper troposphere and negative corrections in the lower. Still, this was not sufficient to fully mitigate the extreme bias associated with turning off convection, but the results show that GOSAT retrievals possess sensitivity to biases in vertical transport and can distinguish them even when the sources and magnitude of model errors are unknown.



Figure 7 shows the mean vertical distribution of the a priori and a posteriori residual biases in the $CH_4$ state over equatorial South America, equatorial Africa, equatorial Southeast Asia, and Europe. In mid-latitudes over Europe, the convection bias was much weaker than over the tropics and reached just about 16 ppb near the surface. At altitudes above 600 hPa the WC 4D-Var method was able to strongly mitigate this bias, and below 800 hPa it reduced the bias by more than a factor of two. The worst results in terms of the fractional reduction of the bias were achieved over equatorial Southeast Asia, most likely due to fewer GOSAT retrievals over this region and limited constraints on the $CH_4$ distribution in the outflow region over the ocean. The assimilation also removed a large fraction of the bias in the $CH_4$ fields over Equatorial Africa and South America, particularly in the middle and upper troposphere over Africa and in the lower troposphere over South America.

In the third OSSE, a chemistry bias was created by turning off the reaction of $CH_4$ with OH. This was the least challenging bias for the WC 4D-Var scheme to mitigate. This bias was rather smooth in the troposphere and did not contain small-scale features. Although the actual chemistry bias in the model may have more complex vertical structure, we do not expect chemical biases to be as strongly localized as the biases associated with emissions and vertical transport. The a priori and a posteriori residual biases, as well as WC forcing terms, are shown in the Fig. 8. The WC state optimization performed best over land where the a priori biases were almost completely removed. The optimization was least successful over the oceans in the lower troposphere. This situation is consistent with the distribution of the adjoint sensitivities shown in Fig. 3, which showed lower GOSAT sensitivity to variations in $CH_4$ in the lower troposphere as compared to the upper troposphere, due in part to the absence of GOSAT observations over oceans in our analysis. Shown in Fig. 9 are the mean vertical profiles of the prior and posterior bias over the same four regions considered in Fig. 7. The model does indeed successfully mitigate the bias. Over the convection regions in the tropics, there is some compensatory corrections in the lower troposphere and in the UTLS, which is probably due to the fast vertical transport in these regions and the limited vertical information in the GOSAT retrievals.

In the final OSSE we biased the initial conditions by introducing biases in the vertical distribution of $CH_4$ (by turning off convection) and in the total $CH_4$ mass in the atmosphere (by running the model with different surface emissions in the previous seven months). The initial condition bias is shown on the left panel in Fig. 10. The stratosphere and southern troposphere were positively biased, whereas the northern troposphere was negatively biased. The right panel shows the structure of the a posteriori bias after the WC assimilation, on the last day of the assimilation window, May 31st. It shows that the $CH_4$ state converged to the "true" concentrations everywhere except in the upper stratosphere; the positive upper stratospheric bias was compensated for in the column by a small negative $CH_4$ bias in the troposphere and the lower stratosphere.

In Fig. 11, we show the evolution of the initial condition bias, relative to the total atmospheric $CH_4$ mass, in four altitude bins in each hemisphere: 1000-700 hPa, 700-400 hPa, 400-200 hPa, and 200-0 hPa. What the figure does not show is how much the bias was adjusted in the actual initial conditions. The perfect observing system would completely remove the initial condition bias at the start of the assimilation period (on February 1). However, what is shown on February 1 is just an 8% reduction in the bias in each of the eight regions, relative to the a priori, with the rest of the bias propagated onto the assimilation period. The different regions converged to the "true" $CH_4$ mass at different rates. The tropospheric $CH_4$ burden in both hemispheres (in the 1000-700 hPa, 700-400 hPa, and 400-200 hPa bins) converged mainly during the first month, however, convergence was slower near the surface in the SH. Above 200 hPa, the convergence rate was slow, such that by the third month the $CH_4$





mass had not fully recovered, particularly in the SH where there is reduced sensitivity due to the limited GOSAT observational coverage. The slower convergence above 200 hPa (compared to the troposphere) is expected due to weak vertical transport in the stratosphere. This suggests that additional vertical correlation between forcing terms in the stratosphere would be beneficial to accelerate convergence in the stratosphere.

## 3.2 Assimilation of Real GOSAT Retrievals

The bias between the GOSAT data and the $4° \times 5°$ a priori and a posteriori model is shown in Fig. 12. Here we will refer to the a posteriori results as the **WC_4x5** assimilation, which is our standard WC 4D-Var assimilation at $4° \times 5°$ resolution with a three-day forcing time window and a forcing mask **G** comprising the entire vertical extent of the atmosphere. As can be seen, there are large positive a priori biases at high latitudes in the northern hemisphere and in some low-latitude regions, such as Equatorial Africa and eastern China. The **WC_4x5** assimilation successfully reduces the a priori bias. There is some residual high latitude bias, which resembles noise or bias in the GOSAT observations. In a companion analysis, Stanevich et al. (submitted), in which we examine the impact of model resolution on the modelled $CH_4$ distribution, we showed that the large positive a priori $CH_4$ bias over China may partly be explained by weakening of the vertical transport in the model due to the coarse $4° \times 5°$ resolution. In Stanevich et al. (submitted), we also showed that a significant fraction of the high-latitude bias comes from the stratosphere and is a consequence of running the model at $4° \times 5°$ resolution. As a result, here we repeated the GOSAT WC assimilation at the higher resolution of $2° \times 2.5°$. The results, which are shown in Fig. 13, reveal that the high latitude a priori bias is indeed smaller in the $2° \times 2.5°$ model. At the higher resolution, the WC assimilation also successfully reduces the model bias. For comparison, we repeated the assimilation at $4° \times 5°$, but optimized the emissions instead of the $CH_4$ state. The results for this experiment, referred to as **SC_4x5**, are shown in Fig. 14. As can be seen, the SC assimilation leaves significantly larger residual biases. The pattern of the residual bias indicates that there were other biases that the assimilation could not fit at the expense of the emissions. We will investigate possible sources for these biases in the sections below.

The signal of surface emissions is mixed with possible model errors in the troposphere, such as those related to vertical transport. Biases in the $CH_4$ fields caused by incorrect surface emissions will in some cases have identical structure to those caused by biased vertical transport, which may complicate the interpretation of WC 4D-Var state corrections in the troposphere. On the other hand, it takes much longer for the surface emissions signal to mix into the stratosphere. We therefore assumed that, on the short (four-month) time scale of the simulation, optimized forcing corrections $u_i$ in the stratosphere can be considered independent from the influence of surface emissions. The third column in Figs. 12 and 13 shows the actual mean monthly bias in the a priori $CH_4$ fields that was corrected by the stratospheric forcing terms. The bias corrections in the $2° \times 2.5°$ $CH_4$ simulation are smaller than for the $4° \times 5°$ simulation, which is consistent with Stanevich et al. (submitted), who suggested that part of the stratospheric bias at $4° \times 5°$ resolution is due to the model resolution itself. The WC inversion results suggest that the $4° \times 5°$ model is positively biased in the stratosphere at the high latitudes and weakly negatively biased in the tropics. In contrast, the $2° \times 2.5°$ model is mainly negatively biased in the stratosphere, particularly, around 30-40°N, except for few high latitude regions, possibly related to the polar vortex.



### 3.2.1 Evaluation With TCCON and NOAA Data

Table 1 presents the results of the evaluation of the **SC_4x5** and the **WC_4x5** assimilation with the in situ and TCCON data, whereas Table 2 gives the comparison results at individual TCCON sites. Based on the OSSE results in Sect. 3.1, and provided that the only model bias is due to incorrect surface emissions, we would anticipate the WC assimilation to produce generally

worse fits to the surface measurements than the SC assimilation. The comparisons show that both approaches produced similar improvements in the fit to the NOAA in situ observations, with slightly better performance from the WC method. The WC assimilation had a significant impact on the overall fit to the TCCON XCH$_4$ retrievals, whereas the SC assimilation had a much more limited impact. Table 2 shows the benefits of using the WC method at the individual TCCON sites. With the exception of Park Falls and Lamont, the WC assimilation significantly improved the correlation and reduced the bias between the model

and the TCCON observations. The results suggest that GEOS-Chem a priori CH$_4$ simulation suffered from biases that were not related only to incorrect surface emissions.

The evaluation of the WC tuning experiments is summarized in Fig. 15. The series of WC experiments described in Sect. 2.5 were organized into four groups. The most sensitive indicator of the quality of the model-observations fit is the correlation. The scatter was close to the level of the GOSAT measurement noise and did not change much among the different assimilation

experiments. In the first group of experiments (first panel in Fig. 15), we changed the vertical extent of the forcing mask **G**. We found that restricting the optimized forcing to the stratosphere (altitudes above 200 hPa) resulted in correlation statistics that were only slightly worse than when we optimized the forcing throughout the whole atmosphere. This suggests that a significant part of model errors above all TCCON stations may be related to the representation of the stratosphere in the model. In addition, the bias and scatter plots show that optimization of forcing terms above 200 hPa produced the best fit to NOAA

surface observations. In the second group of experiments (second panel in Fig. 15), we modified the horizontal extent of the forcing mask **G**. We found that optimization of the forcing throughout the stratosphere and only over North America, Europe, China, and Equatorial Africa in the troposphere, as described in Sect. 2.5, produced almost identical fits to the case of the "full state assimilation". These four regions are major sources of CH$_4$ and our results suggest that at the TCCON sites the model was likely affected by errors in emissions and the transport of the emission signal over these regions. Henceforth, we refer to these

assimilation results as **WC_4REG_4x5**. In the third group of experiments (see the third panel in Fig. 15), we varied the length of the forcing window from 3 days to 7 days, 14 days, and 30 days. We found that the agreement at some of the stations, such as Lamont, Park Falls, and Sodankylä, were generally insensitive to increasing the length of the forcing window, which could suggest that the model above these stations was affected by slowly varying biases. The model fit at other stations, particularly, Bialystok, Bremen and Karlshure, degraded when the window length was increased. The three later stations are located close to each other and are, probably, affected by the similar model errors on synoptic times scales of about one week.

to each other and are, probably, affected by the similar model errors on synoptic times scales of about one week.

In the last group of experiments (see the fourth panel in Fig. 15), we compared the performance of the two 4D-Var assimilation modelling approaches (WC "full state assimilation" and SC "flux assimilation") at the two model resolutions, ($4° \times 5°$ and $2° \times 2.5°$). The comparison suggested that, in the absence of a priori bias correction, the SC method brings limited improvements to the a prior CH$_4$ fields at both resolutions. Indeed, we conclude that the SC assimilation at the $4° \times 5°$ resolution





is futile as the a priori model at $2° \times 2.5°$ resolution produces a better fit to the TCCON observations than the SC $4° \times 5°$ assimilation. The performance of the SC assimilation at the $2° \times 2.5°$ resolution was similar to but was surpassed by the "best fit" WC state assimilation at the $4° \times 5°$ resolution in term of its fit to TCCON and NOAA in situ measurements. Overall, the WC state assimilation at $2° \times 2.5°$ resolution generated the best model fit to TCCON observations. However, in all $2° \times 2.5°$

resolution experiments the model bias against NOAA surface measurements was larger compared to the $4° \times 5°$ experiments. For example, the smallest WC a posteriori bias at $4° \times 5°$ was about 10 ppb, whereas at $2° \times 2.5°$ it was about 17 ppb.

Another important conclusion can be drawn from the fact that the WC assimilation at both model resolutions significantly improved the model fit to Izana measurements (see the fourth panel in Fig. 15). The Izana station is located at an altitude of 2370 m above sea level on a small island near the coast of Africa that has no local $CH_4$ emission sources. The model at $2° \times$

$2.5°$ and $4° \times 5°$ resolutions is not able to resolve the inland. Therefore, the model transport in the vicinity of a high-altitude station, particularly, in the lower troposphere, may be subject to similar errors. Hence, the improvement in the assimilated $CH_4$ fields should mainly be related to the corrected transport in the upper troposphere and the stratosphere, which also supports the conclusions drawn from the first group of experiments.

The WC full state assimilation at $4° \times 5°$ leaves a weak positive biases in the GEOS-Chem fields against the TCCON

observations (excluding Sodankylä) in most of the experiments. Mean a posterior inter-station bias at $4° \times 5°$ ($2° \times 2.5°$) resolution is 3.4 (4.0) ppb (excluding Sodankylä), while the scatter is 8.6 (7.3) ppb (including Sodankylä). It is not clear if the GOSAT data is positively biased or if this could be caused by differences between the GOSAT and TCCON averaging kernels in the stratosphere and the fact that, for example, the stratospheric model bias was not fully recovered by the assimilation, particularly, during the first couple of months of the assimilation period (see Fig. 11). The results also do not indicate the

presence of a latitudinal bias between TCCON and GEOS-Chem and, hence, between TCCON and GOSAT.

There is a larger positive $XCH_4$ bias between the model and Sodankylä measurements, 12.6 ppb and 11.2 ppb for the WC assimilation at $4° \times 5°$ and $2° \times 2.5°$ resolution, respectively, however, the correlation is also high, 0.81 and 0.93, respectively. Tukiainen et al. (2016) and Ostler et al. (2014) pointed to the fact that polar vortex conditions at high-latitude stations may induce biases in TCCON $XCH_4$ retrievals. It has been claimed that a priori profiles in the retrievals do not account for and are

25 not adjusted to these dynamic conditions, hence, they significantly deviate from the real $CH_4$ profiles. When there is not enough information in the spectra to correct for such discrepancies, the $XCH_4$ retrievals can be systematically biased. It is possible that both the GOSAT and TCCON could have been affected by the polar vortex conditions during some days in February-April 2010 so that the biases in co-located retrievals are partially cancelled. It should also be noted that the negative a priori correlation between the model and Bialystok $XCH_4$ measurements is partly caused by the limited number (84) of measurements during

the four-month assimilation time window.

### 3.2.2 Evaluation With ACE-FTS and HIPPO-3 Data

Figures 16 and 17 show the results of the GEOS-Chem comparison with the ACE-FTS and HIPPO-3 data. Model versus ACE-FTS data is shown only in the stratosphere in order to exclude potentially biased data due to interference with clouds in the upper troposphere. The mean $XCH_4$ difference between GEOS-Chem and ACE-FTS that is shown was obtained by artificially





extending the ACE-FTS $CH_4$ profiles down into the troposphere using the GEOS-Chem fields and then applying the GOSAT column averaging kernels. Consistent with Saad et al. (2016), the $CH_4$ differences reveal that the a priori $4° \times 5°$ model has a positive stratospheric bias that can be as large as 250 ppb averaged zonally (see Fig. 16). HIPPO-3 comparison also showed that the $4° \times 5°$ model is positively biased in the stratosphere and slightly negative in the troposphere. Wang et al. (2017) showed

that similar positive $CH_4$ biases in mid- and high latitudes exist in TM3, TM5 and LMDz CTMs. The $4° \times 5°$ WC assimilation reduced the positive stratospheric bias with respect to both HIPPO-3 and ACE-FTS, but it did not remove it completely. For example, the maximum model minus ACE-FTS $XCH_4$ bias due to the stratosphere was reduced from about 40 ppb to 30 ppb. The average negative tropospheric $CH_4$ bias relative to HIPPO-3 was reduced. It is possible that the WC method was not able to properly localize the stratospheric bias. However, the validation analysis may also reflect the influence of the slow recovery

of the stratospheric $CH_4$ fields from the bias in the initial conditions. Therefore, discrepancies in the stratospheric $CH_4$ field from the initial conditions in the first two months of the WC assimilation could be contributing to the observed HIPPO-3 and ACE-FTS bias. Unfortunately, the measurements are either too sparse or limited in space and time to verify this assumption.

The positive a priori stratospheric bias relative to ACE-FTS and HIPPO-3 was significantly smaller at $2° \times 2.5°$ than at the $4° \times 5°$ resolution (see Fig. 17), however, it was not completely removed. Stratospheric $CH_4$ fields in the NH above 200 hPa

even became negatively biased at $2° \times 2.5°$, particularly around 30°N-40°N, where the absolute bias became larger than at $4° \times 5°$. The WC assimilation at $2° \times 2.5°$ further corrected the positive biases and significantly reduced the negative bias around 30°N-40°N. As can be inferred from Fig. 13, the latter covered the entire latitudinal band but was particularly pronounced over the Himalayas. Despite the reduction of the stratospheric bias, the $2° \times 2.5°$ WC assimilation introduces a positive $CH_4$ bias relative to HIPPO-3 in the NH lower troposphere.

## 4   Discussion of Model Biases

### 4.1   Stratospheric Bias

The sensitivity experiments carried out in Sect. 2.4 suggested that a stratospheric bias introduced in the system through the initial conditions has the slowest correction rate. However, by the start of the last month of the assimilation, May 2010, the bias is either removed or does not change much with time. Therefore, we focus the discussion here on the stratosphere in the month

of May 2010, with the assumption that the model is free of the influence of the initial conditions. Figure 18 compares the a priori $CH_4$ fields to the optimized fields from the **WC_4x5** and **SC_4x5** assimilations. The top panel shows that corrections in the stratospheric $CH_4$ abundance are the most pronounced feature of the WC optimized $CH_4$ fields, and that changes are smaller in the zonal mean tropospheric fields. The bottom panel is presented to contrast the behaviour of the two 4D-Var approaches. It shows that the SC assimilation attempts to correct the positive high-latitude stratospheric $CH_4$ bias at the expense of surface

emissions. This results in a negative $CH_4$ bias in the lower troposphere, while the surface signal hardly impacts the stratosphere. In the WC assimilation, stratospheric $CH_4$ was significantly reduced at high latitudes and increased in the tropics relative to the a priori, which is consistent with the correction of the biases shown in Fig. 16 and 17. The changes are more substantial in




the northern hemisphere due to the asymmetrically larger number of GOSAT measurements in the northern hemisphere (see the adjoint sensitivity in Fig. 3).

Large biases in the stratosphere were previously identified in GEOS-Chem (Saad et al., 2016) and in other chemistry transport models (Strahan and Polansky, 2006; Patra et al., 2011; Ostler et al., 2016). The problem was mainly linked to biases in

the meridional Brewer-Dobson circulation in the stratosphere and in the rate of troposphere-stratosphere exchange. However, neither mechanism was analysed in detail. Indeed, the observed changes in Fig. 18 may partly reflect discrepancies in the Brewen-Dobson circulation projected from the initial conditions. In particular, too-rapid meridional overturning in the months prior to the assimilation would have transported excess of $CH_4$ from the tropics and to the high latitudes. In the companion study, Stanevich et al. (submitted) show that the stratospheric bias in GEOS-Chem can also be due to increased numerical dif-

fusion at the coarse horizontal model resolution. This leads to additional unphysical horizontal mixing between the troposphere and the stratosphere and between the high latitudes and the tropics in the stratosphere.

## 4.2 Tropospheric Bias

### 4.2.1 Pattern of forcing terms

The forcing terms are corrections applied to the $CH_4$ fields at each model time step. This time step is equal to 30 min and 15

15  min for the $4° \times 5°$ and $2° \times 2.5°$ simulations, respectively. In order to compare the forcing terms in the two simulations, we added together the state corrections at two successive $2° \times 2.5°$ time steps. Therefore, all forcing terms discussed in this section are presented for 30 min time intervals. The first column in Fig. 19 presents forcing terms in the troposphere optimized by the **WC_4x5** assimilation. The observed structure of the forcing terms simultaneously mitigated model errors from multiple sources. In this section, we attempt to give the most likely explanation of the retrieved pattern of the state correction and

identify sources of regional biases.

In general, the original a priori $CH_4$ fields can be affected by model errors that either occurred during the assimilation period or have been projected onto the assimilation window from the initial conditions. Here, we investigate the former case. Given the results of the OSSE with biased initial conditions in Sect. 3.1, we focus in Fig. 19 on the mean forcing terms in the last three months of the assimilation (March-May 2010) as they are much more likely to be related to recent model errors rather

than to biases in the initial conditions. The temporally averaged structure also gives insight into systematic model errors and is easier to interpret. Figure 19 (first column) shows that negative forcing terms dominate near the surface and in the lower troposphere, particularly over Europe, Equatorial Africa and East Asia. The $CH_4$ reduction at the surface is consistent with NOAA observations. Positive state corrections are more frequently found in the upper troposphere, mainly in mid-latitudes over the Pacific and Atlantic oceans as well as over Europe and significant part of Russia. There are also several regions,

such as eastern China and equatorial Africa, where the forcing terms are negative throughout the entire tropospheric column. Vertical slices over mid-latitudes (bottom right panel) show that strong negative corrections over the east coast of Asia and North America are accompanied by positive corrections in the upper troposphere downwind of the continents. Forcing terms are generally weaker in the lower troposphere over the oceans where we lack GOSAT observations.



Generally, corrections of one sign with monotonically decaying magnitude from the surface to the upper troposphere could be associated with biases in the surface emissions, while the dipole structures with corrections of the opposite sign in the upper and lower troposphere could be related to errors in vertical transport. However, it is not feasible to uniquely identify the origin of model errors from the pattern of forcing terms because model errors from separate sources are mixed in the atmosphere and

the estimation of the forcing terms is an under-constrained inverse problem.

Still, we may try to identify possible sources of model errors. For example, initial assessment of the state corrections pointed to potential issues in vertical transport. Indeed, the dipole structure of the forcing terms could indicate that upward transport of $CH_4$ in mid-latitudes may be insufficient, particularly, over regions with strong vertical $CH_4$ gradients that are present over large sources of $CH_4$. In NH mid-latitudes the major $CH_4$ source regions are China, the US, and Europe. Moreover, the eastern

parts of China and North America are located in regions of significant extra-tropical cyclone activity (Stohl, 2001; Shaw et al., 2016), where $CH_4$ emitted from the surface is being lifted into the free troposphere in warm conveyor belts associated with these cyclones (Kowol-Santen et al., 2001; Li et al., 2005; Sinclair et al., 2008; Lin et al., 2010). Moist convection over land could also contribute to the total transport bias, however convective transport is not strong over these mid-latitude regions during the months of February-May.

Similar vertical structure in the forcing terms was identified above and downwind of eastern North America and China (see the first column, forth row of Fig. 19). The WC method applied negative corrections over the land, from the surface to the upper troposphere, and large positive corrections in the upper troposphere and weakly negative correction in the lower troposphere over the oceans downwind of the continents. The WC method may suggest that vertical transports over eastern parts of the continents has to be stronger. In such case, more $CH_4$ emitted from local sources reaches the middle to upper troposphere and

is transported away from the continents by strong westerly winds. Meanwhile, $CH_4$ concentrations in the entire atmospheric column over land and in the lower troposphere over the adjacent oceans are reduced. Therefore, the large positive a priori bias between the model and GOSAT over China shown in Fig. 12 (first column) may partly be attributed to weak local uplift of $CH_4$.

The observed structure of the forcing terms cannot be uniquely attributed only to biases in vertical transport. The WC

method significantly reduced $CH_4$ in the stratosphere at high latitudes. If biases in the stratospheric $CH_4$ fields are induced by transport errors, the total $CH_4$ budget has to be conserved. Therefore, $CH_4$ removal from the high-latitude stratosphere has to be compensated for in either the tropical stratosphere or the upper troposphere. Hence, the positive forcing terms in the upper troposphere, particularly, in the vicinity of the westerly jet, may also be partly related to model errors in the troposphere-stratosphere exchange and may correct for $CH_4$ leaking from the troposphere to the stratosphere. The negative forcing terms

over China and North America may also partly correct for positively biased a priori surface emissions.

Another region of interest, as suggested by the WC assimilation (Fig. 19, first column, third row), is equatorial Africa. Similar to China, a large positive a priori model $XCH_4$ bias was found here. However, due to the observational coverage, there are limited direct constraints on the $CH_4$ outflow from equatorial Africa except for sparse GOSAT observations over South America. While the African $XCH_4$ bias could be related to positively biased local a priori surface emissions, the WC

assimilation also suggested another transport related explanation. The WC assimilation applied negative $CH_4$ forcing terms





over central Africa and positive forcing terms downwind in the middle troposphere (between 400 and 800 hPa) over the Atlantic Ocean. Such a pattern of state correction could point to potential errors in $CH_4$ outflow from the African continent. Southern Africa is characterized by a persistent high pressure system that drives easterly outflow from southern tropical Africa to the Atlantic in the lower to middle troposphere (Garstang et al., 1996). In their analysis of the sources of moisture in the

5 Congo Basin, Dyer et al. (2017) showed that there is a strong export of moisture from southern tropical Africa to the Atlantic between 800-500 hPa. Furthermore, Arellano et al. (2006) found, in their inversion analysis of carbon monoxide (CO) data from the MOPITT instrument, a discrepancy between their a posteriori CO and observations at Ascension Island, which they speculated could be due to errors in the altitude dependence of the outflow from Africa in the GEOS-Chem model. It is possible that too-much $CH_4$ is being convectively lofted to the upper troposphere over central Africa and not enough is exported out

over the Atlantic in the lower troposphere. Figure 4 (first column) displays the bias in $CH_4$ fields when convection was turned off in the model. This caused $CH_4$ emitted over Africa to take a different transport pathway. Instead of being lifted up over the continent, more $CH_4$ was transported out to the Atlantic in the lower to middle troposphere between 500 and 900 hPa. Under such conditions, $CH_4$ is simultaneously depleted over the continent and increased over the Atlantic, which is similar to what the WC forcing terms suggest. We cannot determine the exact origin of the $XCH_4$ bias over Africa, but the forcing terms do

suggest the presence of a transport bias.

The estimation of the forcing terms is an under-constrained inverse problem. Consequently, here we evaluate the impact of reducing the dimensionality of inverse problem by limiting the region of the atmosphere where the forcing terms should be applied. This was done in the **WC_4REG_4x5** assimilation, in which we restricted the forcing optimization to the stratosphere and only over the main $CH_4$ anthropogenic emission regions in the troposphere. The results presented in Sect. 3.2.1 suggested

that the **WC_4x5** and **WC_4REG_4x5** assimilations produced similar fits to the independent observations. Therefore, errors affecting the model, at least, at the location of the validation stations could emerge from either the NA, CH, EU, EQAf, or STRAT regions. The second column in Fig. 19 presents the structure of optimized forcing terms from the **WC_4REG_4x5** assimilation where the number of optimized variables was reduced using the forcing mask **G**. Over China and North America, the forcing terms acquired a better defined dipole structure with positive correction in the upper troposphere and negative

correction in the lower troposphere. Over equatorial Africa, the region of positive corrections in the mid-troposphere moved closer to the continent.

### 4.2.2 Dependence of the forcing terms on model resolution

Coarsening the model resolution from $2° \times 2.5°$ to $4° \times 5°$ can be considered as equivalent to introducing errors in the finer resolution model. Yu et al. (2017) and Stanevich et al. (submitted) showed that at coarse resolution vertical transport in GEOS-

30 Chem is weakened due to loss of eddy mass flux and incorrect regridding of meteorological fields. Stanevich et al. (submitted) also showed that the efficiency of transport barriers is reduced due to increased numerical diffusion, which causes unphysical mixing between the interior and the exterior of the polar vortex, too rapid mixing of $CH_4$ between the tropical and extratropical branch of the Brewer-Dobson circulation, and increased troposphere-stratosphere exchange. Thus, in Fig. 19 we compare the forcing terms from the $4° \times 5°$ assimilation (**WC_4x5**) with those from the $2° \times 2.5°$ WC assimilation (**WC_2x25**).





Differences between the $2° \times 2.5°$ and $4° \times 5°$ forcing represent the response of the WC method to the resolution-induced transport errors. We found that the magnitude of the negative forcing term was reduced in the lower troposphere, particularly, over China. Similarly, the magnitude of positive forcing terms was reduced in the upper troposphere. The pattern of forcing terms on the vertical slice at mid-latitudes became significantly weaker. Comparison of Figs. 12 and 13 also suggested smaller

stratospheric corrections at the $2° \times 2.5°$ resolution. At the same time, the structure and magnitude of forcing terms at the equator (particularly, over equatorial Africa) was not significantly affected by the increase of resolution.

Several conclusions follow from Fig. 19. First, the results suggest that a large fraction of model errors at $4° \times 5°$ resolution, particularly, in the stratosphere and over mid-latitudes in the troposphere are resolution-induced. Second, although the magnitude of the forcing terms at the $2° \times 2.5°$ resolution is smaller, the pattern remains similar, which implies that the $2° \times$

$2.5°$ resolution model may still be affected by the same type of transport errors. Third, the assumptions made about sources of model errors in the tropics, particularly, over equatorial Africa, still apply to the $2° \times 2.5°$ simulation as the structure and magnitude of forcing terms remained unresponsive to the model resolution. It is possible that these regions are dominated by discrepancies in moist convective transport that are large at $2° \times 2.5°$ and $4° \times 5°$. Finally, the results strongly suggest that the WC assimilation and the GOSAT observations have the potential to diagnose transport errors at both model resolutions.

**5   Conclusions**

In this study, we assessed errors in the global GEOS-Chem chemistry transport model during the four-month period of February-May 2010 using the weak constraint 4D-Var data assimilation method at the model resolutions of $4° \times 5°$ and $2° \times 2.5°$. This was done by constraining simulated $CH_4$ fields with GOSAT $XCH_4$ retrievals. This represents the first application of WC 4D-Var scheme for assimilation of GOSAT $XCH_4$ retrievals to characterize model errors in a CTM.

An analysis of the sensitivity of the GOSAT measurements to the atmospheric $CH_4$ state found that the $XCH_4$ retrievals are most sensitive to $CH_4$ mass changes in the stratosphere and in the upper troposphere in the northern hemisphere, which was explained by the GOSAT observational coverage and stronger horizontal winds in the UTLS, allowing the $CH_4$ perturbations to be observed by a larger number of measurements. Sensitivity at the equator was about half that at northern mid-latitudes. In a series of OSSEs, the observations and the WC method were tested to determine the ability of the system to recover "unknown"

errors in $CH_4$ fields associated with artificially introduced biases in emissions, convection, chemistry, and initial conditions. We found that when not supplied with any information about the errors, the WC method was able to significantly mitigate biases in the $CH_4$ fields with slowly changing spatial structures, but was not able to correct strongly localized biases, particularly, those in the boundary layer. Despite having almost flat averaging kernels in the troposphere, our analysis showed that the GOSAT $XCH_4$ retrievals could help constrain the vertical distribution of model errors when convection was turned off in the model.

The WC method needed about a month to recover the bias introduced in the initial condition in the troposphere and about two months to do so in the stratosphere. Generally, the method was successful in mitigating model errors of "unknown" origin and magnitude. However, more optimal performance could be achieved by supplying the method with additional information





about model errors, such as their temporal and spatial correlation using the model errors covariance matrix **Q**. However, characterizing these correlations will be challenging.

The WC method was tuned in a set of experiments to diagnose real model errors in the GEOS-Chem CTM at the $4° \times 5°$ resolution. The a posteriori model fit to independent observations, such as ACE-FTS, HIPPO-3, TCCON and NOAA surface

measurements, was used to evaluate the assimilation. Initial comparisons suggested that GEOS-Chem was affected by biases not solely related to discrepancies in surface emissions. Results suggested that the modelled $CH_4$ fields at the location of most NH TCCON stations were affected by slowly varying biases, however, a few stations, such as Bialystok, Bremen and Karlsruhe, were more likely influenced by errors varying on time scales of one week. The evaluations pointed to a large positive bias in the stratosphere relative to ACE-FTS and HIPPO-3 measurements, and weakly negative bias in the middle to upper troposphere

relative to HIPPO-3 data. The WC assimilation was able to mitigate the negative tropospheric bias and partly removed the stratosphere bias. We found that the SC 4D-Var assimilation that optimized the surface emissions had only limited impact on the model fits. Furthermore, the WC assimilation at $4° \times 5°$ resolution performed better than the SC assimilation at $2° \times 2.5°$ resolution. Meanwhile, the results showed that running the a priori model at $2° \times 2.5°$ resolution produced better agreement with TCCON observations than the a posteriori fields from the SC 4D-Var surface emission optimization at $4° \times 5°$.

State corrections at the $4° \times 5°$ resolution also explicitly pointed to issues with vertical transport, suggesting that vertical transport of $CH_4$ in mid-latitudes over the large $CH_4$ source regions of eastern China and North America is too weak. In the tropics, the WC inversion corrected for large positive $XCH_4$ biases over equatorial Africa. From the pattern of forcing terms, it remained unclear whether the bias over Africa was related to surface emissions. However, the WC method suggested the possibility of biased $CH_4$ outflow from the African continent to the Atlantic Ocean in the mid-troposphere, which could be

related to a discrepancy in the partitioning between deep convection transport to the upper troposphere and shallow outflow to the Atlantic Ocean.

In a companion analysis, Stanevich et al. (submitted) examined the impact of model resolution on the $CH_4$ simulation and found larger model biases at $4° \times 5°$ compared to $2° \times 2.5°$. We found that assimilating the GOSAT data at the higher resolution of $2° \times 2.5°$ produced state corrections that were similar to those obtained at $4° \times 5°$, however, the magnitude of

25 these corrections in the stratosphere and in the mid-latitude troposphere was significantly reduced at the higher resolution. This suggested that the model at both resolutions was affected by transport errors of similar origin, although less so at the $2° \times 2.5°$ resolution, and a significant fraction of these errors was induced by the model resolution itself. The WC assimilation also corrected for the negative $CH_4$ bias relative the ACE-FTS and HIPPO in the northern mid-latitude stratosphere, found only at the $2° \times 2.5°$ resolution, and located this bias particularly over the Himalayas. However, the origin of this bias remained

unclear.

In our analysis, we used only GOSAT $CH_4$ data over land. However, $XCH_4$ glint measurements over oceans could help better constrain the vertical structure of the model errors. The WC 4D-Var assimilation of shorter-lived species, such as CO, could also help better diagnose model errors, especially when transport and emission errors mask each other in $CH_4$ fields, although shorter-lived species may also be more strongly affected by errors in chemistry. The advantage of $CH_4$ is its longer

memory of model transport, however shorter-lived gases are more strongly affected by and, hence, may be more sensitive to the





same model errors. Clearly, the detected transport error at the $4° \times 5°$ resolution would have considerable impact on inferred emissions if the model were assumed to be perfect, as is the case in SC 4D-Var. Instead of reducing positive high-latitude bias in the stratosphere, the $4° \times 5°$ SC 4D-Var surface flux assimilation negatively biased the lower troposphere. The SC inversion also significantly reduced Chinese $CH_4$ emissions by incorrectly attributing model errors in vertical transport. Some of the
detected transport error were significantly smaller at the $2° \times 2.5°$ resolution, while others remained resolution-independent. The effect of these remaining errors at the $2° \times 2.5°$ resolution has to be further investigated.

Potentially, any CTM may be improved if the signal from the surface emissions can be separated from other model errors. This would be a rather challenging task for GOSAT $XCH_4$ measurements. Further analysis is needed on this problem, particularly on the design of the model error covariance matrix $\mathbf{Q}$. For example, Trémolet (2007) proposed a design based on statistics
of model tendencies. The $\mathbf{Q}$ matrix had a rather primitive structure in our analysis, although sufficient for the objectives of this work. Based on our initial assessment of model errors, the structure of $\mathbf{Q}$ can be further improved. In the meantime, the WC 4D-Var method has a number of immediate useful applications. In general, it is a valuable instrument for diagnosing model errors. It can also be used as a tool to produce a better estimate of the $CH_4$ state in the model in order to provide boundary and initial conditions for forecasting purposes or regional-scale analysis at higher spatial resolution.

**6   Data and code availability**

The GOSAT satellite data are available at http://www.esa-ghg-cci.org/sites/default/files/documents/public/documents/GHG-CCI_ DATA.html. The TCCON data are available at http://tccondata.org/. The NOAA-ESRL Global Greenhouse Gas Reference Network data are available at https://www.esrl.noaa.gov/gmd/dv/data/. The HIPPO aircraft data are available at http://hippo.ornl. gov/data_access/. The ACE-FTS data are available at https://databace.scisat.ca/level2/ace_v3.5_v3.6/, and registration is re-
quired to download the data. The code for the GEOS-Chem model and its adjoint (with the weak constraint capability) is publicly available and can be downloaded from www.geos-chem.org. The output from the GEOS-Chem model simulations used in this analysis are available upon request.

*Acknowledgements.* This work was supported by funding from Environment and Climate Change Canada and the Natural Science and Engineering Research Council (NSERC) of Canada. We thank NOAA-ESRL for making their $CH_4$ surface measurements publicly available.
We thank S. C. Wofsy for providing HIPPO aircraft data and R. J. Parker for providing GOSAT $XCH_4$ data. R. J. Parker was funded via an ESA Living Planet Fellowship. R. J. Parker and H. Boesch acknowledge funding from the UK National Centre for Earth Observation (NCEO), the ESA Greenhouse Gas Climate Change Initiative (GHG-CCI) and the EU Copernicus Climate Change Service (C3S). We thank the Japanese Aerospace Exploration Agency, National Institute for Environmental Studies, and the Ministry of Environment for the GOSAT data and their continuous support as part of the Joint Research Agreement. This research used the ALICE High Performance Computing
Facility at the University of Leicester for the GOSAT retrievals. Funding for Wollongong TCCON is provided in part by the Australian Research Council (ARC) grants DP160101598, DP140101552, DP110103118 and LE0668470. The Atmospheric Chemistry Experiment (ACE), also known as SCISAT, is a Canadian-led mission mainly supported by the Canadian Space Agency and NSERC.





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





**Table 1.** Evaluation of a priori, **SC_4x5** and **WC_4x5** optimized $CH_4$ fields using TCCON $XCH_4$ from the stations listed in Table 2 and NOAA surface *in situ* observation (mean statistics for the period of February-May 2010). The first, second and third columns represent mean difference, standard deviation and correlation between the model and measurements, respectively. The fourth column represents the slope of the regression line with modelled data on the y-axis and measurements on the x-axis.

|  | Bias [ppb] | | | Scatter [ppb] | | | Correlation ($R$) | | | Slope of regression | | |
|---|---|---|---|---|---|---|---|---|---|---|---|---|
|  | Prior | SC | WC | Prior | SC | WC | Prior | SC | WC | Prior | SC | WC |
| TCCON | 9.1 | 8.2 | 5.3 | 15.0 | 13.8 | 9.9 | 0.83 | 0.86 | 0.93 | 1.16 | 1.14 | 1.07 |
| In situ | 15.1 | 9.7 | 9.6 | 34.5 | 30.3 | 28.9 | 0.88 | 0.89 | 0.90 | 0.90 | 1.02 | 0.99 |





**Table 2.** Evaluation of a priori, **SC_4x5** and **WC_4x5** optimized $CH_4$ fields using TCCON $XCH_4$ (mean station-wise statistics for the period of February-May 2010). The first, second and third columns represent mean difference, standard deviation and correlation between the model and measurements, respectively.

| | Bias [ppb] | | | Scatter [ppb] | | | Correlation ($R$) | | |
|---|---|---|---|---|---|---|---|---|---|
| | Prior | SC | WC | Prior | SC | WC | Prior | SC | WC |
| Sodankylä (67.37°N, 26.63°E) | 30.0 | 25.7 | 13.7 | 18.9 | 19.1 | 12.6 | 0.49 | 0.50 | 0.81 |
| Bialystok (53.23°N, 23.03°E) | 11.9 | 7.3 | 5.1 | 9.3 | 10.6 | 8.0 | 0.39 | 0.43 | 0.65 |
| Bremen (53.10°N, 8.85°E) | 6.3 | 3.2 | 0.7 | 14.3 | 15.2 | 10.5 | -0.37 | -0.28 | 0.47 |
| Karlsruhe (49.10°N, 8.44°E) | 6.4 | 4.4 | 0.8 | 9.7 | 9.9 | 8.9 | 0.33 | 0.29 | 0.49 |
| Orleans (47.97°N, 2.11°E) | 3.9 | 3.5 | 2.5 | 8.9 | 9.6 | 8.3 | 0.31 | 0.30 | 0.51 |
| Garmish (47.48°N, 11.06°E) | 9.9 | 10.0 | 5.7 | 9.0 | 9.7 | 8.4 | 0.46 | 0.56 | 0.65 |
| Park Falls (45.95°N, 90.27°W) | 1.9 | 3.6 | 2.3 | 9.7 | 10.6 | 8.5 | 0.37 | 0.47 | 0.65 |
| Lamont (36.60°N, 97.486°W) | 1.4 | 3.7 | 4.4 | 11.1 | 11.5 | 9.4 | 0.27 | 0.30 | 0.49 |
| Izana (28.30°N, 16.5°W) | -5.8 | -5.3 | 3.1 | 7.6 | 8.2 | 6.7 | 0.64 | 0.58 | 0.72 |
| Wollongong (34.41°S, 150.88°E) | 7.5 | 3.9 | 3.7 | 8.9 | 8.8 | 8.4 | 0.58 | 0.55 | 0.59 |
| Lauder (45.04°S, 169.68°E) | 9.6 | 9.2 | 5.9 | 5.6 | 5.7 | 5.4 | 0.72 | 0.72 | 0.73 |





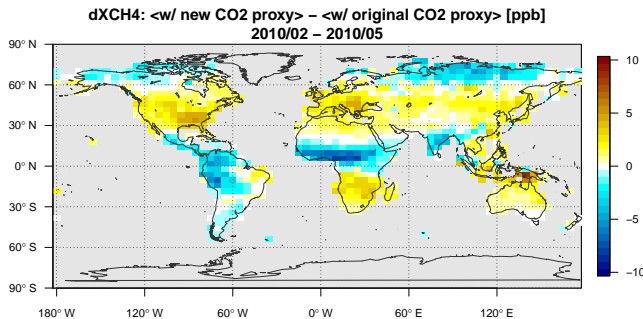

**Figure 1.** Mean difference from 1 February - 31 May 2010 between GOSAT $XCH_4$ retrievals based on the original and new $CO_2$ proxy fields. The original $CO_2$ proxy is based on the median of the $CO_2$ distributions from the GEOS-Chem (from the University of Edinburgh), LMDZ/MACC-II and NOAA CarbonTracker models constrained by in-situ surface $CO_2$ observations. The new $CO_2$ proxy is from a $CO_2$ surface flux inversion analysis using the GEOS-Chem model constrained by GOSAT ACOS $CO_2$ retrievals over land.



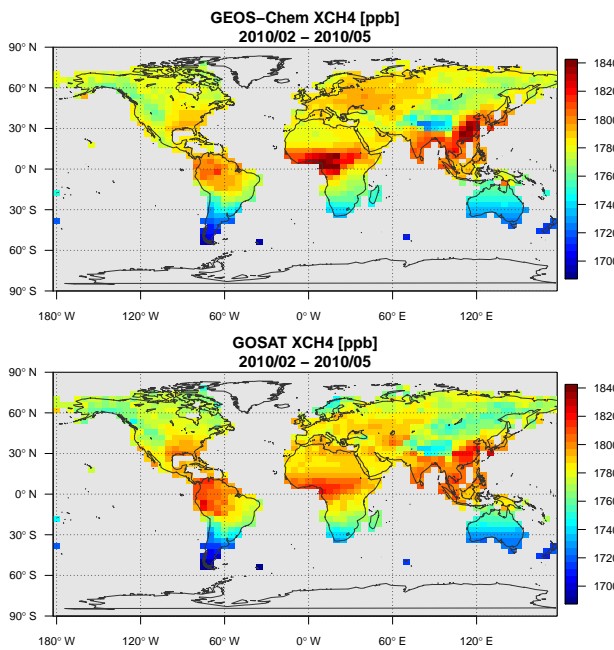

**Figure 2.** Mean XCH$_4$ fields from February to May 2010. Top panel: XCH$_4$ from the GEOS-Chem model at a resolution of $4° × 5°$. The model was sampled at the locations and times of the GOSAT observations and smoothed with the GOSAT averaging kernels. Bottom panel: GOSAT XCH$_4$ retrievals based on the new CO$_2$ proxy fields.

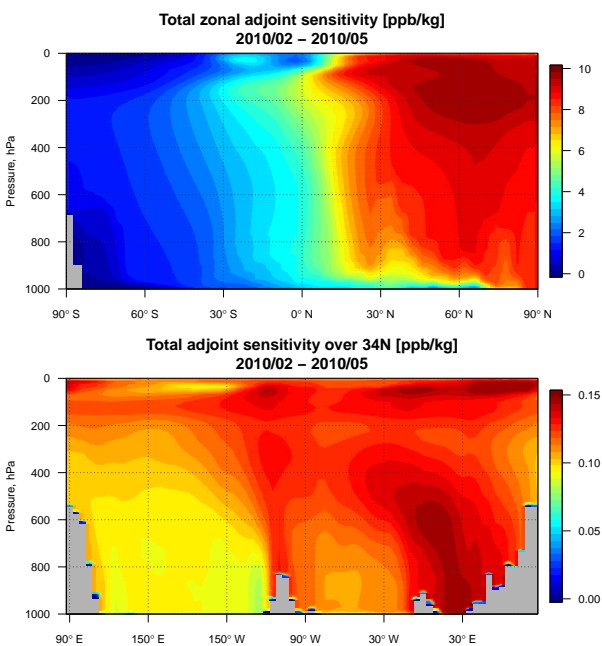

**Figure 3.** Distribution of total adjoint sensitivity in GEOS-Chem of GOSAT observations in February-May 2010 to the modeled $CH_4$ distribution (the model state) during the same period. The adjoint sensitivities have been summed over the time period. Shown are (top) the total zonal mean adjoint sensitivities and (bottom) the altitude-longitude cross section of the sensitivities along 34°N.

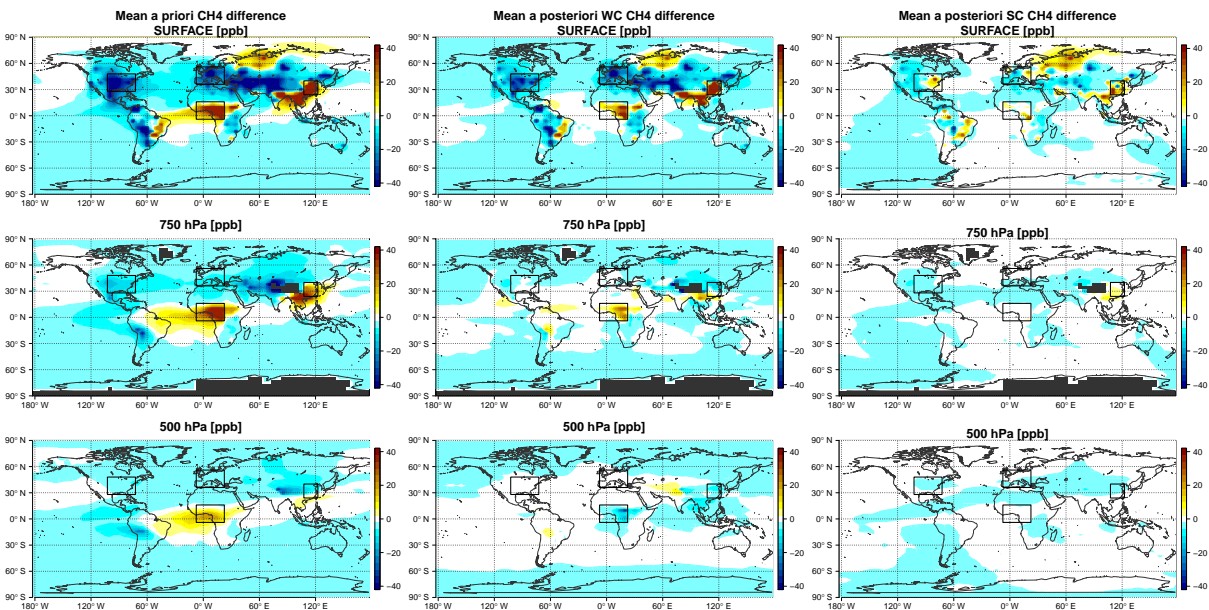

**Figure 4.** Mean differences in the CH$_4$ distribution in March 2010 in the OSSE with biased surface emissions. Left column: mean differences between the priori simulation and that based on the "true" emissions (the true state). Middle column: mean differences between the WC optimized CH$_4$ state and the true state. Right column: mean differences between the SC optimized CH$_4$ state and the true state. The rows show the results at the surface (upper), 750 hPa (middle), and 500 hPa (lower). The black boxes indicate the four domains considered for the regional analysis discussed in the text and shown in Fig. 5.

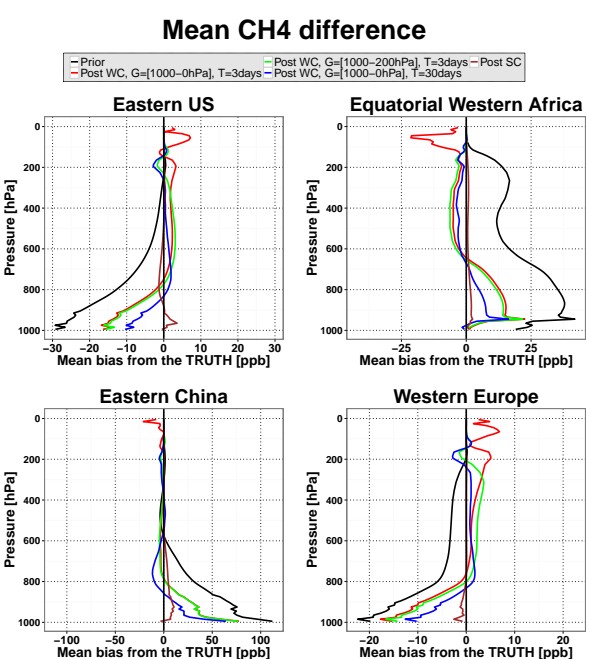

**Figure 5.** Mean vertical profiles of the CH$_4$ differences in March 2010 in the OSSE with biased surface emissions for the four regions depicted in Fig. 4. The differences are between (black lines) the biased a priori and the "true" CH$_4$ state, (dark red lines) the SC optimized state and the "true" CH$_4$ state, and between the WC optimized state and the 'true' CH$_4$ state for the following cases: (light red line) the standard WC optimization with the forcing calculated over the depth of the entire atmosphere (i.e., $G = [1000-0 \text{ hPa}]$) and with a constant forcing window ($T$) of 3 days; (green lines) the forcing calculated from the surface to 200 hPa with a constant forcing window of 3 days; and (blue lines) the forcing calculated over the depth of the entire atmosphere, but with a forcing window of 30 days.

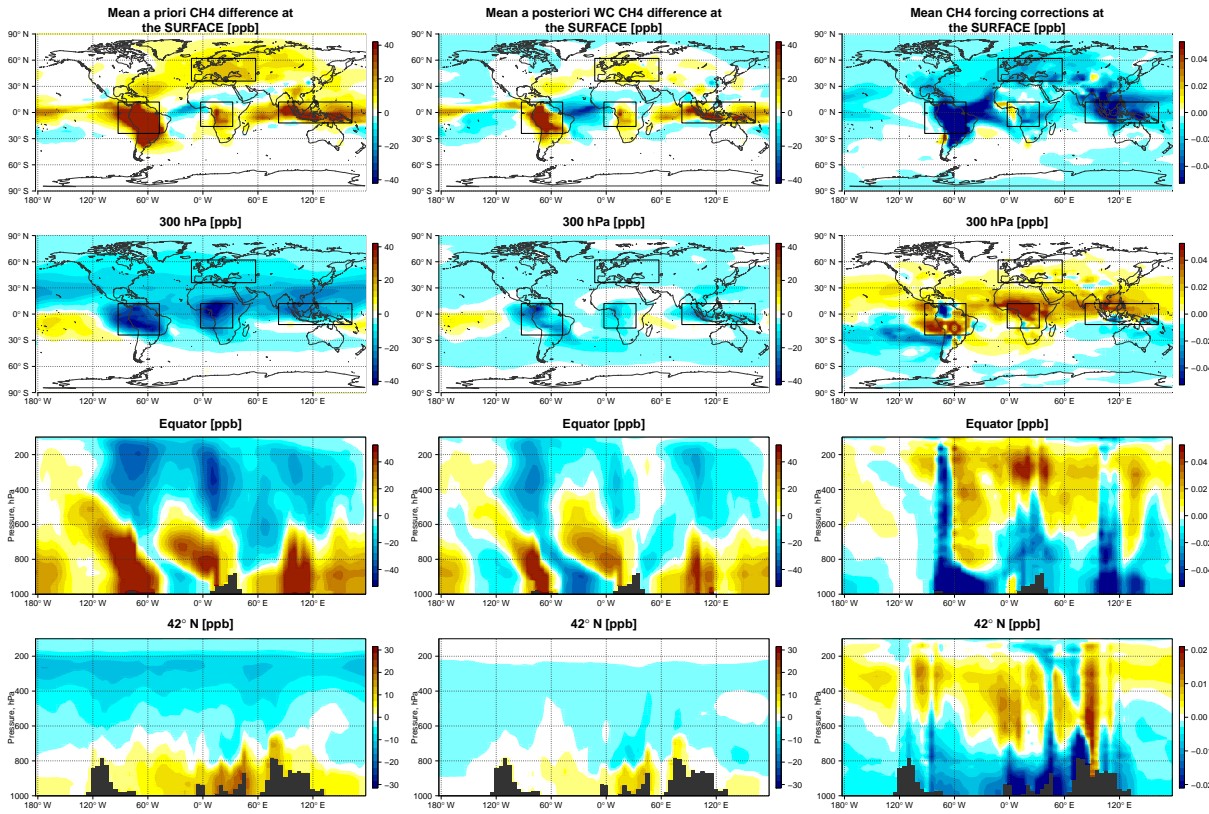

**Figure 6.** Mean differences in the CH$_4$ distribution in March 2010 in the OSSE with biased convection. Left column: mean differences between the a priori CH$_4$ state and the "true" CH$_4$ state. Middle column: mean differences between the WC optimized CH$_4$ state and the "true" CH$_4$ state. Right column: the mean WC state corrections (the forcing terms), in units of ppb. Shown are the latitude-longitude differences at (top row) the surface and (second row) at 300 hPa, as well as the altitude-longitude differences (third row) along the equator and (bottom row) along 42°N. The black boxes indicate the four domains considered for the regional analysis discussed in the text and shown in Fig. 7.

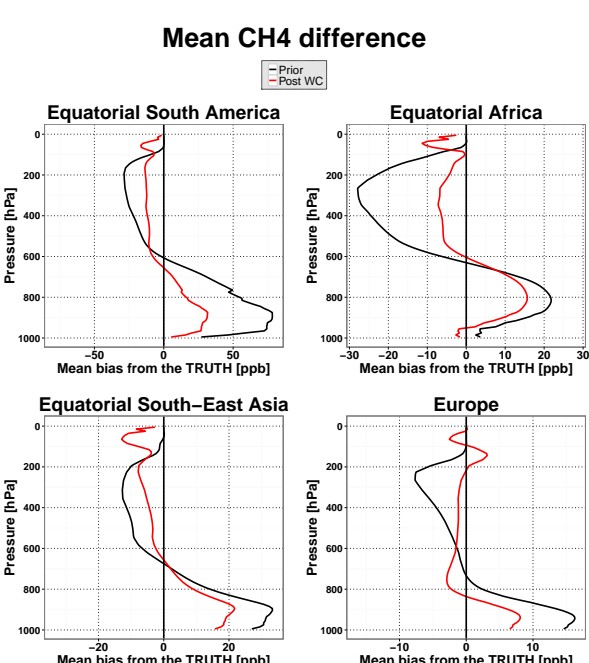

**Figure 7.** Mean vertical profiles of the CH$_4$ differences in March 2010 in the OSSE with biased convection for the four regions depicted in Fig. 6. The differences are between (black lines) the a priori and the "true" CH$_4$ state and between (red lines) the WC optimized state and the "true" CH$_4$ state.

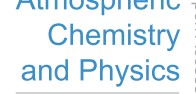
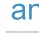

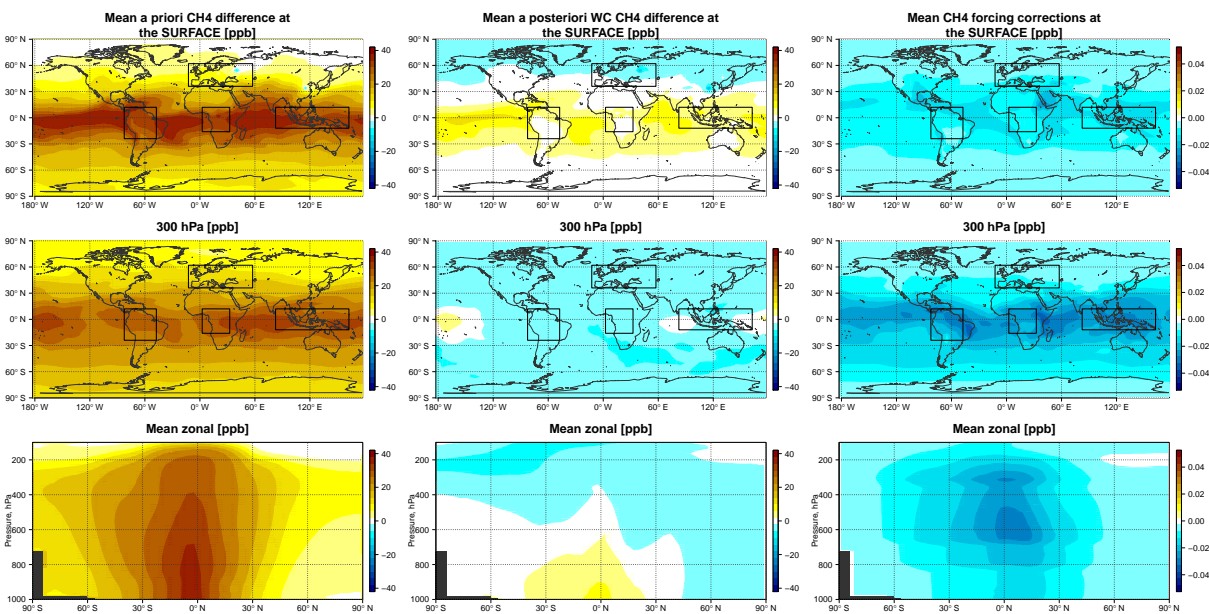

**Figure 8.** Mean differences in the $CH_4$ distribution in March 2010 in the OSSE with biased chemistry. Left column: mean differences between the a priori $CH_4$ state and the "true" $CH_4$ state. Middle column: mean differences between the WC optimized $CH_4$ state and the "true" $CH_4$ state. Right column: the mean WC state corrections (the forcing terms), in units of ppb. Shown are the latitude-longitude differences at (top row) the surface and (second row) at 300 hPa, as well as the altitude-longitude differences (third row) along the equator. The black boxes indicate the four domains considered for the regional analysis discussed in the text and shown in Fig. 9.



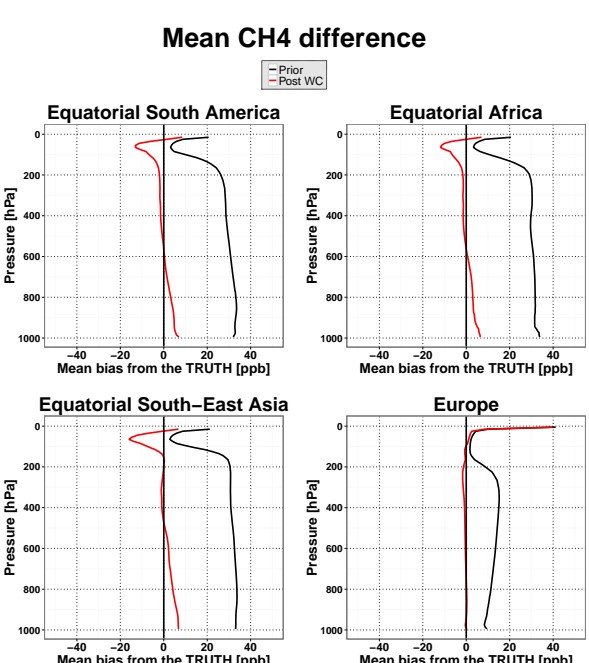

**Figure 9.** Mean vertical profiles of the CH$_4$ differences in March 2010 in the OSSE with biased chemistry for the four regions depicted in Fig. 8. The differences are between (black lines) the a priori and the "true" CH$_4$ state and between (red lines) the WC optimized state and the "true" CH$_4$ state.





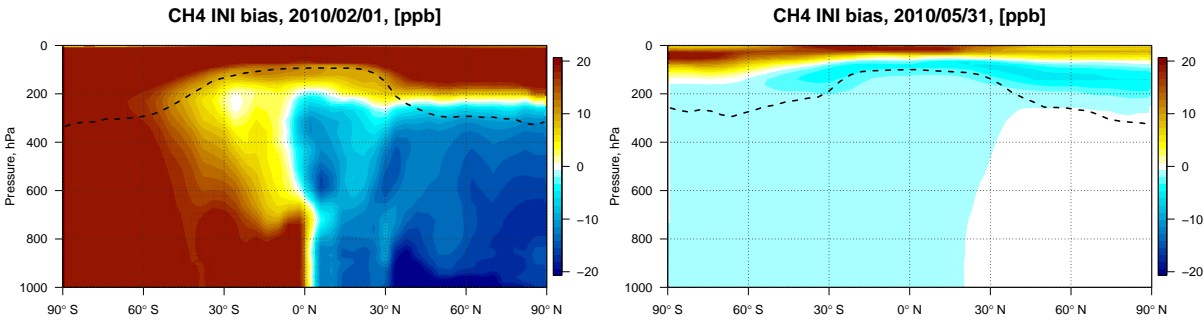

**Figure 10.** Results of the OSSE with biased initial conditions. Left: a priori bias in initial conditions. Right: a posteriori bias at the end of the assimilation window. The dashed line represents the mean tropopause height on 31 May 2010 taken from GEOS-5 meteorological fields.





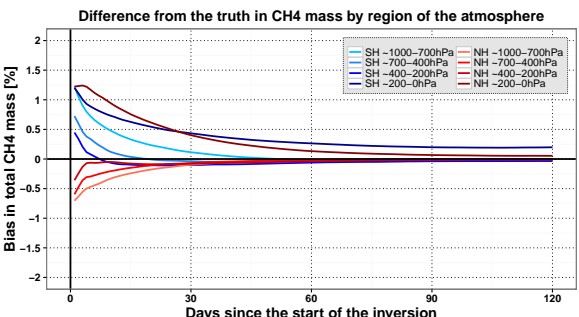

**Figure 11.** Differences in the atmospheric $CH_4$ mass between the WC optimized state and the "true" $CH_4$ state, as a function of time, for the OSSE with biased initial conditions. The $CH_4$ mass was calculated over the eight different regions of the atmosphere as given in the legend. "SH" and "NH" indicate the southern hemisphere and northern hemisphere, respectively.



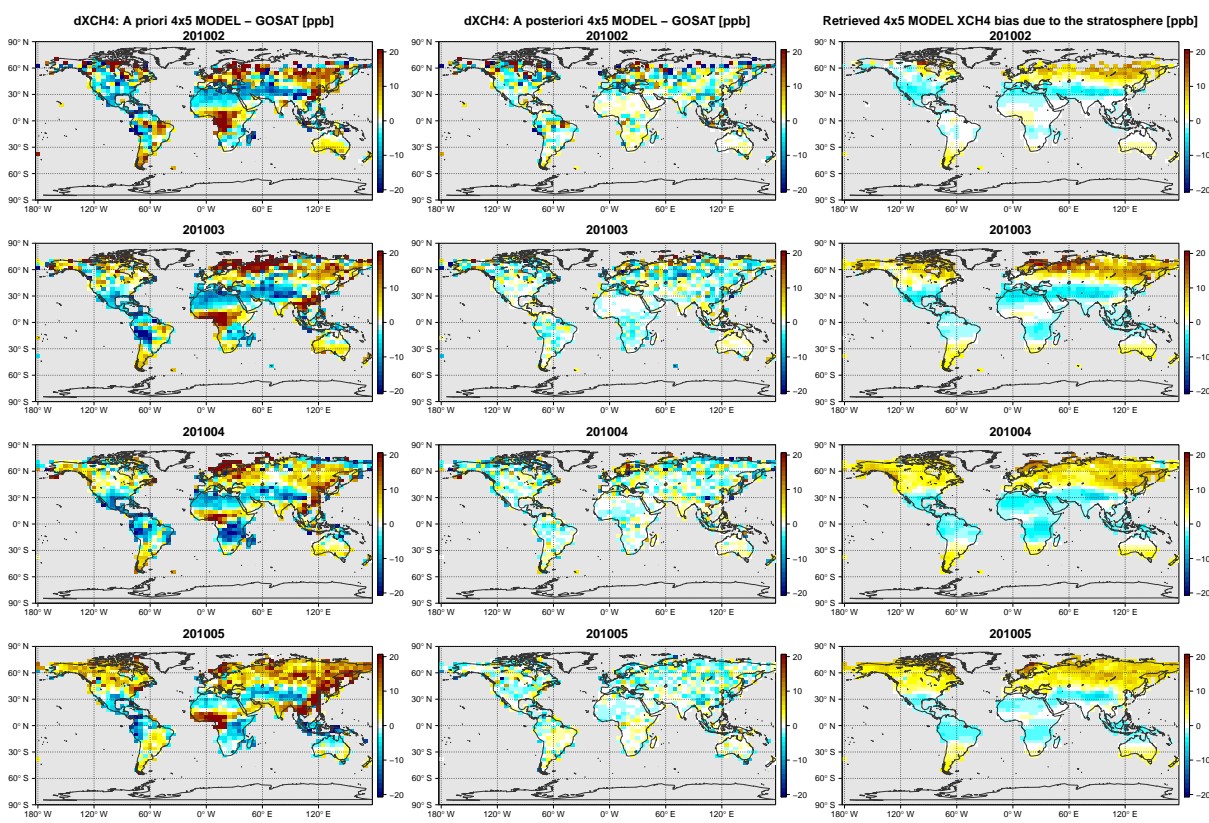

**Figure 12.** Monthly mean fields from the $4° \times 5°$ resolution model for February - May 2010. First column: differences between the GEOS-Chem a priori $XCH_4$ state and the GOSAT data. Middle column: differences between the a posteriori **WC_4x5** $XCH_4$ state and the GOSAT data. Right column: the optimized stratospheric $XCH_4$ bias correction, calculated as the difference between the model simulation with optimized forcing corrections everywhere and the model simulation with the forcing corrections estimated only in the troposphere. The rows represent results for (top row) February, (second row) March, (third row) April, and (bottom row) May 2010. All model simulations were sampled at the locations and times of the GOSAT observations and smoothed with the GOSAT averaging kernels.



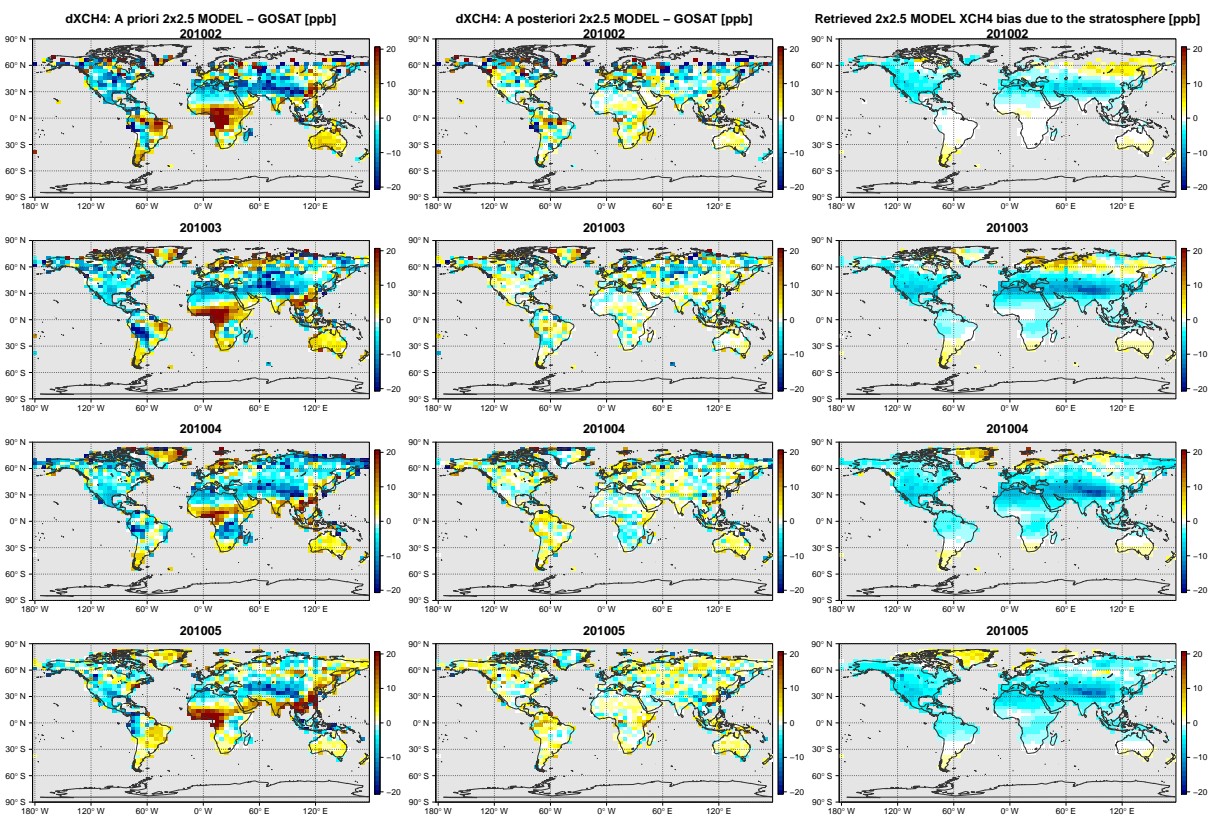

**Figure 13.** Same as Fig. 12 but for the $2° \times 2.5°$ resolution model.

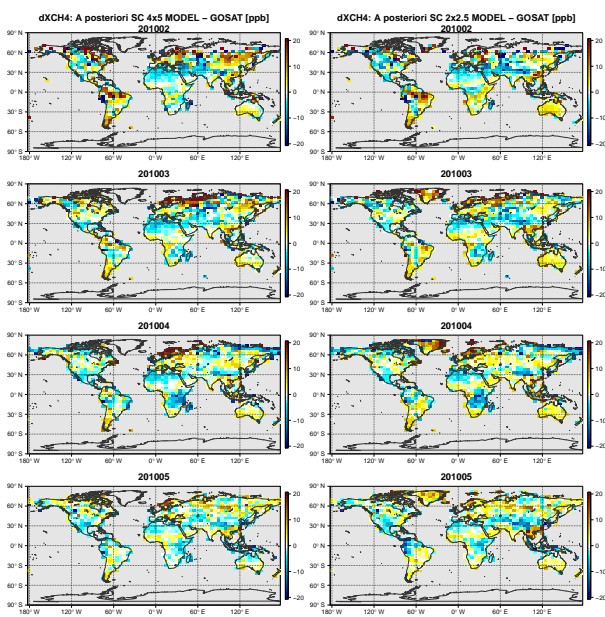

**Figure 14.** Monthly mean differences between the GEOS-Chem a posteriori XCH$_4$ state and GOSAT: Left column: differences between the a posteriori state from the SC "flux assimilation" at $4° \times 5°$ (**SC_4x5**) and GOSAT. Right column: differences between the a posteriori state from the SC "flux assimilation" at $2° \times 2.5°$ (**SC_2x25**) and GOSAT. The rows represent results for (top row) February, (second row) March, (third row) April, and (bottom row) May 2010.



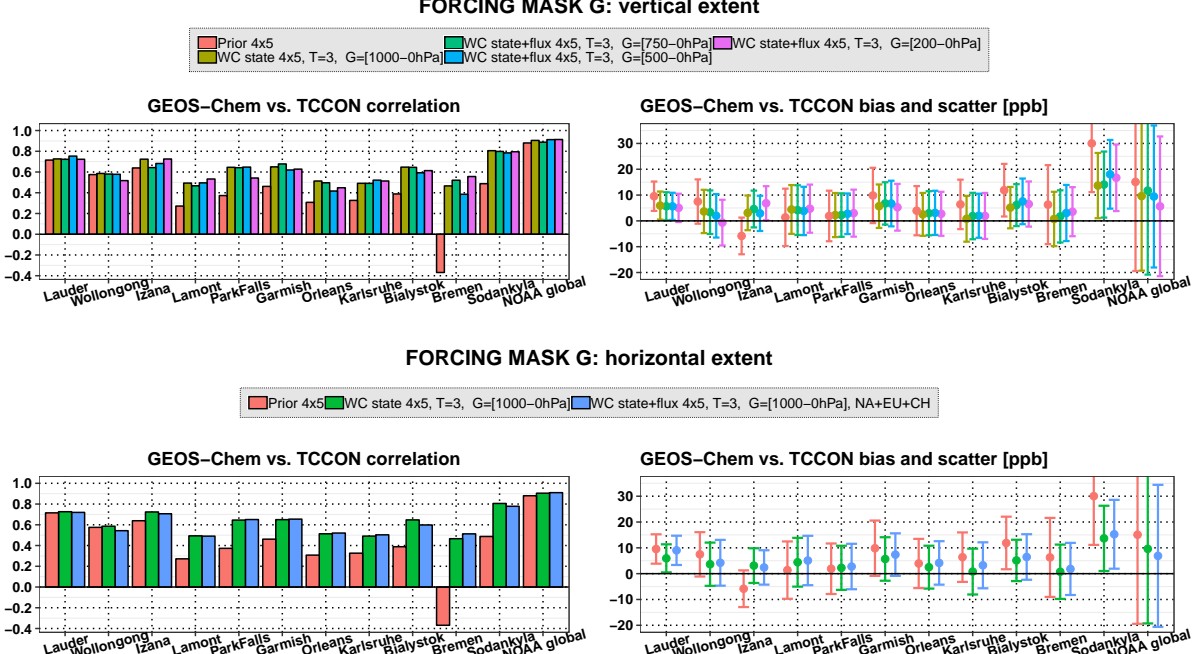

**Figure 15.** Evaluation of the mean (February-May 2010) a priori and optimized $CH_4$ fields using TCCON $XCH_4$ and NOAA surface in situ observation. Results are shown for a set of inversions organized in separate characteristic groups for better representation. The left column shows the correlation with respect to the TCCON and NOAA data, whereas the right column shows the mean bias and scatter. Top row: comparison of (red) the a priori fields, (light green) the standard WC assimilation (with the forcing terms estimated throughout the atmosphere and with $T = 3$ days), (dark green) the WC assimilation with the forcing terms estimated at altitudes above 750 hPa, (blue) the WC assimilation with the forcing terms estimated above 500 hPa, and (purple) the WC assimilation with the forcing terms estimated above 200 hPa. Second row: comparison of (red) the a priori fields, (dark green) the standard WC assimilation (with the forcing terms estimated throughout the atmosphere, (blue) the WC assimilation with joint estimation of the state and surface emissions, and (blue) the WC assimilation with the forcing terms estimated only over North America, Europe, and Asia. Third row: comparison of (red) the a priori fields, (light green) the standard WC assimilation, (dark green) the WC assimilation with a constant forcing window of 7 days, (blue) the WC assimilation with a constant forcing window of 14 days, and (purple) the WC assimilation with a constant forcing window of 30 days. Bottom row: comparison of (red) the a priori fields, (light green) the standard SC assimilation, (dark green) the standard WC assimilation, (light blue) the a priori fields at $2° \times 2.5°$, (dark blue) the SC assimilation at $2° \times 2.5°$, and (purple) the WC assimilation at $2° \times 2.5°$. Each plot shows the statistics for the comparison with each of the 11 TCCON sites considered as well as the global mean statistics for the comparison with the NOAA surface data.



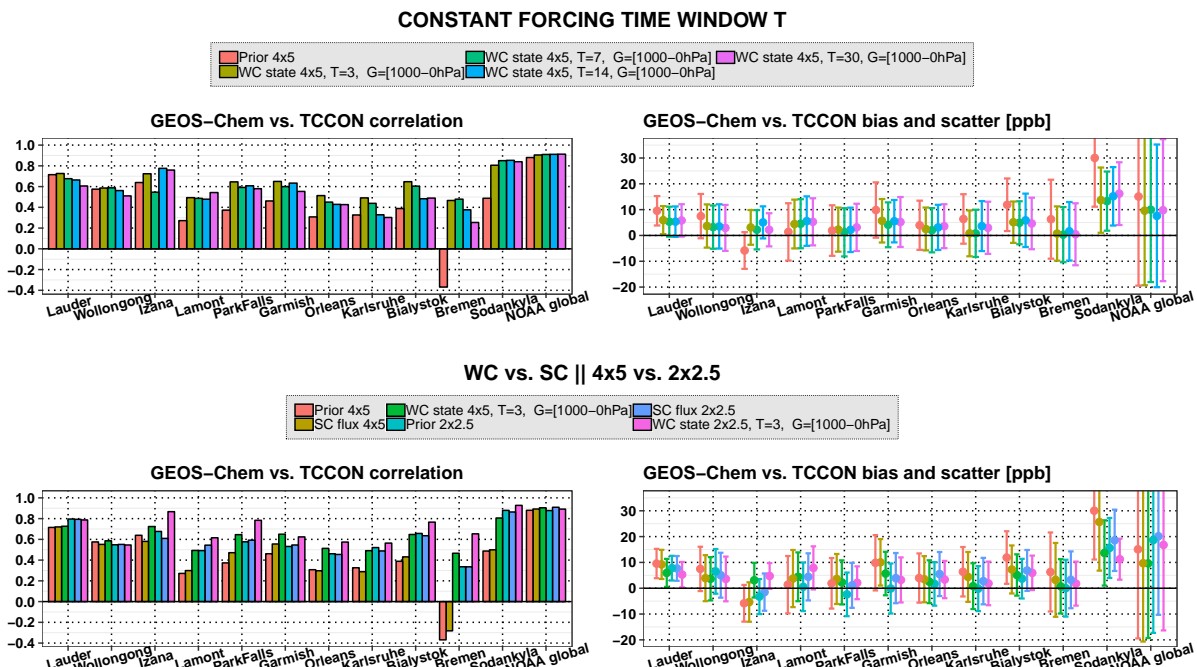

**Figure 15.** Continued.

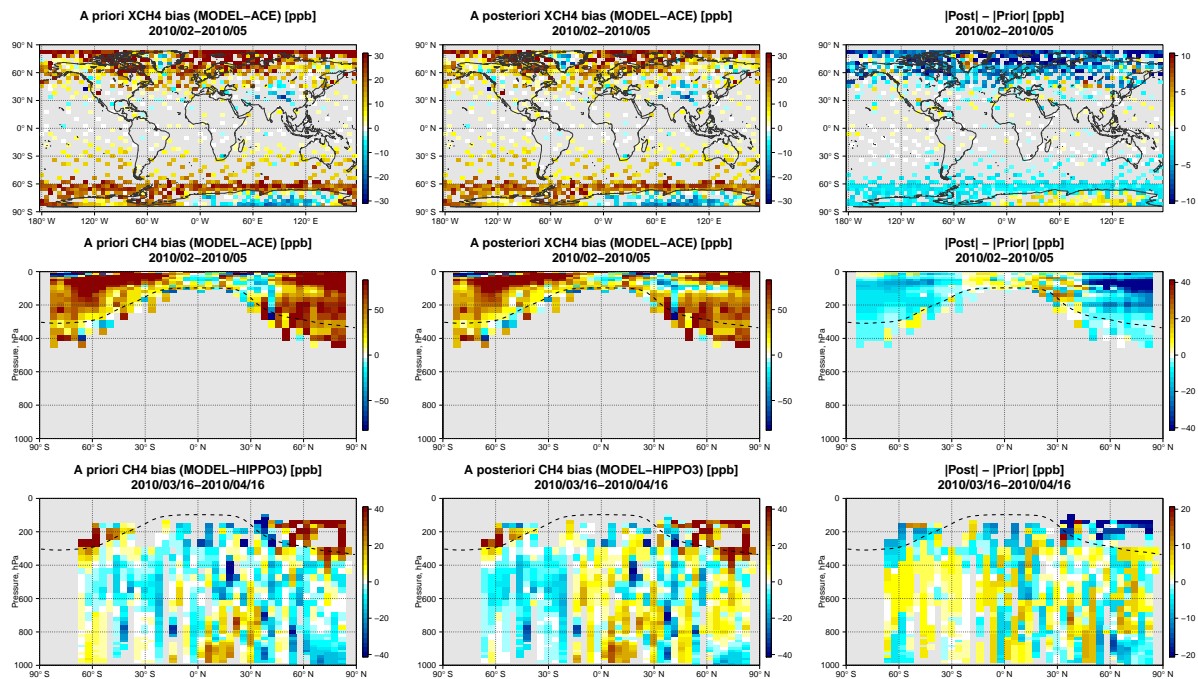

**Figure 16.** Evaluation of the mean (February-May 2010) a priori and **WC_4x5** optimized CH$_4$ fields using ACE-FTS and HIPPO-3 CH$_4$ measurements. Shown is (left column) the a priori bias, (middle column) the a posteriori bias, and (right column) the reduction in absolute bias. Top row: XCH$_4$ bias between GEOS-Chem and ACE-FTS. Middle row: zonally averaged CH$_4$ bias between GEOS-Chem and ACE-FTS. Bottom row: CH$_4$ bias between GEOS-Chem and HIPPO-3. We used ACE-FTS retrievals only in the stratosphere. The XCH$_4$ bias between GEOS-Chem and ACE-FTS was obtained by augmenting the ACE-FTS profile in the stratosphere with the GEOS-Chem profile in the troposphere and smoothing the vertical CH$_4$ profile with mean meridional GOSAT averaging kernels. The dashed line represents the mean tropopause height.



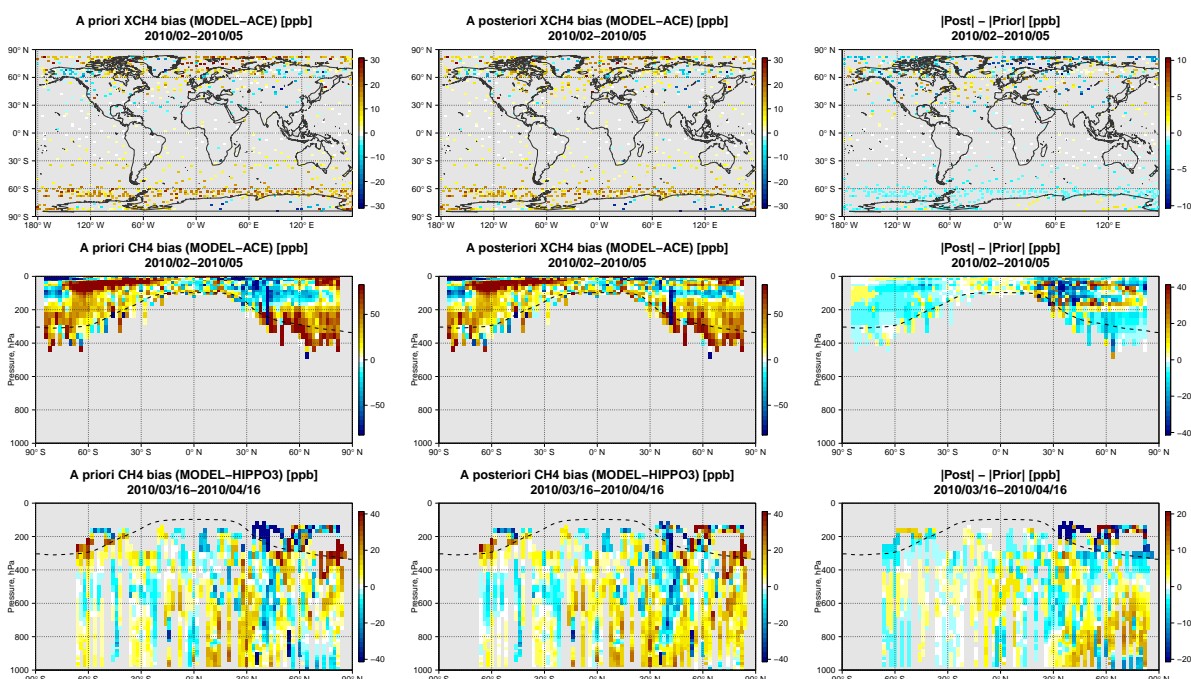

**Figure 17.** Same as Fig. 16 but for the $2° \times 2.5°$ resolution model.



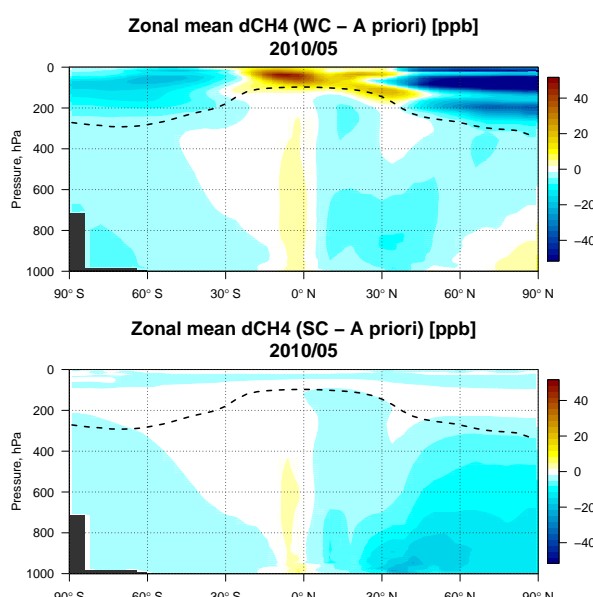

**Figure 18.** Zonal mean $CH_4$ differences (in ppb) in May 2010 between (top panel) the **WC_4x5** optimized state and the a priori fields and (bottom panel) between the **SC_4x5** optimized state and the a priori fields. The dashed line represents the mean monthly tropopause height.



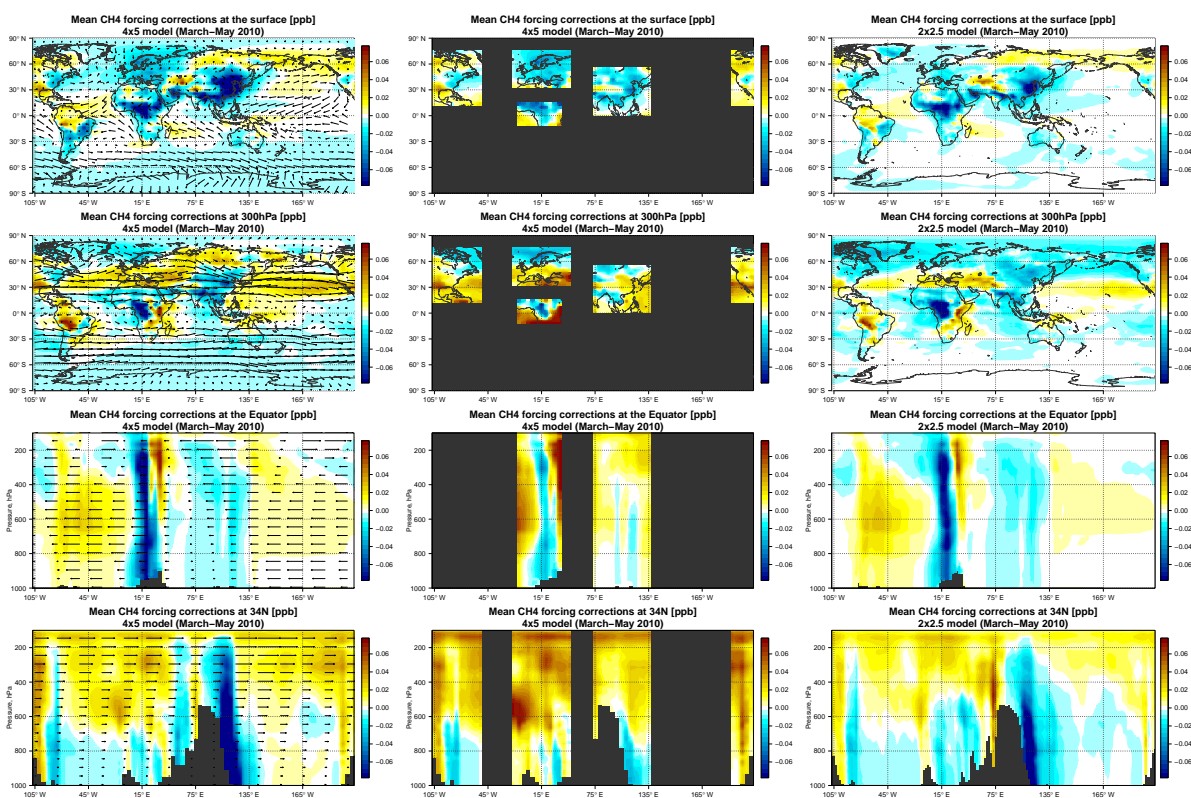

**Figure 19.** Mean optimized forcing terms (in ppb) for March-May 2010. Left column: **WC_4x5** assimilation at $4° \times 5°$. Middle column: **WC_4REG_4x5** assimilation at $4° \times 5°$. Right column: **WC_2x25** inversion at $2° \times 2.5°$. Top row: forcing terms at the surface. Second row: forcing terms at 300 hPa. Third row: altitude-longitude distribution of the forcing terms along the equator. Bottom row: altitude-longitude distribution of the forcing terms along $34°N$. In the plots in the left column, arrows represent the direction and relative magnitude of horizontal winds.