# Peer review of "Characterizing model errors in chemical transport modelling of methane: Using GOSAT XCH4 data with weak constraint four-dimensional variational data assimilation"

_Atmospheric Chemistry and Physics, 2019_

## Referee Comment (RC1) · Anonymous Referee #3 · 12 Dec 2019

The authors aim at using the information contained in GOSAT XCH4 data to correct biases in in CH4 concentrations simulated by GEOS-Chem and improve the comparison with independent data such as TCCON XCH4, NOAA surface CH4 from flasks and HIPPO data.

**General comments**

The effort at assessing the errors in the (chemistry-)transport model before going into flux inversions is necessary and the issue remains too rarely treated adequately in inversion papers. It is therefore a very good idea to present a methodology to tackle this issue. The explanations of the methodology and most of the interpretations of the results, regarding mainly biases in the transport, are clear and interesting. Nevertheless, parts of the study seem convoluted and misleading:

1. what is the purpose of the "comparison" to the so-called "strong constraint" (SC) inversion? SC is said to consider that the model is perfect but this is not true since the covariance matrix R can contain errors in the model as well as errors in the measurement, representativity errors, etc. Moreover, it is always possible to invert transport variables/parameters along with the emission fluxes and boundary conditions - which may require assimilating more data, as is suggested but in the case of WC and only for glint measurements. Finally, the comparison between two data assimilating systems would be "fair" and useful only if the same information is provided to both. Here, it is not very clear what the configuration of the SC inversion is but it seems that it includes a lot less information than WC: for example, WC uses data to estimate the noise on GOSAT XCH4 (p. 9) but it is not stated how this knowledge is used in SC. I would suggest to simply drop this "comparison" between SC and WC: the inconsistency between assessing errors in the model (WC) and inverting fluxes without taking these errors into account (as seems to be the case in SC) is too strong.

2. what is the final aim of the characterization of the model errors? Is it to improve the model? In this case, the study lacks suggestions on how to do this (change convection schemes? use only high resolutions?). Is it to gain insights on how to invert fluxes? In this case, two questions are to be answered, which are not discussed as such in the paper:

(a) are the model errors too large compared to the errors due to the emission fluxes to allow meaningful inversions of the fluxes?

(b) if inversions are to be run with the model as it is, how can the errors in the model be taken into account so that the optimized fluxes are actually meaningful (e.g recommendations on the building of the R matrix for a so-called SC set-up)?

3. what does the proposed methodology bring to the assessment of the model errors? In the discussion of the results of this study, it is not clear what it brings compared to previous studies: the errors assessed in this paper seem to be already well-known through other methodologies.

**Specific comments**

General

Beware of the use of 4D-Var: it should be used only in cases when the problem is actually 4 dimensional. This is the case in meteorology, where the problem is on the initial conditions. Usually, it is not the case for flux inversions, which are problems on the boundary conditions: the relation between the 3D maps of fluxes through time is not taken into account in the model (only the relation of concentration fields through time is). It would be best to use only "variational" and drop the 4D in these cases of flux inversions.

Abstract

- p.1, l.6-7: "capable of differentiating the vertical distribution of model errors": what does differentiating mean here?

- p.2, l.2: "indicating the presence of resolution-dependent model errors": do you mean errors directly linked to the parameterizations in the physical core of the model or representativity errors also?
- p.2, l.4: "However, a major limitation of this approach is the need to better characterize the specified model error covariance in the assimilation scheme": this is true for all data assimilating systems in our domain. What does this study bring to this issue?

Section 1 Introduction

- p.2, l.21-22: "the impact of biases in chemistry and transport are often neglected": this is not exactly true, a lot of studies try and deal with biases in various ways e.g debiasing previous to the inversion itself, specifying adapted R matrices, etc. Please look deeper into the available studies.
- p.2, l.32: " In contrast to the traditional "strong constraint" (SC) 4D-Var method, the WC scheme does not assume that the model is perfect": same remark as above, the so-called SC method does not imply that the model is assumed to be perfect, it implies that all errors are taken into account in the R (and B) matrix so that what is not is "perfect".
- p.3, l.16-17: "Highly accurate aircraft $CH_4$ profile measurements would be an ideal source of information, but they are limited in space and time.": what about aircores?

Section 2 Data and Methods

- p.4, l.11-12: "due to the dependence of wetland emissions on the meteorological fields": do you mean that wetland emissions are actually recomputed from the references provided above with the regridded GEOS meteorological fields? Or computed on-line in GEOS-Chem?

- p.4, l.21: "the period of 1 February 2010 to 31 May 2010": this is a very short period of time, which does not allow for one full seasonal cycle for mid-latitudes. Why not work on a full year?

- p.4, l.22-25: "5.5 years ... initial condition for the analysis period": this seems a bit convoluted. Why not run a spin-up of about 9 years (life time of $CH_4$) OR assimilate data to obtain initial conditions?

- p.5, l.22-25: "The use of the alternative $CO_2$ fields did not change any of the findings about model errors in our study. There may still be unidentified biases in both retrieval products. However, the fact that both $CO_2$ fields were obtained using different methodologies gives us confidence in our results." Were both fields used for all the following results? But only one set of results is presented and there is no comment about a sensitivity test. Also, why not use the default field? How was it not satisfying?

- p.5, l.30-31: "However, we expect vertical structure to emerge from atmospheric transport patterns": what about the OH field patterns (e.g its vertical structure)?

- p.6, l.4-6: "Such precision could be enough in many regions of the world to improve knowledge about $CH_4$ a priori surface emissions. However, the presence of potential model errors significantly undermines this assumption." This is not clear: which regions? Improve how? By decreasing the uncertainties by how much? What would be the required ratio between the model errors and the precision and the expected improvement of knowledge?

- p.6, l.19-23: why use only flasks and not continuous measurements?

- p.7, l.2: "Retrievals are bias corrected based on comparisons with calibrated aircraft and AirCore profiles." Which aircraft profiles? Be sure not to use them in the assimilation if the validation data is to be kept totally independent. Why not use also aircores (others than the ones used by TCCON) as

validation data?

- p.7, l.27: "the adjoint forcing commonly used in 4D-Var": please clarify what "adjoint forcing" means.

- p.7, l.21-22: "This is the assumption that is employed in standard 4D-Var, which is also referred to as "strong constraint" 4D-Var because the model trajectory is used as a strong constraint in the optimization." Same remark as in the General comments about the use of "4D-Var": the concentrations are linked through time but not the fluxes.

- p.7, l.29-30: "The 4D-Var problem to estimate surface emissions is transformed into a 3D sources and sinks estimation problem": same remark as in the General comments: flux inversions are generally not actual 4D-Var and the cost function used is the same as what is shown after, but for the Q term. Does anything prevents Q from being included in R or in B? Please clarify the mathematical and technical differences between SC and WC.

- p.9, l.9-14: how long does it take to run for practical cases?

- p.9, l.18-23: "For each WC inversion ... about 10 ppb": this is a good idea but the validation data are not independent anymore since information contained in them is actually used in the inversions. Please clarify how this issue is dealt with when comparing to the data for validation.

- p.10, l.3: "Therefore, we did not attempt to characterize global pattern of model errors on shorter time scales": even with a period of three days for GOSAT data, information is available at shorter time scales e.g. in the GOSAT data and in the meteorology. Would this not make it possible to characterize patterns of model errors at shorter time scales?

- p.10, l.4-6: "Little is known about the a priori structure of the model errors": this is not what appears from discussions and references cited afterwards: at least some elements such as the role of the horizontal resolution or the

tropo-strato gradient are known. Is this information not usable in the inversion framework?

- p.10, l. 9-end and p.11, l.1-2: the issues described here are the same as when building the R and B matrices in the so-called SC case. Therefore, the exploration of the nature of errors in the modelled $CH_4$ uses the strong assumption of a diagonal Q. What are then the advantages of this methodology compared to the usual R and B matrices in the so-called SC, with various set-ups for R for example?

- p. 11, l.9-11: "Therefore, we considered a uniform structure for Q to be a satisfactory assumption for this initial assessment of model errors in the context of the WC 4D-Var analysis": is this statement justified by "expert-knowledge"? Is it not possible to design sensitivity tests to assess the impact of this strong assumption?

- p.12, l.9: "we believe that the OSSEs should reveal the best performance of the WC method": it is a bit dangerous to show the performances of a methodology only in best-case scenarios since the application to realistic cases may show the tool to be very limited.

- p.12, l.15: "for the real world applications, we expect less extreme model errors": would the method proposed here be able to characterize the errors if they are smaller? See also comment above.

- p.12, l.19-21: "Here, we intend to investigate the performance of the measurements and the assimilation method when no information is given about the sources and magnitude of model errors": to which configuration does this sentence refer to? Is it really useful to assess the performances in a case that is almost never implemented in actual inversions (even though the taking into account of model errors is never thorough)?

- p.12, l.21-22: "We also conducted SC 4D-Var assimilation experiment for comparisons with the WC approach in the OSSE with biased surface emissions": see General comments. Without more details, it is not possible to understand the differences between WC and SC. If SC contains less information than WC, the comparison is not really meaningful and interesting.

Section 3 Results

- p.13, l.26 - p.14, l.4: how do the sensitivities in the model compare to the errors in the actual GOSAT data?
- p.14, l.7-8: "when using the SC method, we implicitly supply the assimilation with knowledge about the source of the bias": how?
- p.14, l.11-12: "Due to weak vertical sensitivity of the pseudo-data, it is difficult for the WC 4D-Var method to mitigate strong localized vertical bias." The low sensitivity is a problem for data assimilation in general. What is particular to WC here?
- p.14, l.13-14: "Instead, it compensates for the bias by applying relatively weak $CH_4$ state adjustment of the opposite sign in the column of the atmosphere above, particularly in the stratosphere": what are the consequences of the creation of such a dipole?
- p.14, l.15-20: the additional information explicitly provided to WC could also be used in SC in the covariance matrices R and B.
- p.14, l.33-34: "GOSAT retrievals possess sensitivity to biases in vertical transport and can distinguish...": this is not a property of the GOSAT data as such but of the whole data assimilation framework. Maybe "GOSAT retrievals contain information on the vertical transport, which can be used in our set-up to distinguish..." would be clearer.
- p.15, l.11-12: "we do not expect chemical biases to be as strongly localized as the biases associated with emissions and vertical transport": why?

- p.15, l.30-32: "The perfect observing system would completely remove the initial condition bias at the start of the assimilation period (on February 1). However, what is shown on February 1 is just an 8% reduction in the bias in each of the eight regions, relative to the a priori, with the rest of the bias propagated onto the assimilation period". What are the consequences of this?

- p.16, l.3-4: " This suggests that additional vertical correlation between forcing terms in the stratosphere would be beneficial to accelerate convergence in the stratosphere." Why not test the sensitivity to such correlations?

- p.16, l.11: "residual high latitude bias, which resembles noise or bias in the GOSAT observations." Is this due to the period of interest being winter in the Northern hemisphere?

- p.16, l.19-20: "the SC assimilation leaves significantly larger residual biases." What is the impact of these on the optimized fluxes and on the uncertainty reduction?

- p.17, section 3.2.1: the TCCON and NOAA data are not actually independent from the WC (see above). How is this dealt with? If they are actually independent from the SC, the comparison between SC and WC against the fit to these data is not meaningful.

- p.18, l.11: "similar errors": similar to what? The sentence is not very clear to me.

- p.18, l.9-13: these sentences seem a bit convoluted. Is the idea to state that a station such as IZO sees the upper troposphere and is best compared to the model's upper troposphere?

- p. 18, l.27: what about the impact on GOSAT retrievals of the still long nights around the North pole during the period of interest?

[Figure]

Section 4 Discussion of Model Biases

- p.19, l.22-23: "a stratospheric bias introduced in the system through the initial conditions ": is there no other way than the ICs to introduce a bias in the stratosphere?
- p.19, l.29-30: "it shows that the SC assimilation attempts to correct the positive high-latitude stratospheric $CH_4$ bias at the expense of surface emissions": this seems to be a direct consequence of the set-up of the SC inversion. More details on this inversion are required to discuss its results.
- p.21, l.6-14: can you deduce some recommendations for improvements in the model? If not, how can this knowledge be used for setting up inversions for fluxes?
- p.21, l.15-23: same questions as above: recommendations for improvements in the model? Information for flux inversion?
- p.22, l.25-26: what could be concluded? Recommendations for improvements in the model? Information for flux inversion?
- p.23, l.13-14: "Finally, the results strongly suggest that the WC assimilation and the GOSAT observations have the potential to diagnose transport errors at both model resolutions" What would prevent the assimilation of data to bring information on errors at any resolution (if the data are relevant)?

Section 5 Conclusions

- p.24, l.1-2: " However, characterizing these correlations will be challenging": is it possible at least to design sensitivity tests to explore various possibilities for these error correlations?
- p.24, l.5-6: "Initial comparisons suggested that GEOS-Chem was affected by biases not solely related to discrepancies in surface emissions." This is

not very informative as it is well known and flux inversion would be relatively straightforward otherwise.

- p.24, l.13-14: "Meanwhile, the results showed that running the a priori model at 2x2.5resolution produced better agreement with TCCON observations than the a posteriori fields from the SC 4D-Var surface emission optimization at 4x5." What can be deduced from this result?

- p.24, l.31-32: "glint measurements": a lot of other data could be assimilated, not only satellite data related to the concentrations of a given species.

- p.25, l.2: "if the model were assumed to be perfect, as is the case in SC 4D-Var": see General comments: this is not so simple.

- p.25, l.7: "Potentially, any CTM may be improved if the signal from the surface emissions can be separated from other model errors." This is only relevant if the objective is to optimize fluxes. A given CTM and set up may work very well with errors compensating one another for other objectives (e.g forecasting). Be more specific on what improvements could be made, how and to which purposes.

- p.25, l.14: "regional-scale analysis at higher spatial resolution": how in practice? The link to what it is used for in this study is not plain for me.

**Technical corrections**

- p.10, l.33: "the errors in the model $CH_4$ simulation" -> the errors in the modelled $CH_4$ concentrations?

- p.11, l.19: "The sensitivity of the GOSAT observations to the modelled state" -> The sensitivity of the equivalent of GOSAT observations to the modelled state

- p.15, l.13: "in the Fig.8" -> remove "the"

- p.18, l.10: "inland" -> island?
- p.18, l16: homogenize including/excluding Sodankyla.
- p.22, l.9: "lofted" -> lifted

Tables and Figures

- Tables 1, 2: please use consistent names for the columns in the legend and table.
- Fig. 4: the black boxes are not so easy to see: maybe use a very different color (try pink?)
- Fig. 15: do not repeat in the legend what appears in the graph itself so that is it easier to read and may be kept on one page.
- Fig. 16: reduction (third column) may be easier to evaluate in % and absolute value.

---

## Referee Comment (RC2) · Anonymous Referee #2 · 12 Dec 2019

The authors propose a novel method to quantify errors in transport modelling. It uses GOSAT measurements within a new assimilation framework. In particular, the authors propose to use part of the knowledge provided by the adjoint of the transport model to constrain model errors. I am personally a great enthusiast of using the adjoint in a more comprehensive way than what is currently done by the community. Therefore, I recommend supporting the authors in their efforts to do so, and their work should make an interesting contribution to the community.

However, the present state of their manuscript requires significant changes, clarifica-

tions and rearranging before suitable for publications.

**1 General comments**

**1.1 Structure**

With the current structure and organization, it is hard to filter the take away messages. Some details are missing, other are not necessary. I have some doubts on the choice made by the authors in the way they split the content of their work between the present manuscript and the sister paper in GMD (gmd-2019-248). At lot of details about the method itself is given here, while it would be more suitable for a model description paper in GMD. Also, many details about the method, in particular the tuning choices, are set aside, while they are critical for validating the reliability of the approach. Similarly, all the OSSEs would make more sense in a GMD paper.

Only the part directly concerning real data would be suitable for an ACP content in my opinion, as well as the discussion on model resolution?

I suggest the author dramatically reconsider the way they organize their presentation of their work to help the reader navigate through the results.

**1.2 Weak constraint vs Strong constraint**

The authors insist on sticking to the 4D-VAR formulation of the surface flux inversion problem. Such an approach, following the work from the numerical weather forecast community, is artificial and ends up in clumsy and artificial formulations. Even though the equations are correct, they are uselessly complicated. The surface flux inversion problem is a 3D-VAR problem, the time step of the NWF community having no meaning
in our case (the author implicitly acknowledge this fact by putting the emissions in the model parameters $\mathbf{p}$).

I may be wrong, but the author's formalism could be easily replaced by the classical inversion equation, simply adding model bias in the control vector. Thus, the matrix $\mathbf{Q}$ would only be a sub matrix of $\mathbf{B}$, and the Lagrangian terms would not be needed; they would be implicitly solved for in the problem as a 4D CH4 atmospheric source/sink in the transport model.

**1.3   Period of interest**

The authors chose a very short period of interest (4 months) in 2010. This seems to be guided by the availability of validation data.

Such a duration is very short considering the global atmospheric transport patterns. The author show that the biases in the initial conditions can be corrected quite quickly, which would excuse the short period. However, what about long-term biases?

In the present conditions, the WC inversion seems to only allow for short term corrections, at the cost of a loss in the mass balance. It would then limit the inversion conclusions to very regional patterns, reducing the interest of running global models...

**1.4   Uncertainty matrices**

It feels that your results are so dependent to the subjective choice of $\mathbf{Q}$ that they are hardly exploitable. It is already rather challenging to find a balance between $\mathbf{R}$ and $\mathbf{B}$ in a classical inversion. The final taste of the work as it is describe is that the method does not really fit its purpose. Quantifying biases would be as efficient with simple forward simulations as it is presented...

I am convinced that the use of the adjoint to quantify model errors is a good approach,

but with the author's framework and no additional data, it seems quite impossible to deduce any conclusive results...

**2 Specific comments**

- p.2 l.24: regional and global scale

- p.2 l.33: "assume that the model is perfect": this statement is misleading; the so-called strong-constraint inversion never assumes that the model is perfect. Errors are represented in the $\mathbf{R}$ observational error matrix. Of course, in most inversion framework, the $\mathbf{R}$ matrix is too simple and misses most of the error patterns, but that is only a technical choice in the application.

- p.4 l.7: EDGAR 2004 is quite outdated; could the author justify such a choice?

- p.4-6 Sect. 2.2.1: This section mixes observation information with preliminary studies and side conclusions. Shorten and put results in the results section if really needed or in Supplement, or in a GMD style paper

- p.6 Sect. 2.2.2: a lot of information is given about the instrument precision and techniques; are such details really needed in a OSSE paper?

- p.7 l.20: $\mathbf{p}$ vary over time? it is not clear from equation (1)

- p.7 l 26: it is not clear at all what are the dimensions and spaces of the object presented here

- p.8 eq.3: are $\mathbf{Q}$ and $\mathbf{R}$ always the same for each $i$? by design, you make it impossible to have temporal correlations in the observation space. It is often the case in practice in classical inversions, but should be highlighted as a limitation of this formulation

[Figure]

- p.8 l.10: 4D-VAR artificial and makes it hard to understand. Justified in NWF where the state is directly optimized with respect to observations, but our interest is p (surface emissions). We rather do 3D-VAR!!!

- p.8 eq.4: should $\mathbf{H}$ be $\mathbf{H_i}$? should be different at each so-called time step?

- p.8 eq.4: The Lagrangian factor part would be automatically solved with a 3D-var SC including a 3D source-sink atmospheric term...

- p.9 l.11: What is exactly the size of $u_i$? and what are i standing for? days? minutes? It is not fully clear from the text neither as it changes over the course of sections...

- p.10 l.1: It is a big problem to assume diagonal $\mathbf{B}$ (as well as diagonal $\mathbf{Q}$) as you give too many degrees of freedom to your inversion compared to the number of observations

- p.10 l.17: the choice of $u$ is very shortly justified (and unconvincingly); consider extending such justification or all the results appear untrustworthy

- p.10 l.20-30: very clearly and accurately written paragraph stating the limitations of the method. But later sections contradicts the acknowledgement of the limitations

- p.11 l.3-6: important sensitivity results! should be more extensively detailed, either in the result part, or supplement, or GMD companion paper...

- p.11 eq.4: is there a link with the cost function? the use of $J$ is misleading; the equation is artificial and does not make sense. If the purpose is to introduce the total sensitivity in eq.12, the author should rather change both eq 11 and 12 and write them with the adjoint of the model, evaluated at an increment observation vector equal to 1 at every GOSAT obs.

- p.12 l.15: with less extreme situations, the balance between $\mathbf{Q}$ and $\mathbf{B}$ is expected to be even more subtle...

- p.12 l.27: not clear what tuning you are talking about; please consider adding a table detailing all the OSSE and inversion set-ups to help the reader

- p.12 l.35: negativity bound: ad-hoc unjustified choice; might be reasonable, but needs some details

- p.13 sect.3.1: more than 2 full pages, 8 figures, a lot for one section... consider splitting

- p.13 l.27: fig.3 may be misleading; it shows the integrated 'footprint' of GOSAT observations, but the inversion uses increments depending on the deviation from the prior and observations (or truth)

- p.14 l. 7: I disagree. It only shows that the matrix $\mathbf{Q}$ you chose is incorrect for that set-up... it feels that your results are so dependent to the subjective choice of $\mathbf{Q}$ that they are hardly exploitable

- p.14 l. 31: again, it only shows that $\mathbf{Q}$ is ill specified

- p.14 fig.6: Right column: Due to ill specified Q, WC applies correction upwind in the Atlantic ocean to improve the situation over the Amazon bassin. This could be highly misleading for diagnosing model errors!

- p.15 fig 6-7-8: why not including SC in these figures for comparison?

- p.15 fig.11 and last OSSE: I didn't really get this last OSSE. why initial conditions at different dates? In the end, it is probably on of the most important OSSEs as it show the capability of the WC to correct for long-term errors; but the way it is presented makes it hard to understand

- p.18 l.34: is it really necessary to extent ACE-FTS profiles to compare with GEOS-CHEM? can't you produce real equivalents?

- p.19 l.30: such results is quite obvious with a one month assimilation window; please comment accordingly...

- p.20 l.5-10: Gives the impression of fitting pre-conceived conclusion at all costs... the results are not convincing in that direction
* * *

---

## Referee Comment (RC3) · Anonymous Referee #4 · 10 Jan 2020

General Review Comments:

This is a significant and important body of work and I believe the community will find it very interesting. The main limitations I see are in terms of the figures which don't always back up the assertions in the text. Sometimes, in the case of Fig6/Fig7, they clearly support the text but seem to fail because of lack of difference plots or poor choice of color scales. The text is sometimes hard to follow because of the multiple terms being used, eg. state/forcings/3D CH4 adjustments and parameters vs. surface fluxes. After providing the theory, it might be wise to work with common terminology.

[Figure]

Furthermore, I particularly found the assumed variability of the "forcings" in time and space, somewhat hard to follow. In summary though, the work is quite interesting and novel in its handling of the main source of error in atmospheric trace gas inversions (transport).

Specific Comments:

caption Figure 2: "and smoothed with the GOSAT averaging kernels" The authors should give the exact equation they use to calculate modeled XCH4 from their model: is the prior CH4 profile assumed in the retrieval used in this calculation, in addition to the averaging kernel, or not?

Figure 6: Fig 7 provides a motivating summary of Fig 6. However, looking at Fig 6 relative to its colorbar, it is hard to see that the two even provide the same information. I'd really consider adopting a more dynamic color palette for this image.

p6 l10: "...with smaller [modeled] XCH4 in the SH..." A difference plot would be helpful in Figure 2. I am having a very hard time seeing how XCH4 is lower in the SH in the model (top panel) than in the measurements (bottom panel) – the opposite appears to be the case, to my eye.

p6 l11: I agree that China, India, and equatorial Africa are higher in the model, but South America appears to be lower, at least over the Amazon. Again, a difference plot in Figure 2 would help.

p7 l29-on: "The 4D-Var problem to estimate surface emissions is transformed into a 3D sources and sinks estimation problem..." I think you ought to be a bit more precise with the wording here. The 4Dvar method in general can be used to estimate both surface sources/sinks and 3D fields, either using the strong- or weak-constraint dynamics. I think what you want to say is that for your application, which in the past was solving only for surface fluxes and not 3D fields, you are now allowing this 3D forcing term to be solved for, as well. You might want to mention here that you are not too concerned with

what this 3D field represents in your application – it could be a chemical source/ sink, a correction to the 3D concentration field, or a dynamical error. That comment would help those who are accustomed to thinking of this term as being solely a dynamical error term in the weak-constraint approach.

p8 L15: There is a spurious extra term in the cost function here that should lead to some "double counting" of the dynamical errors. To be specific, the last two terms in the cost function both involve the dynamical mismatches, while only one is needed, and that term ought to be: ... + [-Gu_i]ˆT [Q_i]ˆ-1 [-Gu_i] , which equals ... + (x_i - M(x_i-1,p))ˆT [Q_i]ˆ-1 (x_i - M(x_i-1,p))

If we note that lambda_i = [Q_i]ˆ-1 (x_i - M(x_i-1,p)), then there is no need to carry around the additional variable u_i, and u_i can be calculated from lambda_i as

G u_i = - [Q_i] lambda_i

Similarly, it is not necessary to have equations for the partial of L with respect to both lambda_i and u_i.

p10 L5: "... a priori estimates of the model errors were set to zero." Just to be sure here, you are setting the actual initial model errors themselves equal to zero (u_i=0) and not the assumed uncertainty in those same errors (Q_i=0), correct? Because setting Q_i=0 would be equivalent to reverting back to the hard dynamical constraint. Writing this out with the mathematical symbols would eliminate any uncertainty the reader might have on that score. p10 second paragraph: This discussion of the relative weighting of the forcing versus emission parts of the solved-for control vector points to something that could be simplified in this approach. It can be shown that the forcing vector -Gu_i at any iteration is simply [Q_i] lambda_i – that is, the adjoint state vector times the assumed model error covariance matrix. The forcing vector does not need to be solved for in the control vector – it is already solved for in the strong-constraint 4Dvar, essentially, since lambda_i is solved for. All that needs to be done to implement the weak-constraint version is to add the forcing vector (=[Q_i] lambda_i) onto the

state at each step during the forward runs (although saving lambda_i at fine temporal resolution, at the timestep of the model, for example, can take a lot of memory and I/O time). Once this is recognized, it is clear that the full parameter space of the forcing vector can be solved for, up to the temporal resolution limits just mentioned. Since it is not too clear what the magnitude of [Q_i] ought to be, some experimentation with the relative weighting between the forcing and emissions parts of the cost function is probably still needed, even with the simplification just noted.

p10 L32-34: "Instead, we use the WC 4D-Var method to optimally constrain the CH4 state and explore the nature of the errors in the model CH4 simulation." Actually, you are not really solving for an optimal estimate of the CH4 state (the 3D CH4 field at each timestep), because you have not put this in the control vector that you are solving for. This state evolves according to your model M and has dynamical constraints upon it. What you are really solving for is some sort of error term on the state, which might be thought of as a dynamical error, or a 3D source/sink (e.g. a chemical one) if one is not modeled, or an error on a modeled 3D source/sink. Since the measurements have error on them, this also allows large measurement errors to be given less importance in the problem (the measurement error is turned into a large forcing error and its effect is on the actual parameters solved-for is lessened).

p10 L33-34: "We performed two types of inversions: "full state assimilation" and "flux+state assimilation"." Point back to your description of these two cases in Section 2.3 here. I had already forgotten that you described them earlier by this point. Also, it wouldn't hurt to say explicitly that the "full state" case solves only (or mainly) for u, while the "flux+state" case solves for both u and p. Spell it out, especially since the state is defined as "x" earlier, and you don't actually solve for the full 3-D field "x" in your control vector, no? I guess you get the 3D CH4 fields by adding on the dynamical errors that you do solve for onto the state using equation (2), right?

p11 L13: This would be a good place to mention that, in all these OSSEs, the same transport model was used to generate the truth as was used in the inversion. In other

words, these are all "perfect model" experiments in all respects except the isolated error signal looked at in each experiment. This is important to say, because most readers will be thinking of the weak constraint as a way to handle general model errors, and might be thinking that you used a different transport model for the generating the truth and doing the inversion (I will actually advocate adding an OSSE of this sort below in the emissions case.)

p13 L29-31: Isn't the greater sensitivity in the upper troposphere / lower stratosphere also due to the fact that the GOSAT Xch4 averaging kernel is weighted most heavily in the mid- to upper-troposphere (it being a thermal IR measurement)? A small figure showing what the GOSAT XCH4 averaging kernel actually looks like might be valuable here.

p14 L9: "The results confirm that the SC 4D-Var method better removes CH4 biases due to emissions." This is misleading. Yes, in this setup the SC does better than the WC because it puts all the measurement information into the cause of the difference, the emissions, by design (it doesn't solve for the 3D forcing corrections) as you have noted. But in general case of when there are model errors, the WC case should solve for emissions more accurately than the SC case. It would be really useful if you could have done an additional OSSE here in which you introduced emissions errors AND dynamical errors in the truth, then tried to estimate both using the SC case and the WC "flux+state" setup. That ought to show the WC case doing better, since it would correctly partition the dynamical and emissions errors to the forcing terms and emissions parameters, whereas the SC would incorrectly attribute the dynamical errors solely as emissions errors. You should use a completely different transport model in generating the truth from what is used in the inversion to make this case realistic. The dynamical errors in Q should be based on the differences between the two transport models. I think that that new OSSE would demonstrate the benefit of the WC approach for that case that a lot of readers care about: how much their surface emissions estimates are degraded by model error when using the SC approach.

p16 L15-18: I can not tell from Fig 12 and 13 that the high latitude a priori bias is smaller in 2x2.5 compared to 4x5. Similar question for the model bias reduction. It may just be that the color scale saturates so quickly in the images? Are they zonal avg summary stats you could provide to back this up?

p17 L10: "The results suggest that GEOS-Chem a priori CH4 simulation suffered from biases that were not related only to incorrect surface emissions." Since GEOS-Chem is not perfect (i.e. the real world has different transport) this is the most obvious source of the errors.

p18 L5: I find it a bit perplexing that 2x2.5 sims are generally argued to provide better atmospheric transport but perform more poorly against in situ obs, which one would think would be the most sensitive to things like vertical transport. Would you consider this an indication of residual error in surface CH4 fluxes?

Tables 1 & 2: These comparisons to independent data are powerful demonstrations that using the weak constraint approach on the GOSAT XCH4 data really does improve the estimated 3D CH4 fields better than the SC approach. Apparently either the transport or the measurements are bad enough that the WC results in a big improvement. If the GOSAT data are badly biased, adding flex in the state trajectory would help things, even if the transport were not too bad.

p22 L30: I get the averaging out of resolved eddy motions causing problems but what are you referring to by "incorrect" regridding?

p23 L28: "Despite having almost flat averaging kernels in the troposphere..." Show a plot of the GOSAT XCH4 averaging kernel – since GOSAT measures it in a thermal IR band, I would have thought that the averaging kernel would be mainly sensitive to the upper troposphere.

Minor Comments: p5 line 13: should this read "and the GHG_CCI group"??

p5 L20: "produced a comparable fit...": to what, the old three-model suite used in the

retrievals?

p5 L24: you should say GOSAT "XCH4" here, since that is what you are looking at; "CH4" does better than that, I think.

P21 L25-26: Reword these lines, I don't think the CH4 budget is conserved because there is bias in the stratosphere induced by transport error.

P23 l8: add a comma before "are"

In general, I would prefer full descriptions like "the difference between A and B" as opposed to the "bias in A". The term "bias" means a lot of different things, and can relative to another unspecified quantity, and using the full description is always going to be less confusing albeit more wordy.
* * *

---

## Author Comment (AC1) · 3 Dec 2020

We thank the referees for their careful and thoughtful comments on the manuscript. We also thank the editor are giving us additional time to respond to the comments. We will begin with a general response to all referees, followed by a more detailed response to the comments from each referee.

Referees #2 and #4 both expressed concern with referring to the flux inversion approach as 4D-Var. Referee #2 suggested that it is a 3D-Var problem, whereas Referee #3 suggested we "drop the "4D" and "use only variational" to describe the approach. We would like to stress that this approach has been widely referred to as "4D-Var" by the inverse modeling community for quite some time. This has been the case in inversion analyses to estimate CO2 fluxes (e.g. Basu et al., 2013, Deng et al., 2016; Zheng et al., 2018), CH4 emissions (e.g., Meirink et al., 2008; Bergamaschi et al., 2010; Turner et al. 2015), CO emissions (e.g., Kopacz et al., 2010, Jiang et al., 2017), NOx and SO2 emissions (Qu et al., 2019), and emissions of SOx, NOx, and NH3 (e.g., Henze et al., 2009). We agree with the referees that the 4D-Var methodology in the flux inversion context is different from that in numerical weather prediction (NWP), but it is unclear what terminology would be better. It is not a 3D-Var scheme as the fluxes are time dependent. In our analysis we solve for monthly CH4 emission over a four-month period (February – May). This means that the February fluxes are being influenced by observations from February to May, relying on the adjoint model to propagate information from observations in the future back in time to update the February emissions. If we were solving for mean fluxes over the whole four-month period, it would be more similar to a 3D-Var approach. In addition, just referring to the approach as a variational scheme is vague. It might be a useful exercise for the inverse modeling community to develop new terminology for these methods that have been adapted from the NWP community, but that is not an objective of this study. Introducing new terminology for the flux inversion community would require a methodology study that clearly highlights the similarities and differences in how the 4D-Var technique is used in the flux inversion and NWP communities so that it can justify the proposed terminology.

Another concern raised by Referees #2 and #4 is our statement that the model is assumed perfect in strong constraint 4D-Var. The referees stated several times that this is not true. We should have been more precise in our wording. In strong constraint 4D-Var it is assumed that the model perfectly evolves the state in time. As noted by Trémolet (2007), "current operational implementations of 4D-Var rely on the assumption that the numerical model representing the evolution of the atmospheric flow is perfect, or at least that model errors are small enough (relative to other errors in the system) to be neglected." The full 4D-Var cost function is

$$J(\mathbf{x}) = \sum_{i=0}^{N} \frac{1}{2} (\mathbf{y}_{i} - \mathbf{H}_{i} \mathbf{x}_{i})^{T} \mathbf{R}_{i}^{-1} (\mathbf{y}_{i} - \mathbf{H}_{i} \mathbf{x}_{i}) + \sum_{i=1}^{N} \frac{1}{2} [\mathbf{x}_{i} - M_{i} (\mathbf{x}_{i-1})]^{T} \mathbf{Q}_{i}^{-1} [\mathbf{x}_{i} - M_{i} (\mathbf{x}_{i-1})] + \frac{1}{2} (\mathbf{x} - \mathbf{x}^{b})^{T} \mathbf{B}^{-1} (\mathbf{x} - \mathbf{x}^{b})$$

where **x** is the model state,  $\mathbf{y}_i$  are the observations, **H** is the observation operator, and **R**, **B**, and **Q** are is the observation, background, and model error covariance matrices, respectively. Here *M* is the forecast model that evolves the state in time from time *t* to *t*+1 as

$$\mathbf{x}_{i+1} = M_{i+1}(\mathbf{x}_i) + \mathbf{u}_{i+1}$$

with model error **u**. In the cost function above, we are solving for the full 4D state vector. This is weak constraint 4D-Var. However, by assuming that the model evolution is perfect we can neglect

the second term in the cost function and optimize only the initial state  $(x_0)$  (since this initial state is evolved forward in time without error). Thus, the cost function becomes

$$J(\mathbf{x}_0) = \sum_{i=0}^{N} \frac{1}{2} (\mathbf{y}_i - \mathbf{H}_i \mathbf{x}_i)^T \mathbf{R}_i^{-1} (\mathbf{y}_i - \mathbf{H}_i \mathbf{x}_i) + \frac{1}{2} (\mathbf{x}_0 - \mathbf{x}^b)^T \mathbf{B}^{-1} (\mathbf{x}_0 - \mathbf{x}^b)$$

where  $\mathbf{x}_i = M_{i,0}(\mathbf{x}_0)$  is the state at time *i* based on the perfect evolution of the initial condition  $\mathbf{x}_0$  forward in time. This is strong constraint 4D-Var. We have modified the text to make it clear that the model evolution is assumed to be perfect in strong constraint 4D-Var.

It was suggested by the referees that the **R** matrix accounts for model errors in strong constraint 4D-Var. In theory, **R** accounts for errors in the observations and for representative errors in the observation operator. It does not account for errors in the propagation of the state in time. In the strong constraint approach, one could adjust **R** to capture errors in the evolution of the state, but that would be inconsistent with the framework. Nevertheless, this is what is done in practice. It was also suggested by the referees that there are alternative means of estimating the model bias, and we agree. For example, one could incorporate the bias **b** into the first term of the cost function (i.e.  $[\mathbf{y}_i - \mathbf{H}(\mathbf{x}_i - \mathbf{b}_i)]$  and solve for the bias together with  $\mathbf{x}_0$  (i.e., *J* becomes a function of  $\mathbf{x}_0$  and **b**). A similar approach was proposed by Dee and DaSilva (1998) in the context of a sequential assimilation scheme. We are not claiming that our approach is the only means of estimating model bias. We have adapted the weak constraint 4D-Var framework, as described by Trémolet (2007), for the GEOS-Chem 4D-Var scheme because in the context of a 4D-Var assimilation framework, the weak constraint scheme provides a means of estimating the model bias that is consistent with the 4D-Var formalism.

**References**

- Basu, S., et al., Global CO2 fluxes estimated from GOSAT retrievals of total column CO2, Atmos. Chem. Phys., 13, 8695–8717, https://doi.org/10.5194/acp-13-8695-2013, 2013.
- Bergamaschi, P., et al., Inverse modeling of European CH4 emissions 2001–2006, J. Geophys. Res., 115, D22309, doi:10.1029/2010JD014180, 2010.
- Deng, F., et al., Combining GOSAT XCO2 observations over land and ocean to improve regional CO2 flux estimates, J. Geophys. Res. Atmos., 121, 1896–1913, doi:10.1002/2015JD024157.
- Dee, D. P., and Da Silva, A. M., Data assimilation in the presence of forecast bias, Q. J. R. Meteorol. Soc., 124, 269–295, 1998.
- Henze, D. K., Seinfeld, J. H., and Shindell, D. T., Inverse modeling and mapping US air quality influences of inorganic PM2.5precursor emissions using the adjoint of GEOS-Chem, Atmos. Chem. Phys., 9, 5877–5903, https://doi.org/10.5194/acp-9-5877-2009, 2009.
- Kopacz, M., et al., Global estimates of CO sources with high resolution by adjoint inversion of multiple satellite datasets (MOPITT, AIRS, SCIAMACHY, TES), Atmos. Chem. Phys., 10, 855–876, https://doi.org/10.5194/acp-10-855-2010, 2010.
- Jiang, Z., et al., A 15-year record of CO emissions constrained by MOPITT CO observations, Atmos. Chem. Phys., 17, 4565–4583, https://doi.org/10.5194/acp-17-4565-2017, 2017.
- Meirink, J. F., Bergamaschi, P., and Krol, M. C., Four-dimensional variational data assimilation for inverse modelling of atmospheric methane emissions: method and comparison with synthesis inversion, Atmos. Chem. Phys., 8, 6341–6353, https://doi.org/10.5194/acp-8-6341-2008, 2008.

Qu, Z., Henze, D. K., Theys, N., Wang, J., & Wang, W., Hybrid mass balance/4D-Var joint inversion of NOx and SO2 emissions in East Asia. Journal of Geophysical Research: Atmospheres, 124, 8203–8224. https://doi.org/10.1029/2018JD030240

Trémolet, Y., Model-error estimation in 4D-Var, Q. J. R. Meteorol. Soc. 133: 1267–1280, 2007.

- Turner, A. J., et al.: Estimating global and North American methane emissions with high spatial resolution using GOSAT satellite data, Atmos. Chem. Phys., 15, 7049–7069, https://doi.org/10.5194/acp-15-7049-2015, 2015.
- Zheng, T., French, N. H. F., and Baxter, M.: Development of the WRF-CO2 4D-Var assimilation system v1.0, Geosci. Model Dev., 11, 1725–1752, https://doi.org/10.5194/gmd-11-1725-2018, 2018.

**Responses to Individual Referees**

The comments from the referees are in **bold Calibri** font and our responses are in plain Times New Roman font.

**Anonymous Referee #2**

The authors propose a novel method to quantify errors in transport modelling. It uses GOSAT measurements within a new assimilation framework. In particular, the authors propose to use part of the knowledge provided by the adjoint of the transport model to constrain model errors. I am personally a great enthusiast of using the adjoint in a more comprehensive way than what is currently done by the community. Therefore, I recommend supporting the authors in their efforts to do so, and their work should make an interesting contribution to the community.

However, the present state of their manuscript requires significant changes, clarifications and rearranging before suitable for publications.

1 General comments

**1.1 Structure**

With the current structure and organization, it is hard to filter the take away messages. Some details are missing, other are not necessary. I have some doubts on the choice made by the authors in the way they split the content of their work between the present manuscript and the sister paper in GMD (gmd-2019-248). At lot of details about the method itself is given here, while it would be more suitable for a model description paper in GMD. Also, many details about the method, in particular the tuning choices, are set aside, while they are critical for validating the reliability of the approach. Similarly, all the OSSEs would make more sense in a GMD paper.

Only the part directly concerning real data would be suitable for an ACP content in my opinion, as well as the discussion on model resolution?

**I suggest the author dramatically reconsider the way they organize their presentation of their work to help the reader navigate through the results.**

We agree that the manuscript was long and consequently it was difficult to navigate through the results. We have significantly shortened the OSSE section. We have removed one of the four OSSE experiments (the perturbed flux experiment), based on the comments from Referee #4, and we have removed the GOSAT observational coverage sensitivity experiment. Overall, we have removed 6 figures from the manuscript. We have also improved the description of the remaining experiments so that it is easier to follow the results.

**1.2 Weak constraint vs Strong constraint**

The authors insist on sticking to the 4D-VAR formulation of the surface flux inversion problem. Such an approach, following the work from the numerical weather forecast community, is artificial and ends up in clumsy and artificial formulations. Even though the equations are correct, they are uselessly complicated. The surface flux inversion problem is a 3D-VAR problem, the time step of the NWF community having no meaning in our case (the author implicitly acknowledge this fact by putting the emissions in the model parameters p).

I may be wrong, but the author's formalism could be easily replaced by the classical inversion equation, simply adding model bias in the control vector. Thus, the matrix Q would only be a sub matrix of B, and the Lagrangian terms would not be needed; they would be implicitly solved for in the problem as a 4D CH4 atmospheric source/sink in the transport model.

Please see our general response above regarding the widespread use of the 4D-Var approach by the inverse modeling community. In addition, the emission optimization problem in our analysis is not 3D, as the emissions are time dependent and rely on the model transport to use observations in the future to quantify the fluxes in a given month. We are taking advantage of the fact that the 4D-Var scheme is a smoother to quantify the monthly mean fluxes over the entire assimilation period.

Yes, one can use the classical inversion equation and add the model bias to the control vector, as we noted in our general response above. However, the weak constraint 4D-Var approach is an alternate approach, in which we are adding the bias term to the control vector, but in a manner that is consistent with the 4D-Var formalism.

**1.3 Period of interest**

The authors chose a very short period of interest (4 months) in 2010. This seems to be guided by the availability of validation data.

Such a duration is very short considering the global atmospheric transport patterns. The author show that the biases in the initial conditions can be corrected quite quickly, which would excuse the short period. However, what about long-term biases?

In the present conditions, the WC inversion seems to only allow for short term corrections, at the cost of a loss in the mass balance. It would then limit the inversion conclusions to very regional patterns, reducing the interest of running global models...

The main goal of the inversion analysis is to quantify regional sources of CH4 (using regional and global models). We are trying the use the imprint of fresh emissions on atmospheric CH4, as measured by GOSAT, to infer the surface emissions. Long simulations are not helpful in this regard as the emission signals become well mixed into the background on long timescales. However, biases associated with transport on long timescales can adversely impact the inversions. For example, a bias in the Brewer-Dobson circulation or in the representation of the polar vortex can result in a bias in CH4 in the lower stratosphere, which will produce a bias in the modeled XCH4. Such a bias cannot be mitigated by a correction in the surface emissions on short timescales, given the age of air in the lower stratosphere, but that would not be desirable in any case since our interest is in estimating emissions that are free of the signature of model biases. Our results show that the weak constraint approach can mitigate such biases in the stratosphere on emission-relevant timescales.

Also, we note that there is no loss in mass balance. The model is mass conserving. The weak constraint is adding sources and sinks throughout the atmosphere to correct for the model errors, as opposed to putting these sources and sinks only at the surface. In doing so, it is redistributing the mass of  $CH_4$  in the atmosphere to better fit the  $CH_4$  mass suggested by the GOSAT measurements.

**1.4 Uncertainty matrices**

It feels that your results are so dependent to the subjective choice of Q that they are hardly exploitable. It is already rather challenging to find a balance between R and B in a classical inversion. The final taste of the work as it is describe is that the method does not really fit its purpose. Quantifying biases would be as efficient with simple forward simulations as it is presented...

I am convinced that the use of the adjoint to quantify model errors is a good approach, but with the author's framework and no additional data, it seems quite impossible to deduce any conclusive results...

The motivation for the study was the fact that there is increasing evidence that the vertical transport in the GEOS-Chem model is too weak, and that latitudinal dependent biases in XCH4 in the model are problematic for CH4 inversion analyses using the model (e.g., Turner et al., 2015). Therefore, we decided to apply the weak constraint approach, as described by Trémolet (2007), to try to better characterize these biases. A major concern was that assimilation of GOSAT XCH4 data, which do not provide vertical profile information, would be inadequate for capturing the model errors. Consequently, we chose to assume the simplest form for  $\mathbf{Q}$  to see what structures in the model error can be identified given the GOSAT column observations. Imposing prior structure in  $\mathbf{Q}$ would have made it more challenging to determine if the XCH4 data, when assimilated into a weak constraint 4D-var framework, can provide constraints in the vertical structure of the model error. Surprisingly, the results show that the estimated bias field is consistent with our developing understanding of the vertical transport bias in GEOS-Chem. For our follow-up application of the approach, we are now trying to develop a better  $\mathbf{Q}$  matrix to further improve the performance of the system.

Reference

Turner, A. J., et al.: Estimating global and North American methane emissions with high spatial resolution using GOSAT satellite data, Atmos. Chem. Phys., 15, 7049–7069, https://doi.org/10.5194/acp-15-7049-2015, 2015.

**2 Specific comments**

**• p.2 l.24: regional and global scale**

Transport can have global impacts, but the focus is on the regional impacts that confound efforts to quantify regional emission estimates of CH4.

• p.2 1.33: "assume that the model is perfect": this statement is misleading; the so- called strong-constraint inversion never assumes that the model is perfect. Errors are represented in the R observational error matrix. Of course, in most inversion framework, the R matrix is too simple and misses most of the error patterns, but that is only a technical choice in the application.

This should have stated "assume that the model evolution is perfect." We have corrected the text. Please see our general response above regarding the nature of the errors captured by the  $\mathbf{R}$  matrix.

• p.4 l.7: EDGAR 2004 is quite outdated; could the author justify such a choice?

This is an older inventory, but it does provide a reasonable a priori for the assimilation of GOSAT data for 2010 conditions. It would be difficult to justify the use of the 2004 inventory for simulation of 2020 conditions, for example.

• p.4-6 Sect. 2.2.1: This section mixes observation information with preliminary studies and side conclusions. Shorten and put results in the results section if really needed or in Supplement, or in a GMD style paper

We have removed Figures 1 and 2 and shortened the section.

**• p.6 Sect. 2.2.2: a lot of information is given about the instrument precision and techniques; are such details really needed in a OSSE paper?**

This is not an OSSE paper. We start with the OSSEs to assess the performance of the system, but the focus is on the assimilation of the GOSAT data. The description of the data used for validation of the assimilation results is important to help the reader interpret our results.

**• p.7 l.20: p vary over time? it is not clear from equation (1)**

The fluxes do vary over time. We solve for monthly fluxes over the assimilation period. We have added text to make this point clear on Page 7, lines 16–17.

• p.7 l 26: it is not clear at all what are the dimensions and spaces of the object presented here

We have added text to specify the dimension of the variables.

• p.8 eq.3: are Q and R always the same for each i? by design, you make it impossible to have temporal correlations in the observation space. It is often the case in practice in classical inversions, but should be highlighted as a limitation of this formulation

In theory, **R** and **Q** change for each i, but in practice, we keep Q constant in our analysis. We now explicitly remind the reader of this on Page 8.

• p.8 l.10: 4D-VAR artificial and makes it hard to understand. Justified in NWF where the state is directly optimized with respect to observations, but our interest is p (surface emissions). We rather do 3D-VAR!!!

Please see our general comment above. We are not using a 3D-Var approach.

**• p.8 eq.4: should H be Hi? should be different at each so-called time step?**

Yes, H is different for each observation that is assimilated. We have corrected that.

• p.8 eq.4: The Lagrangian factor part would be automatically solved with a 3D-var SC including a 3D source-sink atmospheric term...

That is an alternative inversion approach using 3D-Var. Inversion analyses using the 4D-Var scheme benefit from the fact that 4D-Var is a smoother, which allows us to more effectively exploit observations that are distributed in time.

• p.9 l.11: What is exactly the size of ui? and what are i standing for? days? minutes? It is not fully clear from the text neither as it changes over the course of sections...

The size can be equal to or smaller than the size of  $\mathbf{x}$ , depending on the spatial extent over which the model errors are quantified. We have added text to clarify this on Page 8 (line 1).

• p.10 l.1: It is a big problem to assume diagonal B (as well as diagonal Q) as you give too many degrees of freedom to your inversion compared to the number of observations.

The chemical data assimilation problem, whether we are solving for the state or emissions, is an underdetermined problem. We agree that imposing structure in the covariance matrices will help constrain the solution. However, as we noted in our response to General Comment 1.4, we chose the simplest representation of Q since it was unknown to what extent we would be able to capture the spatial patterns in the model error using the GOSAT XCH4 data (given that we use only data over land and are not assimilating CH4 profiles). Without good knowledge of the structure of Q, the most conservative approach is to assume that it is diagonal. In a similar vein, in the flux inversion community it is not uncommon to assume that B is diagonal (e.g., Turner et al. 2015, Jiang et al., 2017). Estimating the covariances for different emission types (e.g. for wetlands, agricultural, fossil fuels) is challenging, and assuming no structure is safer than imposing incorrect structure in the analysis.

**• p.10 l.17: the choice of u is very shortly justified (and unconvincingly); consider extending such justification or all the results appear untrustworthy**

As we explained in the manuscript, the scaling ũ adjusts the direction of the gradient descent, between optimizing the emissions and the state. We believe that it would impair the reader's ability to navigate through the results if we were to expand the manuscript and present a detailed discussion of the sensitivity of the optimization algorithm to ũ.

• p.10 l.20-30: very clearly and accurately written paragraph stating the limitations of the method. But later sections contradicts the acknowledgement of the limitations.

It is unclear which sections the referee thinks contradict our discussion here. We were careful in highlighting the limitations, and our results were presented in the context of these limitations.

**• p.11 I.3-6: important sensitivity results! should be more extensively detailed, either in the result part, or supplement, or GMD companion paper...**

As we described in the manuscript, we conducted a series of sensitivity analyses using a range of values of q between 0.05 ppb and 2000 ppb, and found that the validation results (similar to what is shown in Figure 15 in the original manuscript) did not change for values of q larger than about 50 ppb. As a result, we assumed a value of 50 ppb for q. We do not believe that adding individual plots of the validation results for a range of values of q to the already long manuscript would be any more informative than the description of the results already in the manuscript.

• p.11 eq.4: is there a link with the cost function? the use of J is misleading; the equation is artificial and does not make sense. If the purpose is to introduce the total sensitivity in eq.12, the author should rather change both eq 11 and 12 and write them with the adjoint of the model, evaluated at an increment observation vector equal to 1 at every GOSAT obs.

Our description of this experiment was confusing. The cost function here is different from the weak constraint cost function. In the interest of shortening the OSSE section, we have removed this experiment.

• p.12 l.15: with less extreme situations, the balance between Q and B is expected to be even more subtle...

The objective with the OSSEs was to assess whether the XCH4 data is able to provide information to help mitigate the model bias. Since these were OSSEs in which the pseudo-observations were generated with the same model, we chose extreme biases for the evaluation. The results showed that even when convection was turned off, the model was able to partially mitigate the model bias. They suggested that XCH4 data should be able to correct more modest model biases that may be present in real inversion analyses. Yes, the balance between B and Q would be more subtle with smaller errors. This will be an issue in the case where the model errors mimic emissions errors, such as a bias in the chemical sink for CH4. However, it will not be an issue for transport errors, which cannot be corrected for by adjusting surface emissions. If the biases mimic emission errors, additional information will be needed, such as by assimilating formaldehyde (HCHO) data to provide constraints on the hydroxyl radical (OH). And this will be an issue for any inversion analysis, regardless of the bias correction scheme employed.

**• p.12 l.27: not clear what tuning you are talking about; please consider adding a table detailing all the OSSE and inversion set-ups to help the reader**

This was poorly worded. We meant that we conducted a number of experiments to evaluate the impact of the selected window length on the analysis. We have rewritten this section to more clearly describe the experiments.

**• p.12 l.35: negativity bound: ad-hoc unjustified choice; might be reasonable, but needs some details**

It is ad hoc. But, as we noted in the text, it was previously shown that the GEOS-Chem CH4 simulation was positively biased in the extratropical stratosphere at a resolution of  $4^{\circ} \times 5^{\circ}$ . We chose the negativity bound to speed up convergence in the extratropical stratosphere, to remove this known bias. The L-BFGS-B algorithm provides a means of doing this as part of the optimization.

**• p.13 sect.3.1: more than 2 full pages, 8 figures, a lot for one section... consider splitting**

As we mentioned above in our response to general comment 1.1, we significantly shortened this section. In doing so, we have removed 4 figures.

• p.13 l.27: fig.3 may be misleading; it shows the integrated 'footprint' of GOSAT observations, but the inversion uses increments depending on the deviation from the prior and observations (or truth)

Yes, the inversion uses the increment, but it is unclear how one would show the spatial distribution of the sensitivity for each increment over the whole 4-month period, that is easy for the reader to interpret. The integrated sensitivity over the four months seemed to be the best approach. Regardless, we have removed the figure to shorten the manuscript.

**• p.14 l. 7: I disagree. It only shows that the matrix Q you chose is incorrect for that set-up... it feels that your results are so dependent to the subjective choice of Q that they are hardly exploitable**

We respectfully disagree with the referee. The specification of Q is not the issue here. As Referee #4 noted, "in this setup the strong constraint assimilation does better than the weak constraint because it puts all the measurement information into the cause of the difference, the emissions, by design." Because we have put all of the bias in the emissions, at the surface, it is not surprising that the strong constraint assimilation, that assumes all of the bias is in the emissions, outperforms the weak constraint assimilation. In retrospect, given the comment from Referee #4, we realize that this experiment is not a meaningful evaluation of the weak constraint assimilation. We have therefore removed this experiment.

- p.14 l. 31: again, it only shows that Q is ill specified
- p.14 fig.6: Right column: Due to ill specified Q, WC applies correction upwind in the Atlantic ocean to improve the situation over the Amazon basin. This could be highly misleading for diagnosing model errors!

This discussion on Page 14 (starting from Line 25) was regarding the results of the OSSE in which we turned off convection. As shown in Figure 6, turning off convection results in an accumulation of CH4 near the surface over the continental source regions, and a deficit aloft, downwind of the source regions. The positive correction to the deficit over the Atlantic and the negative correction to the excess CH4 over the Amazon (and central Africa) is exactly what we hope the bias correction would do. It is not an indication of an ill-specified Q. Clearly, changing Q will impact the small-scale features in the corrections, but the large-scale corrections to the excess CH4 near the source regions and to the deficit aloft is an indication that the scheme is working.

**• p.15 fig 6-7-8: why not including SC in these figures for comparison?**

We did not impose any flux errors in these experiments, consequently there would be no value in using the strong constraint assimilation in them. Our objective here is simply to evaluate the performance of the weak constraint scheme to mitigate the convection bias.

• p.15 fig.11 and last OSSE: I didn't really get this last OSSE. why initial conditions at different dates? In the end, it is probably on of the most important OSSEs as it show the capability of the WC to correct for long-term errors; but the way it is presented makes it hard to understand

Figure 11 (in the original manuscript) was not well-described. We only introduce a bias in the initial conditions for 1 February 2010. Figure 11 showed how that initial bias decreased in time in different regions of the atmosphere. We have removed this figure and instead briefly summarize the main result in the text.

**• p.18 l.34: is it really necessary to extent ACE-FTS profiles to compare with GEOS-CHEM? can't you produce real equivalents?**

To map the ACE-FTS data into XCH4 space we do need to extend the ACE-FTS profile to construct the full column. However, when looking at the differences between the modeled and ACE-FTS XCH4, the tropospheric component of the column from GEOS-Chem is removed, leaving only the real differences in the stratosphere and UTLS. To use real observations to extend the ACE-FTS data, we need global profile observations of CH4 that have near spatial and temporal coincidence with the ACE-FTS measurements. We are not aware of such observations for the period of interest.

• p.19 l.30: such results is quite obvious with a one month assimilation window; please comment accordingly...

Please see our response to General Comment 1.3 above.

• p.20 I.5-10: Gives the impression of fitting pre-conceived conclusion at all costs... the results are not convincing in that direction

In all data assimilation applications, if there is a known bias in the model it is important to try to mitigate that bias. It is not an issue of fitting a pre-conceived conclusion in the assimilation. The positive bias in the GEOS-Chem stratosphere at the  $4^{\circ} x5^{\circ}$  resolution was previous identified, as we noted in the text. In their inversion analysis, Maasakkers et al. (2019) accounted for this bias by fitting a second order polynomial to the modeled background as a function of latitude. We

showed that the weak constraint scheme can mitigate this bias in the context of the assimilation. That is a desirable outcome. In addition, as we noted, at  $2^{\circ} \times 2.5^{\circ}$ , the bias changed sign between  $30^{\circ}$ – $40^{\circ}$ N, and the weak constraint scheme was able to reduce both the positive and negative bias in the stratosphere. No one had previously documented the difference in the bias at  $2^{\circ} \times 2.5^{\circ}$ , yet the assimilation was able to mitigate it.

**Reference:**

Maasakkers, J. D., et al.: Global distribution of methane emissions, emission trends, and OH concentrations and trends inferred from an inversion of GOSAT satellite data for 2010–2015, Atmos. Chem. Phys., 19, 7859–7881, https://doi.org/10.5194/acp-19-7859-2019, 2019.

**Anonymous Referee #3**

The effort at assessing the errors in the (chemistry-)transport model before going into flux inversions is necessary and the issue remains too rarely treated adequately in in- version papers. It is therefore a very good idea to present a methodology to tackle this issue. The explanations of the methodology and most of the interpretations of the results, regarding mainly biases in the transport, are clear and interesting. Nevertheless, parts of the study seem convoluted and misleading:

 what is the purpose of the "comparison" to the so-called "strong constraint" (SC) inversion? SC is said to consider that the model is perfect but this is not true since the covariance matrix R can contain errors in the model as well as errors in the measurement, representativity errors, etc. Moreover, it is always possible to invert transport variables/parameters along with the emission fluxes and boundary conditions - which may require assimilating more data, as is suggested but in the case of WC and only for glint measurements. Finally, the comparison between two data assimilating systems would be "fair" and useful only if the same information is provided to both. Here, it is not very clear what the configuration of the SC inversion is but it seems that it includes a lot less information than WC: for example, WC uses data to estimate the noise on GOSAT XCH4 (p. 9) but it is not stated how this knowledge is used in SC. I would suggest to simply drop this "comparison" between SC and WC: the inconsistency between assessing errors in the model (WC) and inverting fluxes without taking these errors into account (as seems to be the case in SC) is too strong.

The comparison between the weak constraint and strong constraint assimilation is intended to highlight the impact of not accounting for the errors in the strong constraint assimilation. The two assimilation schemes are ingesting exactly the same information. There is no difference in the GOSAT data assimilated by the two schemes. The only difference is that the strong constraint scheme adjusts the surface emissions, whereas the weak constraint adjusts the surface emissions as well 3D sources and sinks in the model state. We have expanded the description of the assimilation schemes in Section 2.3 to better explain the strong constraint method.

Regarding the issues as to whether the model is considered perfect in strong constraint 4D-Var, and the possibility of capturing model errors in the  $\mathbf{R}$  matrix, please see our general comment above.

- 2. what is the final aim of the characterization of the model errors? Is it to improve the model? In this case, the study lacks suggestions on how to do this (change convection schemes? use only high resolutions?). Is it to gain insights on how to invert fluxes? In this case, two questions are to be answered, which are not discussed as such in the paper:
  - a) are the model errors too large compared to the errors due to the emission fluxes to allow meaningful inversions of the fluxes?
  - b) if inversions are to be run with the model as it is, how can the errors in the model be taken into account so that the optimized fluxes are actually meaningful (e.g recommendations on the building of the R matrix for a so- called SC set-up)?

Previous work by Yu et al. (2017), based on the GEOS-Chem simulation of 222Rn, 210Pb, and 7Be, suggested that the vertical transport in the model is too weak at the coarse resolution. We therefore applied the weak constraint assimilation here to characterize these biases in the context of the CH4 simulation. In our companion paper in GMD we used the insight gained from the assimilation work presented here to better understand the source of the CH4 biases. As we stated in Stanevich et al. (2020) in GMD, that paper "complements Yu et al. (2017), with a specific focus on the impact of model resolution on the CH4 simulation and the goal of better understanding the source of the biases identified in Stanevich et al. (submitted)." Consequently, it is in the companion paper where we discuss the implications of the bias for flux inversion analyses and ways of improving the model. One conclusion in the companion study was that because of the magnitude of the errors at 4° x 5°, "we do not recommend the 4° x 5° GEOS-Chem model for CH4 inverse modelling." If one wants to use the 4° x 5° model for flux estimation, the weak constraint approach would offer the best means of mitigating the bias, instead of trying to do so through the R matrix. In the companion paper we also suggested that one way to improve GEOS-Chem and reduce the vertical transport bias was "by archiving and globally remapping the native resolution horizontal [air mass fluxes] in order to drive advection at the coarse resolution instead of calculating the horizontal [air mass fluxes] from the coarse-resolution wind fields."

**3. what does the proposed methodology bring to the assessment of the model errors? In the discussion of the results of this study, it is not clear what it brings compared to previous studies: the errors assessed in this paper seem to be already well-known through other methodologies.**

As we discussed above, previous work had identified a stratospheric bias in the CH4 simulation, but previous work in the troposphere had focused on the simulation of 222Rn, 210Pb, and 7Be. The weak constraint scheme enabled us to characterize the structure in the model bias in CH4, which motivated the analysis presented in the companion paper, and in which we quantified the contribution to the bias that arises from diagnosing the vertical transport from the horizontal winds and from the loss of eddy mass flux in the low-resolution meteorological fields. The focus in this manuscript is just on the assimilation results. We refer the referee to the companion paper for the discussion of the implications of the bias identified here.

**Specific comments**

General

Beware of the use of 4D-Var: it should be used only in cases when the problem is actually 4 dimensional. This is the case in meteorology, where the problem is on the initial conditions. Usually, it is not the case for flux inversions, which are problems on the boundary conditions: the relation between the 3D maps of fluxes through time is not taken into account in the model (only the relation of concentration fields through time is). It would be best to use only "variational" and drop the 4D in these cases of flux inversions.

Please see our general response above regarding the issue as to whether this approach should be called 4D-Var.

**Abstract**

• p.1, I.6-7: "capable of differentiating the vertical distribution of model errors": what does differentiating mean here?

We changed this to read "capable of providing information on the vertical structure of model errors."

• p.2, l.2: "indicating the presence of resolution-dependent model errors": do you mean errors directly linked to the parameterizations in the physical core of the model or representativity errors also?

These errors are due to the way the vertical transport is diagnosed from the low-resolution wind fields. A detailed discussion of this is presented in the companion paper.

• p.2, I.4: "However, a major limitation of this approach is the need to better characterize the specified model error covariance in the assimilation scheme": this is true for all data assimilating systems in our domain. What does this study bring to this issue?

The referee is correct. This is an issue for all data assimilation systems. We highlighted this in the abstract and in the conclusions to make it clear that this remains an issue. The weak constraint scheme offers no novel insights into characterizing the covariance matrices.

**Section 1 Introduction**

• p.2, I.21-22: "the impact of biases in chemistry and transport are often neglected": this is not exactly true, a lot of studies try and deal with biases in various ways e.g. debiasing previous to the inversion itself, specifying adapted R matrices, etc. Please look deeper into the available studies.

The referee is correct. We have modified the text to read that "the impact of biases in chemistry and transport are often neglected or accounted for using various ad hoc approaches."

• p.2, I.32: " In contrast to the traditional "strong constraint" (SC) 4D-Var method, the WC scheme does not assume that the model is perfect": same remark as above, the so-called SC method does not imply that the model is assumed to be perfect, it implies that all errors are taken into account in the R (and B) matrix so that what is not is "perfect".

Please see our general comment above regarding the assumption that the model is perfect in strong constraint 4D-Var.

• p.3, l.16-17: "Highly accurate aircraft CH4 profile measurements would be an ideal source of information, but they are limited in space and time.": what about aircores?

AirCores provide excellent information on the vertical distribution, but they are still limited in space and time. We have included AirCores in the revised text.

**Section 2 Data and Methods**

• p.4, l.11-12: "due to the dependence of wetland emissions on the meteorological fields": do you mean that wetland emissions are actually recomputed from the references provided above with the regridded GEOS meteorological fields? Or computed on-line in GEOS-Chem?

The wetland emissions are computed online in the version of GEOS-Chem used in the analysis, but, as described in the manuscript, this results in slight differences in the emissions between the  $2^{\circ} \times 2.5^{\circ}$  and  $4^{\circ} \times 5^{\circ}$  resolutions (due to small differences in the regridded meteorological fields). To ensure consistency in the emissions between the two resolutions in our analysis we regridded the emissions from the  $4^{\circ} \times 5^{\circ}$  simulation to  $2^{\circ} \times 2.5^{\circ}$ .

• p.4, l.21: "the period of 1 February 2010 to 31 May 2010": this is a very short period of time, which does not allow for one full seasonal cycle for mid-latitudes. Why not work on a full year?

We are interested in assessing the utility of the weak constraint 4D-Var scheme in capturing the model bias. To do this there is no need to run the model over the whole seasonal cycle. However, a longer analysis would provide a means of examining seasonally varying biases and evaluating the impact of the seasonally varying observation coverage of GOSAT on the ability of the scheme to capture the biases. Such a study is broader in scope than the work presented here, but would be an interesting follow-up study.

• p.4, I.22-25: "5.5 years ... initial condition for the analysis period": this seems a bit convoluted. Why not run a spin-up of about 9 years (life time of CH4) OR assimilate data to obtain initial conditions?

We did assimilate GOSAT data to obtain optimized emissions for the initial conditions. As we stated in the manuscript, we ran without data assimilation for "5.5 years until July 2009. From July 2009 to January 2010 we assimilated the GOSAT Proxy XCH4 retrievals (Parker et al., 2015) to obtain monthly mean emission estimates at  $4^{\circ} \times 5^{\circ}$  resolution. The optimized emissions were then regridded and used to perform forward model simulations at  $2^{\circ} \times 2.5^{\circ}$  resolution for the same period from July 2009 to January 2010. The updated model fields on 1 February 2010 at both model resolutions were taken as initial condition for the analysis period." Thus, the initial conditions on 1 February 2010 are based on the a posteriori CH4 from optimized emissions obtained from the assimilation of GOSAT data from 1 July 2009 to 31 January 2010.

 p.5, I.22-25: "The use of the alternative CO2 fields did not change any of the findings about model errors in our study. There may still be unidentified biases in both retrieval products. However, the fact that both CO2 fields were obtained using different methodologies gives us confidence in our results." Were both fields used for all the following results? But only one set of results is presented and there is no comment about a sensitivity test. Also, why

**not use the default field? How was it not satisfying?**

Our assumption was that the use of  $CO_2$  fields from GEOS-Chem would be preferable for assimilation of the proxy XCH4 in GEOS-Chem, as the structures in the modeled  $CO_2$  would be consistent with those in CH4. However, as we discussed in the manuscript, the differences in the resulting XCH4 fields were small, with differences typically less than 3 ppb. Not surprisingly, we found that these small differences had no consequential impact of the initial assimilation with the two data sets. It demonstrated that the proxy approach works well. Instead of reverting back to the original  $CO_2$  fields, we used the GEOS-Chem  $CO_2$  fields for all of the analyses presented in the manuscript.

**• p.5, l.30-31: "However, we expect vertical structure to emerge from atmospheric transport patterns": what about the OH field patterns (e.g its vertical structure)?**

Because the lifetime of  $CH_4$  is long, we would not expect much vertical structure in the errors in the  $CH_4$  field due to OH. This can be seen in Figures 8 and 9 in the original manuscript (which are Figures 3 and 4 in the revised manuscript) for the OSSE in which we turned the OH sink during the assimilation period. The chemistry related bias was fairly uniform throughout the troposphere.

 p.6, I.4-6: "Such precision could be enough in many regions of the world to improve knowledge about CH4 a priori surface emissions. However, the presence of potential model errors significantly undermines this assumption." This is not clear: which regions? Improve how? By decreasing the uncertainties by how much? What would be the required ratio between the model errors and the precision and the expected improvement of knowledge?

Regions such as North America, where Sheng et al (2018) showed that enhancements in XCH4 above the background in North American are about 10 ppb (as measured at Lamont). With a random error of about 12 pbb on each retrieval, assimilation of the GOSAT data should provide constraints on regional emissions when they are aggregated in space and time. We now explain this in the revised manuscript. It is unknown what would be the "required" ratio, but clearly one would expect the emission signal to exceed the measurement noise (which will depend on the degree to which the data are aggregated).

**Reference**

Sheng, J.-X., et al.: 2010–2016 methane trends over Canada, the United States, and Mexico observed by the GOSAT satellite: contributions from different source sectors, Atmos. Chem. Phys., 18, 12257–12267, https://doi.org/10.5194/acp-18-12257-2018, 2018.

**• p.6, l.19-23: why use only flasks and not continuous measurements?**

The flask data are widely used by the GHG inverse modeling community, so we chose to use them in our analysis. Given our interest in correcting the vertical distribution of CH4 in the assimilation, we focused more on incorporating the HIPPO, TCCON, and ACE-FTS data in the evaluation, rather than additional surface data.

• p.7, l.2: "Retrievals are bias corrected based on comparisons with calibrated aircraft and AirCore profiles." Which aircraft profiles? Be sure not to use them in the assimilation if the validation data is to be kept totally independent. Why not use also aircores (others than

**the ones used by TCCON) as validation data?**

We refer the referee to the Wunch et al. (2015) paper for details of the TCCON validation. It is not within the scope of our modeling study to discuss the details of the TCCON validation. In our assimilation, we only assimilated GOSAT XCH4 data.

As regards the aircore data, they are another useful validation data set. We did not consider them when we began the project because we were focused on finding validation data with a more global distribution. However, given the increasing availability of aircore data, if we were starting the project today, we would include aircore data in the validation data set.

• p.7, l.27: "the adjoint forcing commonly used in 4D-Var": please clarify what "adjoint forcing" means.

"Adjoint forcing" is often used to describe the gradient of the cost function. Here in the text, we drew attention to the fact that our use of the term "forcing" is different from the traditional use of the expression.

• p.7, l.21-22: "This is the assumption that is employed in standard 4D-Var, which is also referred to as "strong constraint" 4D-Var because the model trajectory is used as a strong constraint in the optimization." Same remark as in the General comments about the use of "4D-Var": the concentrations are linked through time but not the fluxes.

Please see our general comment above regarding the use of 4D-Var to describe the assimilation scheme.

p.7, I.29-30: "The 4D-Var problem to estimate surface emissions is transformed into a 3D sources and sinks estimation problem": same remark as in the General comments: flux inversions are generally not actual 4D-Var and the cost function used is the same as what is shown after, but for the Q term. Does anything prevents Q from being included in R or in B? Please clarify the mathematical and technical differences between SC and WC.

Please see our general comment above regarding issue as to whether the flux inversion problem is a 3D problem.

We have added the strong constraint cost function to the manuscript and additional text in Section 2.3 to better explain the differences between weak and strong constraint 4D-Var as used in our analysis.

• p.9, l.9-14: how long does it take to run for practical cases?

It takes about 12 days to complete 25 iterations of the 4-month weak constraint assimilation at a resolution of  $2^{\circ} \times 2.5^{\circ}$ .

• p.9, l.18-23: "For each WC inversion ... about 10 ppb": this is a good idea but the validation data are not independent anymore since information contained in them is actually used in the inversions. Please clarify how this issue is dealt with when comparing to the data for validation.

We do not assimilate any of the validation data. We only assimilate the GOSAT XCH4 data and compare the resulting a posteriori fields during the 4D-Var iterations (as the cost function is minimized) to the validation data (i.e., the in situ, TCCON, and HIPPO data).

• p.10, I.3: "Therefore, we did not attempt to characterize global pattern of model errors on shorter time scales": even with a period of three days for GOSAT data, information is available at shorter time scales e.g. in the GOSAT data and in the meteorology. Would this not make it possible to characterize patterns of model errors at shorter time scales?

Even with global coverage every three days, there are still significant gaps in the GOSAT observational coverage due to cloud cover, for example. It is possible that we could estimate the model errors at shorter time scales than three days, but we have not investigated this.

• p.10, I.4-6: "Little is known about the a priori structure of the model errors": this is not what appears from discussions and references cited afterwards: at least some elements such as the role of the horizontal resolution or the tropo-strato gradient are known. Is this information not usable in the inversion framework?

The stratospheric bias in CH4 was known. However, it was unknown what was the impact on CH4 of the weakened vertical transport identified by Yu et al. (2017) in their 222Rn, 210Pb, and 7Be analysis.

• p.10, l. 9-end and p.11, l.1-2: the issues described here are the same as when building the R and B matrices in the so-called SC case. Therefore, the exploration of the nature of errors in the modelled CH4 uses the strong assumption of a diagonal Q. What are then the advantages of this methodology compared to the usual R and B matrices in the so-called SC, with various set-ups for R for example?

As we discussed above, this weak constraint method of capturing the model error is consistent with the 4D-Var formalism. We have not explored the utility of trying to account for the model error using the R and B matrices in the context of the strong constraint assimilation.

• p. 11, I.9-11: "Therefore, we considered a uniform structure for Q to be a satisfactory assumption for this initial assessment of model errors in the context of the WC 4D-Var analysis": is this statement justified by "expert- knowledge"? Is it not possible to design sensitivity tests to assess the impact of this strong assumption?

As a follow-up analysis, we are examining the impact of a non-uniform Q on the weak constraint assimilation.

• p.12, l.9: "we believe that the OSSEs should reveal the best performance of the WC method": it is a bit dangerous to show the performances of a methodology only in best-case scenarios since the application to realistic cases may show the tool to be very limited.

The text here was referring to the fact that most of the OSSE results are presented for March, which is the middle of the assimilation window, when one would expect the 4D-var scheme to provide the best estimate of the state. This would the case even with real data. We actually chose extreme model biases for the OSSEs that we expected would be challenging for XCH4 data to help mitigate.

• p.12, l.15: "for the real world applications, we expect less extreme model errors": would the method proposed here be able to characterize the errors if they are smaller? See also comment above.

Please see our response to Referee #2 above (on Page 8).

 .12, I.21-22: "We also conducted SC 4D-Var assimilation experiment for comparisons with the WC approach in the OSSE with biased surface emissions." see General comments. Without more details, it is not possible to understand the differences between WC and SC. If SC contains less information than WC, the comparison is not really meaningful and interesting.

The weak constraint and strong constraint assimilations are assimilating exactly the same observations. We have added a description of the strong constraint assimilation (including the SC cost function) in Section 2.3 to make this clear. Nevertheless, we have removed this experiment since it was not too informative, as noted by Referee #4.

**Section 3 Results**

• p.13, l.26 - p.14, l.4: how do the sensitivities in the model compare to the errors in the actual GOSAT data?

The sensitivity is of XCH4 with respect the CH4 in the model state. We cannot directly compare them to the GOSAT precision. One would have to define a given perturbation in the state and ask how that perturbation, projected through the sensitivity, compares to the GOSAT errors.

• p.14, l.7-8: "when using the SC method, we implicitly supply the assimilation with knowledge about the source of the bias": how?

In this OSSE, as noted by Referee #4, we are imposing all of the bias in the surface emissions, and by design, the strong constraint assimilation assumes that all of the bias is at the surface emissions. As a result of this aspect of the OSSE, and to shorten the manuscript, we have removed this OSSE from the manuscript.

• p.14, l.11-12: "Due to weak vertical sensitivity of the pseudo-data, it is difficult for the WC 4D-Var method to mitigate strong localized vertical bias." The low sensitivity is a problem for data assimilation in general. What is particular to WC here?

We are not claiming that this is particular to the weak constraint assimilation. We are only noting that because of the limited vertical information in the XCH4 data, the assimilation cannot mitigate strong localized vertical bias.

• p.14, l.13-14: "Instead, it compensates for the bias by applying relatively weak CH4 state adjustment of the opposite sign in the column of the atmosphere above, particularly in the stratosphere": what are the consequences of the creation of such a dipole?

We have not examined the consequence of this in the OSSE. We anticipate that it would not be too consequential since the largest bias is in the middle stratosphere, which will have a minimal impact on the total column.

**• p.14, l.15-20: the additional information explicitly provided to WC could also be used in SC in the covariance matrices R and B.**

If one insists in using strong constraint 4D-Var, given the identified model bias, one could try to tune the R and B matrices to improve the performance of the assimilation, but it would be preferable to fix the model to remove the identified model bias.

• p.14, I.33-34: "GOSAT retrievals possess sensitivity to biases in vertical transport and can distinguish...": this is not a property of the GOSAT data as such but of the whole data assimilation framework. Maybe "GOSAT retrievals contain information on the vertical transport, which can be used in our set-up to distinguish..." would be clearer.

We have changed this to "GOSAT retrievals contain information to enable us to capture vertical transport bias even when..."

• p.15, l.11-12: "we do not expect chemical biases to be as strongly localized as the biases associated with emissions and vertical transport": why?

Because of the long lifetime time of CH4, we do not expect vertical transport to provide localized biases. This can be seen in Figure 8 in the original manuscript (Figure 3 in the revised manuscript), in which we turned off the chemical sink of CH4.

 p.15, l.30-32: "The perfect observing system would completely remove the initial condition bias at the start of the assimilation period (on February 1). However, what is shown on February 1 is just an 8% reduction in the bias in each of the eight regions, relative to the a priori, with the rest of the bias propagated onto the assimilation period". What are the consequences of this?

The results suggest that a "spin up" time of about two months would be desirable to avoid the impact of initial condition biases on the analysis. As a result, as noted at the beginning of Section 4.1, we only examine the stratospheric bias in May 2010, to avoid any influence from the initial conditions. To shorten the OSSE section, as requested by Referee #2, we have removed Figure 11 from the revised manuscript and instead briefly summarize the key result in the text.

• p.16, l.3-4: "This suggests that additional vertical correlation between forcing terms in the stratosphere would be beneficial to accelerate convergence in the stratosphere." Why not test the sensitivity to such correlations?

As we discussed in our response to Referee #2, we chose the simplest form of Q with which to evaluate the initial performance of the scheme. We are now doing experiments to quantify the structure in Q, and that will be the focus of a future manuscript.

• p.16, l.11: "residual high latitude bias, which resembles noise or bias in the GOSAT observations." Is this due to the period of interest being winter in the Northern hemisphere?

It is unclear what is the source of this residual bias. One would expect larger errors in the GOSAT retrievals at high latitude in winter/spring, but we have not investigated this.

**• p.16, l.19-20: "the SC assimilation leaves significantly larger residual biases." What is the impact of these on the optimized fluxes and on the uncertainty reduction?**

The larger residual biases are due to the inability of the flux adjustments to mitigate the model bias. We have not tried to quantify what is the impact of these residuals on the fluxes since we can mitigate the biases using the weak constraint scheme.

• p.17, section 3.2.1: the TCCON and NOAA data are not actually independent from the WC (see above). How is this dealt with? If they are actually independent from the SC, the comparison between SC and WC against the fit to these data is not meaningful.

We do not assimilate the TCCON or NOAA data. These data are used only to evaluate the assimilation. We assimilate only GOSAT data, and both the weak constraint and strong constraint schemes assimilate exactly the same data.

• p.18, l.11: "similar errors": similar to what? The sentence is not very clear to me.

We mean that the errors may be similar in both the  $2^{\circ} \times 2.5^{\circ}$  and  $4^{\circ} \times 5^{\circ}$  simulations. We have reworded this on Page 16 (line 21) in the revised manuscript.

• p.18, l.9-13: these sentences seem a bit convoluted. Is the idea to state that a station such as IZO sees the upper troposphere and is best compared to the model's upper troposphere?

We are arguing that the improved model agreement at IZO with the weak constraint 4D-Var scheme is due largely to improvements in the column in the upper troposphere and lower stratosphere, since neither model resolution can capture the topography for the island, which will result in lower tropospheric model transport errors in the vicinity of the island. We have rewritten the text here on Page 16 in the revised manuscript.

• p. 18, I.27: what about the impact on GOSAT retrievals of the still long nights around the North pole during the period of interest?

We are unsure as to how polar night impacts high latitude GOSAT retrieval just outside the terminator. This is beyond the scope of the manuscript.

**Section 4 Discussion of Model Biases**

• p.19, l.22-23: "a stratospheric bias introduced in the system through the initial conditions ": is there no other way than the ICs to introduce a bias in the stratosphere?

There are numerous ways the stratosphere could be biased. We intentionally introduced an initial condition bias to assess the ability of the system to remove such a bias.

• p.19, l.29-30: "it shows that the SC assimilation attempts to correct the positive highlatitude stratospheric CH4 bias at the expense of surface emissions": this seems to be a direct consequence of the set-up of the SC inversion. More details on this inversion are required to discuss its results. It is a direct consequence of the fact that the strong constraint assimilation assumes that all the model bias can be mitigated through adjustment of the surface fluxes. We have added a more detailed description of the strong constraint assimilation in Section 2.3.

• p.21, l.6-14: can you deduce some recommendations for improvements in the model? If not, how can this knowledge be used for setting up inversions for fluxes?

See our response to general comments 2 and 3 above.

• p.21, l.15-23: same questions as above: recommendations for improvements in the model? Information for flux inversion?

See our response to general comments 2 and 3 above.

• p.22, l.25-26: what could be concluded? Recommendations for improvements in the model? Information for flux inversion?

See our response to general comments 2 and 3 above. The assimilation results presented here motivated the work in our companion GMD paper, in which we examined the sources of the model errors and made recommendations for improving the model.

• p.23, l.13-14: "Finally, the results strongly suggest that the WC assimilation and the GOSAT observations have the potential to diagnose transport errors at both model resolutions" What would prevent the assimilation of data to bring information on errors at any resolution (if the data are relevant)?

The referee is correct. If the data have sufficient information it should not matter. We have removed the statement from the manuscript.

**Section 5 Conclusions**

• p.24, l.1-2: "However, characterizing these correlations will be challenging": is it possible at least to design sensitivity tests to explore various possibilities for these error correlations?

Yes, and that will be the focus of a future manuscript. The current manuscript, together with its companion GMD paper, represents a significant effort to adapt the weak constraint scheme for  $CH_4$  assimilation. Developing these correlations and expanding the manuscript to incorporate sensitivity tests with the correlations would have made the manuscript unwieldy.

• p.24, 1.5-6: "Initial comparisons suggested that GEOS-Chem was affected by biases not solely related to discrepancies in surface emissions." This is not very informative as it is well known and flux inversion would be relatively straightforward otherwise.

Yes, it is well known that models have biases, and in the subsequent sentences we discussed the nature of these biases in GEOS-Chem and the fact that the weak constraint 4D-Var scheme was able to mitigate them.

• p.24, l.13-14: "Meanwhile, the results showed that running the a priori model at 2x2.5resolution produced better agreement with TCCON observations than the a posteriori

**fields from the SC 4D-Var surface emission optimization at 4x5." What can be deduced from this result?**

As we concluded in the companion paper in GMD, without correcting for the biases, "we do not recommend the 4° x 5° GEOS-Chem model for  $CH_4$  inverse modelling."

• p.24, l.31-32: "glint measurements": a lot of other data could be assimilated, not only satellite data related to the concentrations of a given species.

Yes, there are a lot of other data, but satellite observations give the best global observational coverage. We stressed the need for glint data here, in the context of the shortwave infrared measurements, to obtain a more global coverage, particularly over the oceans.

• p.25, l.2: "if the model were assumed to be perfect, as is the case in SC 4D-Var": see General comments: this is not so simple.

Please see our general response regarding the assumption that the model is perfect. We now state that "...if the evolution of the model state were assumed to be perfect."

 p.25, I.7: "Potentially, any CTM may be improved if the signal from the surface emissions can be separated from other model errors." This is only relevant if the objective is to optimize fluxes. A given CTM and set up may work very well with errors compensating one another for other objectives (e.g forecasting). Be more specific on what improvements could be made, how and to which purposes.

Our overall interest here is in optimizing fluxes. We have added text to make this clear.

• p.25, l.14: "regional-scale analysis at higher spatial resolution": how in practice? The link to what it is used for in this study is not plain for me.

By optimizing the state, we can provide a 4D field of CH4 that can be used as boundary conditions for a regional model.

**Technical corrections**

• p.10, l.33: "the errors in the model CH4 simulation" -> the errors in the modelled CH4 concentrations?

We have changed this to "explore the structure of the errors in the model."

• p.11, l.19: "The sensitivity of the GOSAT observations to the modelled state" -> The sensitivity of the equivalent of GOSAT observations to the modelled state

We have removed this text to shorted the OSSE section, as requested by Referee #2.

• p.15, l.13: "in the Fig.8" -> remove "the"

Removed.

• p.18, l.10: "inland" -> island?

Fixed.

• p.18, l16: homogenize including/excluding Sodankyla.

Fixed.

• p.22, l.9: "lofted" -> lifted

Changed.

**Tables and Figures**

• Tables 1, 2: please use consistent names for the columns in the legend and table.

Fixed.

• Fig. 4: the black boxes are not so easy to see: maybe use a very different color (try pink?)

We have removed this figure from the manuscript.

• Fig. 15: do not repeat in the legend what appears in the graph itself so that is it easier to read and may be kept on one page.

We have shortened the description in the figure caption so that it is easier to read.

• Fig. 16: reduction (third column) may be easier to evaluate in % and absolute value.

We believe that the change in the absolute bias provides the reader with a meaningful sense of the improvement that the assimilation offers.

**Anonymous Referee #4**

**General Review Comments:**

This is a significant and important body of work and I believe the community will find it very interesting. The main limitations I see are in terms of the figures which don't always back up the assertions in the text. Sometimes, in the case of Fig6/Fig7, they clearly support the text but seem to fail because of lack of difference plots or poor choice of color scales. The text is sometimes hard to follow because of the multiple terms being used, eg. state/forcings/3D CH4 adjustments and parameters vs. surface fluxes. After providing the theory, it might be wise to work with common terminology.

Furthermore, I particularly found the assumed variability of the "forcings" in time and space, somewhat hard to follow. In summary though, the work is quite interesting and novel in its handling of the main source of error in atmospheric trace gas inversions (transport).

We have shortened the manuscript by deleting some of the material in the OSSE section. We have also tried to improve the description of the experiments to make it easier for the reader to follow the discussion.

**Specific Comments:**

**caption Figure 2: "and smoothed with the GOSAT averaging kernels" The authors should give the exact equation they use to calculate modeled XCH4 from their model: is the prior CH4 profile assumed in the retrieval used in this calculation, in addition to the averaging kernel, or not?**

We have added the equation (Eq. (1) in the revised manuscript) describing the GOSAT retrievals in terms of the averaging kernels and a priori profile.

**Figure 6: Fig 7 provides a motivating summary of Fig 6. However, looking at Fig 6 relative to its colorbar, it is hard to see that the two even provide the same information. I'd really consider adopting a more dynamic color palette for this image.**

Our intention with Figure 6 was to give the reader a sense of the spatial patterns of the imposed model bias and the resulting correction. Because we are showing the differences at the surface and in the upper troposphere, and vertically in the tropics and mid-latitudes, it was difficult to find a scale that will allow the reader to quantitatively compare the plots. Thus, we included Figure 7. We have added text to emphasize that the focus of Figure 6 (Figure 1 in the revised manuscript) is the spatial patterns of the model error corrections.

p6 I10: "...with smaller [modeled] XCH4 in the SH..." A difference plot would be helpful in Figure 2. I am having a very hard time seeing how XCH4 is lower in the SH in the model (top panel) than in the measurements (bottom panel) – the opposite appears to be the case, to my eye.

See the response below.

**p6 I11: I agree that China, India, and equatorial Africa are higher in the model, but South America appears to be lower, at least over the Amazon. Again, a difference plot in Figure 2 would help.**

We apologies for the confusion here. The discussion regarding Figure 2 was about the spatial patterns of the XCH4 field in both the model and observations. The XCH4 is smaller in the southern hemisphere than in the northern hemisphere in both the model and observations. Similarly, there is enhanced XCH4 over South America and equatorial Africa in both the model and observations. We have removed Figure 2 to shorten the discussion in this section, as requested by Referee #2.

p7 I29-on: "The 4D-Var problem to estimate surface emissions is transformed into a 3D sources and sinks estimation problem..." I think you ought to be a bit more precise with the wording here. The 4Dvar method in general can be used to estimate both surface sources/sinks and 3D fields, either using the strong- or weak-constraint dynamics. I think what you want to say is that for your application, which in the past was solving only for surface fluxes and not 3D fields, you are now allowing this 3D forcing term to be solved for, as well. You might want to mention here that you are not too concerned with what this 3D field represents in your application – it could be a chemical source/ sink, a correction to the 3D concentration field, or a dynamical error. That comment would help those who are

**accustomed to thinking of this term as being solely a dynamical error term in the weakconstraint approach.**

We have added text clarifying this point on Page 8 (lines 2-3) in the revised manuscript.

p8 L15: There is a spurious extra term in the cost function here that should lead to some "double counting" of the dynamical errors. To be specific, the last two terms in the cost function both involve the dynamical mismatches, while only one is needed, and that term ought to be: ... + [-Gu\_i]^T [Q\_i]^-1 [-Gu\_i], which equals ... + (x\_i - M(x\_i-1,p))^T [Q\_i]^-1 (x\_i - M(x\_i-1,p))

If we note that lambda\_i =  $[Q_i]^{-1} (x_i - M(x_i-1,p))$ , then there is no need to carry around the additional variable u\_i, and u\_i can be calculated from lambda\_i as

**G u\_i = - [Q\_i] lambda\_i**

**Similarly, it is not necessary to have equations for the partial of L with respect to both lambda\_i and u\_i.**

In the Lagrangian, the first term is the observational mismatch, the second term is the a priori constraint on the emissions, the third term is the a priori constraint on the state corrections, and the fourth term is the constrain from the dynamics. We are not "double counting" the influence of the dynamics.

The Referee is correct, we do not need the partial of the Lagrangian with respect to  $\lambda_i$ . We have removed this equation.

p10 L5: "... a priori estimates of the model errors were set to zero." Just to be sure here, you are setting the actual initial model errors themselves equal to zero (u\_i=0) and not the assumed uncertainty in those same errors (Q\_i=0), correct? Because setting Q\_i=0 would be equivalent to reverting back to the hard dynamical constraint. Writing this out with the mathematical symbols would eliminate any uncertainty the reader might have on that score.

Yes, we are setting u = 0. We now write this out mathematically on Page 10 (line 6).

p10 second paragraph: This discussion of the relative weighting of the forcing versus emission parts of the solved-for control vector points to something that could be simplified in this approach. It can be shown that the forcing vector -Gu\_i at any iteration is simply [Q\_i] lambda\_i – that is, the adjoint state vector times the assumed model error covariance matrix. The forcing vector does not need to be solved for in the control vector – it is already solved for in the strong-constraint 4Dvar, essentially, since lambda\_i is solved for. All that needs to be done to implement the weak-constraint version is to add the forcing vector (=[Q\_i] lambda\_i) onto the state at each step during the forward runs (although saving lambda\_i at fine temporal resolution, at the timestep of the model, for example, can take a lot of memory and I/O time). Once this is recognized, it is clear that the full parameter space of the forcing vector can be solved for, up to the temporal resolution limits just mentioned. Since it is not too clear what the magnitude of [Q\_i] ought to be, some experimentation with the relative

**weighting between the forcing and emissions parts of the cost function is probably still needed, even with the simplification just noted.**

We thank the referee for the suggestion. We agree that once the sensitivities have been calculated in the strong constraint case, it is straightforward to extend the model to estimate the forcing terms in weak constraint 4D-Var. We will consider the implications of the referee's suggestion for simplifying the way we have implemented the scheme in the model.

p10 L32-34: "Instead, we use the WC 4D-Var method to optimally constrain the CH4 state and explore the nature of the errors in the model CH4 simulation." Actually, you are not really solving for an optimal estimate of the CH4 state (the 3D CH4 field at each timestep), because you have not put this in the control vector that you are solving for. This state evolves according to your model M and has dynamical constraints upon it. What you are really solving for is some sort of error term on the state, which might be thought of as a dynamical error, or a 3D source/sink (e.g. a chemical one) if one is not modeled, or an error on a modeled 3D source/sink. Since the measurements have error on them, this also allows large measurement errors to be given less importance in the problem (the measurement error is turned into a large forcing error and its effect is on the actual parameters solved-for is lessened).

The referee is correct. This is a more accurate description of what we are doing. We have modified the text to state that we are using "the WC 4D-Var method to constrain the 3D corrections to the CH4 state."

p10 L33-34: "We performed two types of inversions: "full state assimilation" and "flux+state assimilation"." Point back to your description of these two cases in Section 2.3 here. I had already forgotten that you described them earlier by this point. Also, it wouldn't hurt to say explicitly that the "full state" case solves only (or mainly) for u, while the "flux+state" case solves for both u and p. Spell it out, especially since the state is defined as "x" earlier, and you don't actually solve for the full 3-D field "x" in your control vector, no? I guess you get the 3D CH4 fields by adding on the dynamical errors that you do solve for onto the state using equation (2), right?

Thank you for the suggestion. We have modified the text as suggested.

p11 L13: This would be a good place to mention that, in all these OSSEs, the same transport model was used to generate the truth as was used in the inversion. In other words, these are all "perfect model" experiments in all respects except the isolated error signal looked at in each experiment. This is important to say, because most readers will be thinking of the weak constraint as a way to handle general model errors, and might be thinking that you used a different transport model for the generating the truth and doing the inversion (I will actually advocate adding an OSSE of this sort below in the emissions case.)

We have added text stating this on Page 11.

p13 L29-31: Isn't the greater sensitivity in the upper troposphere / lower stratosphere also due to the fact that the GOSAT Xch4 averaging kernel is weighted most heavily in the mid- to

**upper-troposphere (it being a thermal IR measurement)? A small figure showing what the GOSAT XCH4 averaging kernel actually looks like might be valuable here.**

We are using the shortwave infrared retrievals (see Section 2.2.1), so the averaging kernels are heavily weighted toward the lower and middle troposphere. Since the manuscript is already too long (with 19 figures in the submitted version), and because plots of the averaging kernels are in the published literature, we would prefer not to include such a plot in the manuscript. We added the Yoshida et al. (2011) reference in Section 2.2.1, which contains plots of the averaging kernels.

p14 L9: "The results confirm that the SC 4D-Var method better removes CH4 biases due to emissions." This is misleading. Yes, in this setup the SC does better than the WC because it puts all the measurement information into the cause of the difference, the emissions, by design (it doesn't solve for the 3D forcing corrections) as you have noted. But in general case of when there are model errors, the WC case should solve for emissions more accurately than the SC case. It would be really useful if you could have done an additional OSSE here in which you introduced emissions errors AND dynamical errors in the truth, then tried to estimate both using the SC case and the WC "flux+state" setup. That ought to show the WC case doing better, since it would correctly partition the dynamical and emissions errors to the forcing terms and emissions parameters, whereas the SC would incorrectly attribute the dynamical errors solely as emissions errors. You should use a completely different transport model in generating the truth from what is used in the inversion to make this case realistic. The dynamical errors in Q should be based on the differences between the two transport models. I think that that new OSSE would demonstrate the benefit of the WC approach for that case that a lot of readers care about: how much their surface emissions estimates are degraded by model error when using the SC approach.

The referee is correct. In retrospect it is not surprising that the strong constraint performs better when we impose all of the bias in the surface fluxes. Since this OSSE is not that informative, we have removed it from the manuscript, in the interest of shortening the manuscript. The suggestion of the referee to use a different transport model to show that the strong constraint assimilation will incorrectly attribute the transport errors to flux errors would have been helpful earlier in the analysis. In our companion paper we effectively show this. We found larger flux corrections with the strong constraint assimilation at 4° x 5° than at 2° x 2.5°, which is consistent with the strong constraint assimilation incorrectly adjusting the fluxes to compensate for the larger model errors at 4° x 5°.

**p16 L15-18: I cannot tell from Fig 12 and 13 that the high latitude a priori bias is smaller in 2x2.5 compared to 4x5. Similar question for the model bias reduction. It may just be that the color scale saturates so quickly in the images? Are they zonal avg summary stats you could provide to back this up?**

This can be seen more clearly in Figure 2 and 3 in our companion paper in GMD. In Figure 3 in the GMD paper we plot the zonal mean differences of the  $4^{\circ} \times 5^{\circ}$  and  $2^{\circ} \times 2.5^{\circ}$  model with respect to the ACE-FTS data.

p17 L10: "The results suggest that GEOS-Chem a priori CH4 simulation suffered from biases that were not related only to incorrect surface emissions." Since GEOS-Chem is not perfect (i.e. the real world has different transport) this is the most obvious source of the errors.

We agree that this is obvious, but we felt it necessary to point it out nevertheless.

**p18 L5: I find it a bit perplexing that 2x2.5 sims are generally argued to provide better atmospheric transport but perform more poorly against in situ obs, which one would think would be the most sensitive to things like vertical transport. Would you consider this an indication of residual error in surface CH4 fluxes?**

Yes, we believe that this is an indication of residual error in the surface fluxes. Because the constraints that the GOSAT data provide on regional fluxes is limited, the posterior flux field at  $2^{\circ} \times 2.5^{\circ}$  is noisier than at  $4^{\circ} \times 5^{\circ}$ . For example, Maasakkers et al. (2019) found that their inversion of GOSAT data to estimate CH4 fluxes for 2010-2015 had only 128 degrees of freedom for signal on the global  $4^{\circ} \times 5^{\circ}$  GEOS-Chem grid.

Maasakkers, J. D., et al., Global distribution of methane emissions, emission trends, and OH concentrations and trends inferred from an inversion of GOSAT satellite data for 2010–2015, Atmos. Chem. Phys., 19, 7859–7881, https://doi.org/10.5194/acp-19-7859-2019, 2019.

**Tables 1 & 2: These comparisons to independent data are powerful demonstrations that using the weak constraint approach on the GOSAT XCH4 data really does improve the estimated 3D CH4 fields better than the SC approach. Apparently either the transport or the measurements are bad enough that the WC results in a big improvement. If the GOSAT data are badly biased, adding flex in the state trajectory would help things, even if the transport were not too bad.**

Yes, it would help even if the GOSAT data were badly biased. However, based on previous work, we believe that the problem is with the model.

**p22 L30: I get the averaging out of resolved eddy motions causing problems but what are you referring to by "incorrect" regridding?**

This was poorly worded. When we degrade the model resolution, the air mass flux is not conserved from one resolution to another in the regridding. The text now states that the vertical transport is weakened due to "loss of eddy mass flux and air mass flux in the regridding of the meteorological fields."

**p23 L28: "Despite having almost flat averaging kernels in the troposphere..." Show a plot of the GOSAT XCH4 averaging kernel – since GOSAT measures it in a thermal IR band, I would have thought that the averaging kernel would be mainly sensitive to the upper troposphere.**

As we mentioned above, we are using the  $CH_4$  retrievals based on the shortwave infrared (SWIR) measurements, so the sensitivity is relatively flat throughout the lower and middle troposphere. We have added the Yoshida et al. (2011) reference in Section 2.2.1 that the reader can consult for a description of the SWIR averaging kernels.

**Minor Comments: p5 line 13: should this read "and the GHG\_CCI group"??**

We have corrected the text.

**p5 L20: "produced a comparable fit...": to what, the old three-model suite used in the retrievals?**

Comparable to the agreement that we obtain using the  $XCH_4$  data with the original  $CO_2$  fields. We now state that "...a posteriori inversion results using the new  $CO_2$  fields and the original fields generally produced comparable fits to independent..."

**p5 L24: you should say GOSAT "XCH4" here, since that is what you are looking at; "CH4" does better than that, I think.**

We do mean  $CO_2$  fields here; the original LMDZ/MACC-II Co2 fields and the GEOS-Chem a posteriori fields.

**P21 L25-26: Reword these lines, I don't think the CH4 budget is conserved because there is bias in the stratosphere induced by transport error.**

The referee is correct. A transport bias can change the CH4 distribution and thus the overall loss of CH4. We have removed this text.

**P23 l8: add a comma before "are".**

We added the comma.

**In general, I would prefer full descriptions like "the difference between A and B" as opposed to the "bias in A". The term "bias" means a lot of different things, and can relative to another unspecified quantity, and using the full description is always going to be less confusing albeit more wordy.**

We have tried to include the full description where possible, without making the text tedious since the main focus of the analysis is on biases.

---

## Author Response (AR2)

We thank the referees for the additional thoughtful comments on the manuscript. Below, we respond separately to each referee. The comments from the referees are in **bold Calibri** font and our responses are in plain Times New Roman font.

**Report 1 (Referee #3)**

**General comments:**
* * *
**As already stated in the review of the first version of the paper, the effort at assessing the errors in the (chemistry-)transport model before going into flux inversions is necessary and the issue remains too rarely treated adequately in inversion papers. It is therefore a very good idea to present a methodology to tackle this issue. The explanations of the methodology and most of the interpretations of the results, regarding mainly biases in the transport, are clear and interesting and the revisions made after the first review have lead to a still clearer description of the work. The reduced number of figures makes it more focussed and easier to read, which I appreciate a lot.**

**A debate on the terminology has been launched during the first review: using 4D-Var, 3D-Var or variational for the methodology. The authors wish to stick to using 4D-Var and I agree with them that this kind of paper is not the place for deciding on the exact terminology of our domain. Since the description they added makes everything clear mathematically, I think we can drop the debate on the naming of the method.**

**Specific comments:**
* * *
**\*\*Section 2: data and method:**
**- p.8 l.11 and in Eq.5: the way the equation is written is valid for $N$ the number of hourly time steps but actually, there are not N u_i (they vary over 3 days or 1 month, as stated elsewhere) or N Q_i (there is only one as it is constant). Could the equation be written in a more general way so that it would be easy to refer to it directly for any of the experiments?**

The equations certainly could be rewritten to make this explicitly clear. However, it would require changing Eqs. 5, 6, and 10 and adding two additional equations to link the different indices. We are reluctant to introduce these changes at this point in the review process since the equations currently in the manuscript are correct. However, if the Editor believes the manuscript would benefit from making these changes, we will do so.

**- p.8 l.15 in Eq.6: same remark**

See our response above.

**- p.9 l.14: is convergence guarranted?**

Yes. The 4D-Var scheme works and converges, as shown by our results. However, if there are biases in the observations that are strongly inconsistent with the modeled state, or if the error

covariances are poorly specified, then the code could fail to converge to a global minimum, which would be the case with many assimilation methods.

**- p.9 l.22: "the level of noise was estimated to correspond to GOSAT XCH4 uncertainty": do you mean that the point at which the inversions stopped was afterwards found to match the 10 ppb value provided for GOSAT data?**

Yes. We calculated the reduced chi-squared for the model fit to the GOSAT observations and ensured that it was about unity. We have added text to make this clear.

**- p.9 l.26: "R was assumed to be diagonal": and the diagonal was filled from the 13 ppb value given in the previous sentence? An explicit link between the two sentences would make it clearer.**

We have rewritten these sentences to make this clearer.

**- p.10 l.1: "50% uncertainty on CH4 emissions in each surface grid box": how much Tg/y does it lead to? Is it realistic?**

This is applied at the grid box scale and we have not aggregated these errors to the global scale to get a global error in Tg/yr. However, this 50% error is consistent with other estimates used in the literature. For example, Maasakkers et al. (2019) assumed a 50% error for anthropogenic emissions and estimated a mean error of 58% for wetland emissions at the GEOS-Chem 4º x 5 º grid box scale. For non-wetland, natural emissions they assumed an error of 100%. However, in specifying the total error in each 4º x 5 º grid box they capped the errors at 50%.

Maasakkers et al., Global distribution of methane emissions, emission trends, and OH concentrations and trends inferred from an inversion of GOSAT satellite data for 2010–2015, ACP, https://doi.org/10.5194/acp-19-7859-2019.

**- p.11 l.5-6: "to independent observations did not change noticeably": did not change when q increased?**

Yes. We have added "larger" to the sentence so that it now reads "for larger values of q above 50 ppb, the fit of optimized $CH_4$ fields to independent observations did not change noticeably."

**- p.11 l.9: "the WC method was still able to significantly improve": this sentence is not very clear to me: what does the "still" refer to?**

This sentence was unclear. We have changed it to state that "As shown in the experiments described in Sect. 2.4, the WC method was able to improve the model and capture the bias in the $CH_4$ state with q set to 50 ppb."

**- p.11 l.29: "The bias in vertical transport and chemistry": this is for the first two OSSEs, respectively, isn't it?**

Yes. We have modified the sentence to read: "In the first and second OSSEs, the bias in vertical transport and chemistry were introduced by turning off convection and chemistry, respectively,..."

**- p.12 l.1: "and have the freedom to": this part of the sentence is a bit strange. Do you mean that the bias can be placed anywhere in this configuration?**

This was confusing. We mean that the WC scheme has the freedom to determine the location of the bias. We have changed this so that it now states "We configured the WC method to carry out "full state assimilation" (as described in Sect. 2.3) to enable the optimization to determine independently the location and magnitude of the bias in the modeled state."

**- p.12 l.6-7: "We also conducted SC 4D-Var assimilation experiment for comparisons with the WC approach in the OSSE with biased surface emissions.": no OSSE with biased surface emissions is referred to before.**

Thanks for catching this. This OSSE was removed during the last revision. We have fixed the text.

**- p.12 l.22-23: "Short time windows": how long?**

We mean less than 2 days, for example. We have added this to the text.

**- p.12 l.24-25: "for short temporal correlation length scales": here, all matrices are diagonal, so that there are no correlation lengths used at all, is that right?**

Yes, the matrices are all diagonal. The discussion here is about the general implications of choosing short or long forcing windows in the context of the WC scheme.

**- p.12 l.31: "above 750 hPa" = above 750 hPa only or above 750 hPa versus the rest of the atmosphere?**

We mean on all model levels above 750 hPa. We have modified the text to make this clearer.

**- p.12 l.32: does "globally" mean one term for the whole stratosphere?**

We mean on all model levels above the surface (and for each grid box). We have modified the text to make this clearer.

**\*\*Section 3 Results**
**- p.13 l.15: "The state corrections capture the general horizontal and vertical structure of the a priori bias": this seems a bit optimistic when looking at the figure as a whole. Maybe some king of statistical indicator would make it clearer.**

The main feature in the a priori bias associated with turning off convection is excessive $CH_4$ in lower troposphere and a deficit in the upper troposphere, and the state corrections produce a reduction in $CH_4$ in the lower troposphere and an increase in the upper troposphere. We have modified the text so that it now states that "the state corrections capture the general structure of the a priori bias, which consists of excessive $CH_4$ in the lower troposphere and a deficit in the upper troposphere."

**- p.13 l.25: "fewer GOSAT retrievals": a supplementary table with the number of GOSAT retrievals per region and period would be useful.**

We believe that an additional table with the number of GOSAT data in each region in each month is unnecessary as the impact of the varying GOSAT observational coverage on the assimilation is not a focus of the study. This is an interesting issue, but it would be best addressed by a separate analysis.

**- p.13 l.26-27: what about the increase above 100 hPa?**

A similar response was found in the OSSE with biased surface emissions, which was removed from the revised manuscript, and was commented on previously by the referee. As we noted in our response to that previous comment, we have not examined the consequence of this in the OSSE. We anticipate that the increase in the stratosphere would be inconsequential since it will have a minimal impact on the total column. It is important to remember that the assimilation is trying to match the observed total column.

**- p.17 l.8: do the ACE-FTS data and the model always agree on what points are in the stratosphere?**

The model is driven by the NASA GMAO reanalysis fields, which we believe simulates the UTLS well when used at its native resolution. However, when these fields are degraded to 4° x 5° (the resolution used here), errors are introduced in the UTLS, which is one component of the model errors that we are examining in this study and in the companion paper.

**\*\*Section 4 Discussion of Model Biases**
**p.18 l.14: "the asymmetrically larger number of GOSAT measurements in the northern hemisphere." Same remark as above: a table with the number of GOSAT data available in regions discussed in the text would be useful.**

See our response above. However, we note that we are assimilating only data over land, so there is a clear hemispheric asymmetry in the amount of data assimilated in the two hemispheres. We have added a reminder in the text that we are assimilating only data over land.

**\*\*Section 5 Conclusions**
**- p.21 l.29: "artificially introduced biases in emissions, convection,": the part with biases in emissions is not shown any more.**

Thank you for catching that error. We have corrected the text.

**- p.22 l.28 and 30: what so significantly and significant fraction mean here? Can you quantify?**

We have removed this statement.

**Technical corrections:**
**-----------------------------**
**Throughout the text, "a priori" and "a posteriori" are used: shouldn't it be "prior" and "posterior" instead? Note that I am not a native English speaker.**

Both "a priori" and "prior" are used in the literature. Indeed, the ACP English guidelines and house standard states that "Common Latin phrases are not italicized (for example, et al., cf., e.g., a priori, in situ, bremsstrahlung, and eigenvalue)." We prefer to keep these phrases in the text.

**- p.6 l.4: "10 ppm" -> ppb?**

Corrected.

**- p.8 l.2: "surface emissions, however, because of": cut sentence in two "surface emissions. However, because of"**

Changed.

**- p.10 l.9: "the choice of scaling parameter2 -> the choice of the scaling parameter?**

Corrected.

**- p.10 l.34: "as described in Sect. 2.3": the reference is a bit strange since we are in Sect. 2.3; maybe use sub-subsections or paragraphs?**

You are correct. Referring to the section is unnecessary here. We have removed this reference.

**- p.11 l.18: "to produce pseudo GOSAT XCH4 measurements" and l.26 "No noise was added to pseudo-observations": from this I understand that the statistics in R are not used to generate the pseudo-obervations. If so, maybe put all the information about the generation of the pseudo-obs together.**

Thank you for the suggestion. We have moved the text.

**- p.11 l.20: "from one of the four specified sources of model bias": only three are specified above**

Changed to "three".

**- p.12 l.30: "results to vertical extent" -> to the vertical extent**

Corrected.

**- p.13 l.13: "lofted" -> lifted**

Corrected.

**- p.16 l.10: "a prior" -> a priori (or prior)**

We prefer to keep "a priori".

**- p.16 l.24: "leaves a weak positive biases" -> leaves weak positive biases**

Corrected.

**- p.16 l.25: "Mean a posterior inter-station bias" -> a posteriori (or posterior)**

Corrected.

**- p.19 l.29: "forth" -> "fourth"**

Corrected.

**- p.19 l.31: "vertical transports" -> transport**

Corrected.

**- p.20 l.22: "the dimensionality of inverse problem" -> of the inverse problem**

Corrected.

**- p.22 l.10: "slowly varying biases, however, a few stations," -> biases. However, a few**

Changed.

**- p.22 l.27: "5 o, however, the magnitude" -> 5 o. However, the magnitude**

Changed.

**- p.23 l.2: "model transport, however shorter-lived" -> transport. However,**

We prefer to keep the original text.

**- p.23 l.6-7: "incorrectly attributing model errors in vertical transport" to emissions?**

Changed.

**- Figures 1, 3: "units of ppb" -> "ppb"**

Changed.

**- Figure 9: "set of experiment" -> experiments**

Changed

**Report 2 (Referee #2)**

**The authors thoroughly addressed comments by the reviewers and significantly improved the quality of their manuscript.**
**The overall message is much more straightforward and consistent with the figures presented.**
**The shortening of some parts makes it much easier to read and only minor revisions remain before the manuscript is ready for publication.**
**Please find below the list of revisions to apply:**

**- Sect. 2.3, p. 7: even though the WC formulation is now well explained, I would add a sentence stating that an equivalent formulation would be to include the correction term u in the target vector p**

We have added a sentence on Page 8, after Eq. 5, stating that "This approach provides a means of capturing the model errors in the context of the 4D-Var formalism, whereas other approaches may try to account for these errors by including u in p."

**- p.9 l.15: For the convergence criterion, could you please indicate how many iterations are necessary for the different set-ups, for WC and SC?**

It takes about 20 iterations for the SC assimilation and 30-35 iterations for the WC. We have added this to the text.

**- p. 10 l.13: you use L-BFGS-B which does not nicely accommodate very large numbers of unknowns; why not using algorithms common in the community, such as M1QN3 which is designed for large dimensional problems?**

For the work presented here, we simply extended the existing GEOS-Chem 4D-Var scheme, and the 4D-Var optimization in GEOS-Chem uses the L-BFGS-B algorithm.

**- Fig. 5: it would be more illustrative to have both the prior and posterior bias at the end of the period to see how much is corrected by the WC state inversion.**

It is important to remember that the $CH_4$ lifetime is long and, consequently, any bias in the initial conditions will decay slowly. This was why we conducted this initial condition experiment. In the previous version of the manuscript we had a figure showing the temporal evolution of the initial condition bias. That figure was removed in the revision, but we still summarize (in the last paragraph of Section 3.1) the results that were shown in that figure. Specifically, we point out that the stratosphere (above 200 hPa) is region with the longest timescale for removal of the bias, noting "that by the third month the $CH_4$ mass had not fully recovered at these levels." Given the 9-year $CH_4$ lifetime, it is impossible to remove the initial condition bias in less than three months without assimilation, so we do not believe that comparison of the posterior bias with the bias without assimilation would be more informative.